# GenCircuit-RL: Reinforcement Learning from Hierarchical Verification for Genetic Circuit Design

**Noah Flynn** [1]

## Abstract

Genetic circuit design remains a laborious, expert-driven process despite decades of progress in synthetic biology. We study this problem through code generation: models produce Python code in pysbol3 to construct genetic circuits in the Synthetic Biology Open Language (SBOL), a formal representation that supports automated verification. We introduce GenCircuit-RL, a reinforcement learning framework built around hierarchical verification rewards that decompose correctness into five levels, from code execution to task-specific topological checks, and a four-stage curriculum that shifts optimization pressure from code generation to functional reasoning. We also introduce SynBio-Reason, a benchmark of 4,753 circuits spanning six canonical circuit types and nine tasks from code repair to extitde novo design, with held-out biological parts for out-of-distribution evaluation. Hierarchical verification improves task success on functional reasoning tasks by 14 to 16 percentage points over binary rewards, and curriculum learning is required for strong design performance. The resulting models generate topologically correct circuits, generalize to novel biological parts, and rediscover canonical designs from the synthetic biology literature.

## 1. Introduction

Genetic circuit design is a natural target for language-model-based assistance because it combines formal specifications, compositional structure, and difficult search over large design spaces. Synthetic biologists engineer programmable cellular behaviors by assembling standardized parts such as promoters, ribosome binding sites, coding sequences, and terminators into circuits that implement logic, memory, and oscillation (Gardner et al., 2000a; Elowitz & Leibler, 2000a; Nielsen et al., 2016b). These circuits already support applications in biosensing, therapeutics, and bioproduction (Xu et al., 2024; Saltepe et al., 2018; Tellechea-Luzardo et al., 2023; Sedlmayer et al., 2018), but their design still depends on repeated design-build-test-learn cycles and substantial expert intervention (Appendix B.2).

SBOL provides a formal, machine-readable representation for genetic circuits (McLaughlin et al., 2020). It encodes compositional structure and regulatory relationships in a format that supports automated verification; Appendix G.3 gives concrete examples in pysbol3. This combination of formal structure and executable verification makes genetic circuit design a useful setting for reinforcement learning from verifiable feedback (RLVF). The challenge is that generated circuits can fail at several levels: code may not execute, the resulting SBOL may be invalid, the structure may be malformed, annotations may be inconsistent, or the circuit may implement the wrong function. Binary rewards collapse these distinct failure modes into a single outcome. They give the same signal to a circuit that fails at syntax and one that misses only the final functional criterion, which weakens learning on reasoning-intensive tasks.

We therefore decompose verification into five levels: execution, validity, structure, semantics, and function. Each level depends on success at the preceding levels, so a circuit that fails at structure still receives credit for passing execution and validity. We pair this reward with a curriculum that stages tasks by the verification levels they stress, moving from code fundamentals to structural assembly, functional reasoning, and *de novo* design.

We make the following contributions:

- **SynBio-Reason**, the first protocol for language models on genetic circuit reasoning, with 4,753 circuits drawn from procedural generation, the Cello genetic design tool (Nielsen et al., 2016b), and canonical circuits from the synthetic biology literature. It spans nine tasks covering code repair, translation, functional reasoning, and *de novo* design, and every circuit is paired with executable pysbol3 code for automated evaluation.

[1]University of California, Berkeley, CA, USA. Correspondence to: Noah Flynn <noahflynn@berkeley.edu>.

*Proceedings of the $43^{rd}$ International Conference on Machine Learning*, Seoul, South Korea. PMLR 306, 2026. Copyright 2026 by the author(s).

- **GenCircuit-RL**, a training framework that combines hierarchical verification rewards with curriculum learning. We formalize five verification levels aligned with distinct failure modes and use a four-stage curriculum that shifts optimization pressure from code generation to task-specific correctness. Training uses Group Relative Policy Optimization (GRPO) to accommodate the changing reward landscape across curriculum stages.

- **Empirical analysis** showing that hierarchical rewards improve task success rate by 14 to 16 percentage points over binary rewards on functional reasoning tasks, that curriculum learning is required for strong design performance, that trained models reach 52.6% success on held-out repressor-promoter pairs, and that these trends replicate across model families including Llama-3.1-8B and Gemma-3-12B (Appendix E.15).

## 2. Setting & SynBio-Reason

SynBio-Reason provides a comprehensive testbed for evaluating language models on genetic circuit reasoning via code generation. It comprises procedurally generated circuits with guaranteed ground truth for training and in-distribution evaluation along with curated real-world circuits for out-of-distribution evaluation and canonical, gold-standard circuits from literature. Each circuit is paired with reference PySBOL code, enabling execution-based evaluation and code-level analysis.

### 2.1. Genetic Circuits

A genetic circuit consists of fundamental biological parts: promoters (P) that initiate transcription, ribosome binding sites (RBS) that control translation efficiency, coding sequences (CDS) that encode protein products, and terminators (T) that halt transcription. Parts are canonically assembled into expression cassettes following $P \rightarrow RBS \rightarrow CDS \rightarrow T$. Regulatory interactions between cassettes, mediated by repressor or activator proteins, enable implementation of programs that dictate desired functions. A repressor protein, when expressed from one cassette, can bind to and inhibit the promoter of another cassette, creating negative regulation. Conversely, an activator protein can enhance transcription from its target promoter, creating positive regulation.

We produce a dataset of six circuit types with known properties, enabling verifiable evaluation. Expression cassettes establish basic SBOL comprehension. Logic gates test functional reasoning via truth table verification. Feed-forward loops and toggle switches evaluate understanding of regulatory topology and feedback. Oscillators represent the most complex circuits, and cascaded circuits match Cello complexity with multi-layer Boolean functions. Appendix B.1

provides detailed descriptions.

### 2.2. Code Generation Action Space

SBOL provides a standardized representation for genetic designs. An SBOL document contains Component objects representing biological entities (DNA, RNA, protein, small molecule), SubComponent objects indicating relationships (e.g., an expression cassette Component contains SubComponents referencing its constituent promoter, RBS, CDS, and terminator), Constraint objects specifying topological ordering, and Interaction objects describing regulatory relationships. Each part is annotated with Sequence Ontology (SO) terms indicating biological role (a promoter Component has role SO:0000167) and Systems Biology Ontology (SBO) terms specifying interaction types (an inhibition Interaction with role SBO:0000169 indicates one Component represses another).

We formulate genetic circuit reasoning as code generation operating at the Component abstraction. Analogous findings in electronic design automation (Zhang et al., 2025) demonstrated that design at the functional subcircuit level (e.g., current mirrors, differential pairs) rather than individual transistors dramatically reduces search space complexity while aligning agent actions with human design intuition. Reasoning over individual base pairs is similarly intractable, but Components such as RBS and CDS constitute functional units that synthetic biologists use to reason about circuit logic and regulatory flow.

Generating domain-specific serialization formats (e.g., raw SBOL in RDF/XML formats), places substantial burden on models to learn low-level syntactic conventions orthogonal to scientific reasoning. Given a prompt $x$ specifying a task and context, the model generates Python code $\hat{c}$ that uses the pysbol3 library to programmatically construct the target SBOL document. This code is executed to produce a document $\hat{d} = \exec(\hat{c})$, which is evaluated against ground truth using a verification function $\mathcal{V}(\hat{d}, x) \in [0, 1]$.

This formulation leverages the abundance of Python in pre-training data, avoiding the sparsity of RDF serialization formats while mirroring practitioner workflows. Targeting the pysbol3 API offloads syntactic correctness (e.g., namespace management) to the library and yields interpretable, modifiable artifacts. Finally, execution tracebacks provide structured feedback signals that facilitate iterative self-correction.

### 2.3. Procedural Circuit Generation

We procedurally generate SBOL circuits and corresponding pysbol3 code, evaluating results on functional equivalence rather than exact code matching. Our library of 48 biological parts satisfies sufficient combinatorial diversity to generate

meaningful training data across all circuit types (including toggle switches and up to 5-node oscillators and 4-layer cascades), biological realism grounded in well-characterized components, and support for out-of-distribution evaluation using held-out parts. It contains promoters (constitutive and inducible), ribosome binding sites, coding sequences (reporters, repressors, activators), terminators, and operators from the iGEM Registry and Cello characterized collections. Ten orthogonal repressor-promoter pairs are organized into training (5 pairs: LacI, TetR, cI, PhlF, SrpR) and held-out (5 pairs: BM3R1, AmtR, QacR, BetI, AmeR) tiers for out-of-distribution evaluation. Complete specifications appear in Appendix C.1.1.

2.3.1. PROCEDURAL GENERATION PROTOCOL

Parts are split across training and held-out tiers, and each circuit type's procedural circuit generation algorithm (Appendix C.3.1) draws only from training-tier parts. We formalize circuit complexity through four parameters: regulatory depth $d$ (longest path from input to output), maximum fan-in $f$ (regulatory inputs per promoter), feedback indicator $b$ (presence of cycles), and node count $n$ (expression cassettes). Formal definitions and complexity class distributions appear in Appendix C.3.1. During generation, we sample circuits according to a distribution over complexity classes, as shown in Table 13 within Appendix C.3.2, to ensure balanced representation.

## 2.4. Real-World Circuits

To assess generalization, we leverage experimentally characterized circuits from the Cello genetic design tool for out-of-distribution evaluation with held-out parts, and a curated set of historically significant circuits from synthetic biology literature for design capability validation.

**Cello Circuit Corpus.** Beyond procedural circuits, we evaluate on 111 experimentally validated circuits from the Cello design tool (Jones et al., 2022) spanning E. coli and S. cerevisiae, partitioned into 71 in-distribution and 40 out-of-distribution circuits based on repressor systems used. Out-of-distribution circuits use held-out repressors (BM3R1, AmtR, QacR, BetI, AmeR), testing whether models apply the abstract regulatory principle "repressor X inhibits promoter pX" to novel systems. Appendix C.1.2 provides complete specifications.

**Literature-91 Gold Standard.** We curated a set of 91 circuits from seminal publications spanning 2000–2024, including the Gardner toggle switch and Elowitz repressilator (Elowitz & Leibler, 2000a), framed as "rediscovery" tasks against circuits designed by domain experts. The Literature-91 set comprises 50 original canonical circuits (expression cassettes, logic gates, toggle switches, oscil-

lators, FFLs) and 41 extended complex circuits (cascaded circuits, quorum sensing, light-responsive, Cello-designed, advanced oscillators). The complete list appears in Appendix G.1. For each circuit, we created a canonical SBOL3 representation capturing regulatory topology, corresponding pysbol3 code that constructs the circuit, a natural language description, and ground truth for circuit behavior.

**Perturbation-Based Augmentation.** Real-world circuits reflect practical constraints and design choices but are limited in quantity. To expand training data, we apply four perturbation operators to modify seed circuits. Iso-functional part substitution replaces a part with another of the same type and similar properties (e.g., substituting one orthogonal repressor-promoter pair for another), preserving circuit topology and function while altering quantitative behavior. Class-preserving substitution replaces a part with another of the same type but different properties that may alter function (e.g., replacing a constitutive promoter with an inducible promoter). Topology augmentation adds regulatory edges consistent with available parts (e.g., add redundant repression edge). Topology ablation removes regulatory edges, typically breaking circuit function (e.g., removing one repression edge from a toggle switch destroys bistability). From 91 Literature-91 seeds and 71 in-distribution Cello circuits, we generate 810 perturbed variants: 5 variants per seed circuit across all operators and chain lengths of 1–3 perturbations. The 40 OOD Cello circuits remain unperturbed. Ablated circuits are annotated with any functional defects, creating training examples for the circuit debugging task. Appendix C.7 provides detailed perturbation specifications.

## 2.5. Tasks

Nine tasks are organized into three groups: procedural training tasks (T1–T7), Cello evaluation tasks (T8–T9), and Literature-91 evaluation tasks for masked prediction and *de novo* design. Appendix C.4.1–C.4.3 provides full specifications.

Table 1 summarizes task composition. We enforce strict split separation throughout. Cello circuits are deduplicated against the procedural training set with part-aware graph isomorphism, and procedural train/validation/test splits are separated with the same criterion. Appendix C.6 details the split statistics and deduplication procedure.

## 3. GenCircuit-RL

We introduce GenCircuit-RL, a two-phase training framework that combines supervised fine-tuning (SFT) with RLVF. A circuit may fail for many reasons—syntax errors in code, invalid SBOL structure, incorrect ontology annotations, or flawed regulatory logic—and conflating these failure modes into a single binary signal discards information

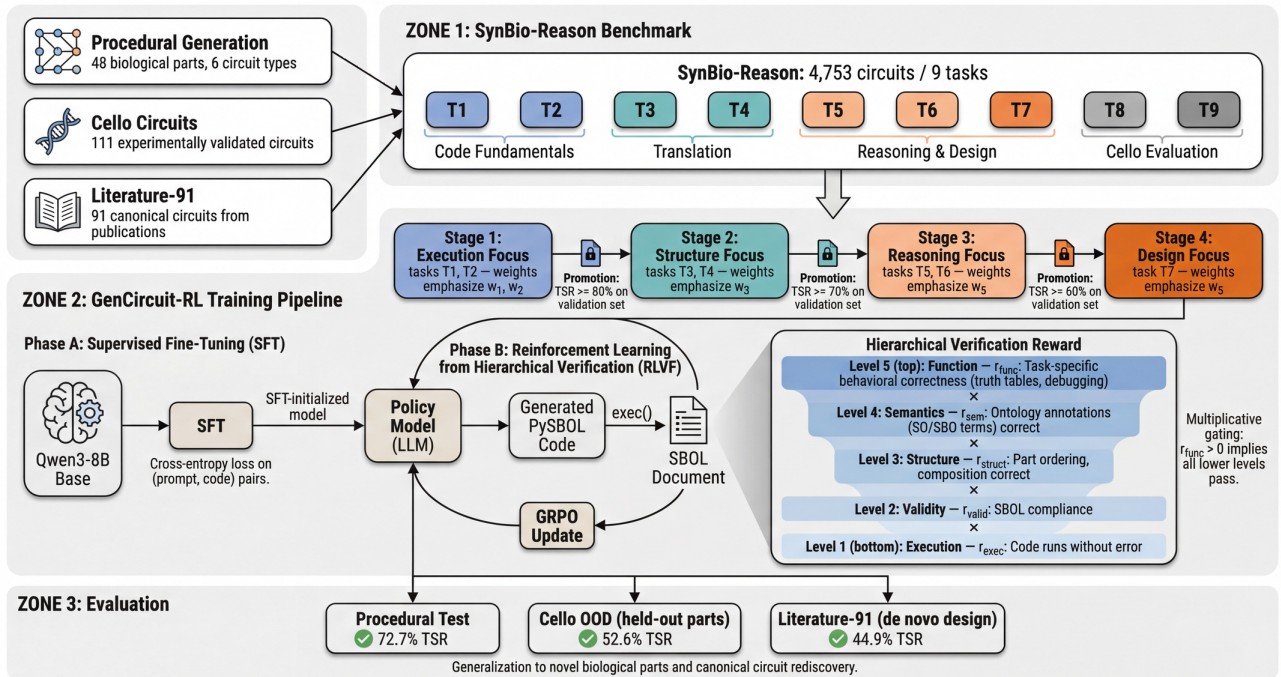

*Figure 1.* **GenCircuit-RL system overview.** *Zone 1:* We construct 4,753 circuits from three sources (procedural generation, Cello, and literature) spanning nine tasks. *Zone 2:* Training proceeds in two phases—SFT followed by GRPO-based RLVF. A four-stage curriculum shifts reward emphasis from execution (Stage 1) through structure (Stage 2) and reasoning (Stage 3) to de novo design (Stage 4), with promotion gated by validation-setbtask success rate. *Zone 3:* Evaluation on held-out procedural circuits, out-of-distribution Cello circuits with novel repressor–promoter pairs, and Literature-91 canonical circuits.

*Table 1.* Task instances are generated via procedures described in Appendix C.4.1.

| DATA SOURCE | CIRCUITS | TRAIN | VAL | TEST | TASK INSTANCES |
|---|---|---|---|---|---|
| PROCEDURAL | 1,247 | 998 | 124 | 125 | 18,700 |
| PROCEDURAL (PERT) | 2,494 | 1,995 | 250 | 249 | 37,400 |
| CELLO (IN-DIST) | 71 | – | – | 71 | 1,100 |
| CELLO (IN-DIST PERT) | 355 | – | – | 355 | 5,325 |
| CELLO (OOD) | 40 | – | – | 40 | 600 |
| LITERATURE-91 (SEED) | 91 | – | – | 91 | 1,365 |
| LITERATURE-91 (PERT) | 455 | – | – | 455 | 6,825 |
| **TOTAL** | **4,753** | **2,993** | **374** | **1,381** | **71,315** |

that could guide learning. GenCircuit-RL decomposes verification into a five-level hierarchy using curriculum learning to progressively shift optimization pressure from basic code generation toward task-specific correctness.

### 3.1. Supervised Fine-tuning

The SFT phase establishes competence with the pysbol3 API and SBOL document structure, providing initialization for RL refinement. We construct a training corpus of (prompt, canonical pysbol3 code) pairs from the procedural circuit dataset, where each prompt specifies a circuit design task. To encourage structured reasoning, we augment a subset of training examples with step-by-step construction traces. For each augmented example, we decompose the canonical code into logical construction stages—namespace initialization, component creation, constraint specification, interaction definition—with inline comments explaining the purpose of each stage.

We fine-tune using standard cross-entropy loss:

$$\mathcal{L}_{\text{SFT}}(\theta) = -\mathbb{E}_{(x,c^*)\sim\mathcal{D}} \left[ \sum_{t=1}^{|c^*|} \log p_\theta(c_t^* \mid x, c_{<t}^*) \right] \quad (1)$$

where $x$ is the prompt, $c^*$ is the target code, and $\theta$ are model parameters.

### 3.2. Hierarchical Verification Reward

A circuit that executes but has incorrect structure is closer to correct than one that fails to parse; this distinction should be reflected in the reward signal. We therefore compute rewards through a five-level verification hierarchy that mirrors established software engineering verification stages, progressing from syntax checking to behavioral testing (see Appendix E.4.4 for justification against prior work).

Given generated code $\hat{c}$ and task specification $s$, let $\hat{d} = \text{exec}(\hat{c})$ denote the SBOL document produced by success-

ful execution, or $\perp$ if execution fails.

$$r_{\text{exec}}(\hat{c}) = \mathbf{1}[\texttt{exec}(\hat{c}) \neq \perp] \tag{2}$$

$$r_{\text{valid}}(\hat{d}) = r_{\text{exec}} \cdot \mathbf{1}[\texttt{validate}(\hat{d})] \tag{3}$$

$$r_{\text{struct}}(\hat{d}, s) = r_{\text{valid}} \cdot \frac{1}{|C_{\text{struct}}|} \sum_c \mathbf{1}[c(\hat{d}, s)] \tag{4}$$

$$r_{\text{sem}}(\hat{d}, s) = r_{\text{valid}} \cdot \frac{1}{|C_{\text{sem}}|} \sum_c \mathbf{1}[c(\hat{d}, s)] \tag{5}$$

$$r_{\text{func}}(\hat{d}, s) = r_{\text{struct}} \cdot r_{\text{sem}} \cdot f_{\text{task}}(\hat{d}, s) \tag{6}$$

Execution verifies code runs; validity checks SBOL compliance; structure verifies part ordering and composition; semantics checks ontology annotations; function evaluates task-specific correctness criteria via topological analysis (truth tables via symbolic propagation for logic gates, motif detection for toggle switches and oscillators, flaw identification for debugging). Multiplicative dependencies prevent reward hacking: $r_{\text{func}} > 0$ implies all prerequisites pass. We note that our verification is structural/topological rather than dynamic: it confirms correct parts, regulatory connections, and motif presence, but does not simulate quantitative behavior (e.g., ODE-based confirmation of bistability or oscillation period). Topological correctness is a necessary prerequisite for functional behavior, and no prior work achieves even this level of verification for LM-generated genetic circuits. We discuss the path from topological to dynamic verification in Appendix G.5. Unlike binary verification where $\nabla R = 0$ for all failures, our hierarchical reward provides non-zero gradient whenever any level passes:

$$\frac{\partial R}{\partial \theta} = \sum_{\ell=1}^{5} w_\ell \frac{\partial r_\ell}{\partial \theta} \prod_{j<\ell} r_j \tag{7}$$

Even a circuit failing at level $\ell$ provides gradient signal for levels 1 through $\ell - 1$. Task-specific formulations appear in Appendix C.5.

### 3.3. Curriculum Learning

Seven training tasks span code repair to de novo design over four curriculum stages, each emphasizing different verification levels through adjusted reward weights. The weight schedule reflects the principle that prerequisite skills must be mastered before dependent skills can be learned: a model cannot construct correct circuit topology without first generating executable code, and cannot design topologically correct circuits without understanding valid structural configurations.

Table 2 specifies the curriculum configuration. Stage 1 emphasizes execution, training on code repair (T1) and completion (T2) tasks where the challenge is producing valid pysbol3 code. Stage 2 shifts focus to structural correctness, introducing part substitution (T3) and natural language translation (T4) tasks that require assembling parts in valid configurations. Stage 3 addresses functional reasoning through

*Table 2.* Curriculum stages with reward weights and promotion criteria. Weights $\mathbf{w} = (w_1, w_2, w_3, w_4, w_5)$ correspond to execution, validity, structure, semantics, and function levels respectively. TSR denotes task success rate on validation data.

| STAGE | FOCUS | WEIGHTS ($w_1$–$w_5$) | TASKS | PROMOTION |
|---|---|---|---|---|
| 1 | EXEC. | .40/.30/.20/.10/.00 | T1, T2 | TSR$\geq$80% |
| 2 | STRUCT. | .15/.15/.35/.25/.10 | T3, T4 | TSR$\geq$70% |
| 3 | REASON. | .10/.10/.20/.20/.40 | T5, T6 | TSR$\geq$60% |
| 4 | DESIGN | .05/.05/.15/.15/.60 | T7 | END |

logic prediction (T5) and circuit debugging (T6), requiring the model to understand the behavioral consequences of regulatory topology. Stage 4 targets de novo design (T7), requiring integration of all prior capabilities.

Advancement between stages occurs when the model achieves the specified task success rate threshold on validation data. Within each stage, training tasks are sampled according to a distribution that emphasizes stage-appropriate tasks while maintaining coverage of earlier skills to prevent catastrophic forgetting. Table 21 in Appendix D.2.3 provides the sampling distribution.

### 3.4. Policy Optimization

We apply Group Relative Policy Optimization (GRPO) (Shao et al., 2024), which estimates advantages through within-group normalization: $\hat{A}_{i,k} = (R_{i,k} - \mu_i)/(\sigma_i + \epsilon_0)$. This eliminates the need for a learned value function that would require retraining at curriculum transitions and handles the sparse distribution of correct outputs, which can result in early training instability when functional circuits are rare. We compare GRPO against PPO in Appendix E.6, finding that curriculum structure provides the dominant contribution while GRPO's advantage stems primarily from stability at curriculum transitions. The policy update uses the clipped surrogate objective with KL penalty ($\beta = 0.05$) to prevent drift from the SFT initialization. Hyperparameters appear in Appendix D.2.

## 4. Experiments

Experiments use the Qwen3-8B model (Yang et al., 2025) as the primary base architecture; to validate cross-architecture generality we additionally train Llama-3.1-8B (Grattafiori et al., 2024) and Gemma-3-12B (Gemma Team et al., 2025) under identical conditions (Appendix E.15). All experiments use instruction-tuned model variants

### 4.1. Baseline Methods

**Frontier models.** We evaluate Claude Opus 4.5 with extended thinking in zero-shot and 5-shot configurations, establishing a reference for what general-purpose mod-

els achieve without domain-specific training. Zero-shot prompts include the pysbol3 API documentation and task specification. For 5-shot prompts, we provide one example per circuit type, selected to demonstrate canonical construction patterns without overlapping with test circuits.

**Supervised fine-tuning (SFT).** We fine-tune the base model on (prompt, canonical code) pairs from the procedural training set using the cross-entropy loss. SFT provides the initialization for all RLVF experiments and serves as a baseline measuring RL's contribution beyond supervised learning.

**RLVF variants.** RLVF-Binary uses GRPO training with binary rewards where $R = 1$ if all five verification levels pass, and $R = 0$ otherwise, testing standard pass/fail verification suffices. RLVF-Hierarchical uses GRPO training with the five-level hierarchical reward but without curriculum staging. Tasks are sampled uniformly across task types throughout training, allowing the model to learn from balanced exposure to all difficulty levels simultaneously. RLVF-Hierarchical-Curriculum represents the full GenCircuit-RL method with hierarchical rewards and four-stage curriculum progression as specified in Table 2. To test sufficiency of parameter-efficient RL for reasoning, RLVF-LoRA represents the full curriculum method using Low-Rank Adaptation (LoRA). To assess scaling behavior, we also apply GenCircuit-RL to Qwen3-30B-A3B, Qwen3-4B, Qwen3-1.7B, and Qwen3-0.6B variants.

### 4.2. Evaluation Metrics

Task Success Rate (TSR) measures the fraction of evaluation instances achieving reward above a task-specific threshold:

$$\text{TSR} = \frac{1}{|\mathcal{E}|} \sum_{(x,s)\in\mathcal{E}} \mathbf{1}[R(\hat{c}, s) \geq \tau] \qquad (8)$$

where $\mathcal{E}$ is the evaluation set, $x$ is the prompt, $s$ is the task specification, and $\hat{c}$ is the generated code. We use $\tau = 0.9$ for tasks requiring complete correctness (T1 code repair, T2 code completion, T6 circuit debugging) and $\tau = 0.8$ for tasks with partial-credit structures (T3, T4, T5, T7, T8, T9).

Pass@$k$ characterizes the relationship between sampling and success probability, we report Pass@$k$ using the unbiased estimator (Chen et al., 2021):

$$\text{Pass@}k = \mathbb{E}_{\text{tasks}} \left[ 1 - \frac{\binom{n-c}{k}}{\binom{n}{k}} \right] \qquad (9)$$

where $n$ is the number of samples per task (we use $n = 20$) and $c$ is the number of samples with $R \geq \tau$. This metric distinguishes whether improvements reflect better ranking of solutions (high Pass@1 relative to Pass@$k$) or expanded coverage of the solution space.

*Table 3.* Task Success Rate (%) across methods and evaluation splits. Lit reports de novo design TSR on the full Literature-91 evaluation set (50 original + 41 extended circuits); see Table 54 for the original/extended breakdown and Table 55 for masked prediction results. Results show mean $\pm$ std over 5 seeds. Bold indicates best; $^\dagger$ indicates significant improvement over SFT ($p < 0.05$, Bonferroni-corrected). Cross-architecture validation with Llama-3.1-8B and Gemma-3-12B confirms consistent trends (Appendix E.15).

| METHOD | PROC. | C-ID | C-OOD | LIT | AVG |
|---|---|---|---|---|---|
| *Capability Reference* | | | | | |
| OPUS 4.5 (0) | 30.8 | 25.4 | 18.6 | 14.8 | 22.4 |
| OPUS 4.5 (5) | 41.8 | 36.5 | 28.4 | 21.7 | 32.1 |
| *Qwen3-8B* | | | | | |
| BASE (0-SHOT) | 10.4 | 8.2 | 5.1 | 4.4 | 7.0 |
| SFT | 53.9±2.8 | 46.2±2.3 | 32.4±2.7 | 27.3±2.5 | 40.0 |
| RLVF-BIN | 58.6±3.3 | 50.4±2.4 | 36.2±2.9 | 31.1±3.2 | 44.1 |
| RLVF-HIER | 67.7±3.0 | 58.3±3.3 | 44.1±3.1 | 37.8±2.8 | 52.0 |
| RLVF-H-C | **72.7**$^\dagger$±3.1 | **66.2**$^\dagger$±2.3 | **52.6**$^\dagger$±3.7 | **44.9**$^\dagger$±2.7 | **59.1** |
| *Llama-3.1-8B* | | | | | |
| SFT | 47.2±3.0 | 39.8±2.5 | 27.6±2.9 | 22.4±2.8 | 34.3 |
| RLVF-H-C | 64.1±3.3 | 57.4±2.6 | 45.2±3.5 | 37.1±3.0 | 51.0 |
| *Gemma-3-12B* | | | | | |
| SFT | 49.0±3.1 | 41.2±2.6 | 28.8±3.0 | 23.4±2.9 | 35.6 |
| RLVF-H-C | 66.2±3.3 | 59.0±2.7 | 46.8±3.5 | 39.0±3.1 | 52.8 |

To diagnose where models fail, we report per-level success rates for $\ell \in \{\text{exec}, \text{valid}, \text{struct}, \text{sem}, \text{func}\}$. To measure overfitting to procedural circuits, we compute $\Delta_{\text{gen}} = \text{TSR}_{\text{procedural}} - \text{TSR}_{\text{real-world}}$, where real-world TSR averages over Cello (in-distribution and OOD) and Literature-91 evaluation sets. Lower values indicate better generalization.

For TSR evaluation, we use greedy decoding (temperature $= 0$) to ensure deterministic outputs. For Pass@$k$ estimation, we use temperature $= 0.7$ with nucleus sampling (top-$p = 0.95$) to generate diverse samples. Maximum generation length is 8192 tokens, sufficient for the longest circuits. We report mean $\pm$ standard deviation computed over 5 independent training seeds with different random initializations. For pairwise method comparisons, we use a two-sided $t$-test and and consider $p < 0.05$ (post-correction) as statistically significant.

## 5. Results

### 5.1. Hierarchical Rewards Enable Genetic Circuit Reasoning

Table 3 shows that hierarchical verification improves task success across every evaluation split, and that curriculum learning adds a further gain on the most demanding tasks. Appendix E reports the full extended results.

Comparing RLVF-Hierarchical to RLVF-Binary isolates the benefit of the dense reward signal. On the Procedural-Test split, hierarchical rewards improve TSR by 9.1 percentage points (67.7% vs 58.6%). The improvement is concentrated on tasks involving regulatory reasoning: for T5 (logic prediction) and T6 (circuit debugging), hierarchical rewards

*Table 4.* Curriculum ablation: TSR (%) on functional tasks under different training regimes. "Direct (S4 weights)" uses Stage 4 task distribution (75% sampling weight on T5–T7) without curriculum transitions. This differs from RLVF-Hierarchical in Table 3, which uses uniform task sampling.

| TRAINING REGIME | T5 | T6 | T7 | T8 | AVG |
|---|---|---|---|---|---|
| DIRECT (S4 WEIGHTS) | 22.4 | 10.6 | 8.2 | 14.8 | 14.0 |
| REVERSE (S4→S1) | 32.6 | 19.4 | 15.2 | 22.6 | 22.5 |
| FULL (S1→S4) | 64.2 | 56.4 | 50.2 | 52.6 | 55.9 |

improve by 14.2 and 15.6 percentage points respectively, compared to 3.8 points for T1 (code repair). This pattern confirms that dense feedback is most beneficial when tasks require multi-step reasoning about circuit behavior rather than surface-level code generation. Curriculum learning (RLVF-H-C) provides further gains of 5.0 percentage points over non-curriculum hierarchical training. The curriculum effect is largest on complex tasks: T7 (de novo design) improves by 8.4 points, while T1 and T2 change little. This pattern is consistent with a staged training process that builds prerequisite capabilities before full design synthesis.

**Cross-architecture validation.** To assess whether these findings depend on the Qwen3 architecture, we replicate the full GenCircuit-RL pipeline on two models from different families: Llama-3.1-8B (Grattafiori et al., 2024) (8B parameters, Meta) and Gemma-3-12B (Gemma Team et al., 2025) (12B parameters, Google DeepMind). Table 3 shows that the key trends hold across all three families: RLVF-H-C improves over SFT by 16.7–17.2 pp on average, compared to 19.1 pp for Qwen3-8B. Curriculum learning remains essential for both models, with direct training collapsing to ~11–12% on T6–T7 versus 44–46% with the full curriculum (Appendix E.15). Absolute performance is lower for both non-Qwen3 models, consistent with their weaker Python code generation baselines on EvalPlus. Notably, Gemma-3-12B achieves similar TSR to Llama-3.1-8B despite having 50% more parameters, underscoring that pre-training data composition—particularly emphasis on code and STEM data—matters more than raw model capacity for this domain. Appendix E.15 extends this analysis to the 4B scale with a Gemma-3-4B vs. Qwen3-4B comparison.

## 5.2. Curriculum Learning Essential to Functional Tasks

Table 4 contrasts curriculum and direct training on functional reasoning tasks. Without curriculum staging, hierarchical rewards fail to achieve meaningful performance on T6 and T7.

Direct training with Stage 4 task distribution achieves only 10.6% TSR on T6 and 8.2% on T7, compared to 56.4% and 50.2% with the full curriculum. Inspection of model outputs reveals degenerate behavior: the model learns to produce

syntactically valid but trivial circuits (e.g., single-component expression cassettes regardless of task specification) that occasionally satisfy structural verification by chance. In contrast, RLVF-Hierarchical with uniform task sampling (Table 3) achieves higher performance (52.0% on T6, 41.8% on T7), demonstrating that balanced exposure to easier tasks prevents collapse even without explicit curriculum staging.

The curriculum provides gains beyond uniform sampling by ensuring that code generation capabilities are established before task-specific correctness becomes the dominant training objective. Reverse curriculum (S4→S1), which emphasizes function first, performs only marginally better than direct training, confirming that ordering—not just presence of staged training—is essential. Appendix E.3 provides detailed analysis of stage transition dynamics, Appendix E.5 provides detailed training curves showing transition dynamics and confirming that T6–T7 capabilities develop only after Stage 2 establishes structural foundations, and Appendix E.4 reports additional ablations on curriculum granularity, promotion thresholds, and reward weight sensitivity.

## 5.3. RLVF for Real-World Circuit Generalization

Table 5 evaluates transfer from procedural circuits to real-world designs. All methods perform worse on real-world circuits than on procedural circuits, reflecting the broader structural variation in the evaluation set. The relative generalization gap decreases from 34.5% for SFT to 24.9% for RLVF-H-C, and RLVF-H-C also achieves the smallest absolute gap at 18.1 percentage points. Training therefore improves real-world transfer in both absolute and relative terms.

*Table 5.* Generalization analysis: TSR (%) on procedural versus real-world circuits. Rel. Gap measures the absolute gap as a fraction of procedural performance. RLVF-H-C achieves both the smallest absolute gap and the lowest relative gap.

| Method | Procedural | Real-World | Abs. Gap ($\Delta_{gen}$) | Rel. Gap |
|---|---|---|---|---|
| SFT | 53.9 | 35.3 | 18.6 | 34.5% |
| RLVF-Binary | 58.6 | 39.2 | 19.4 | 33.0% |
| RLVF-Hier | 67.7 | 46.7 | 21.0 | 31.0% |
| RLVF-H-C | **72.7** | **54.6** | **18.1** | **24.9%** |

The OOD Cello evaluation circuits use 5 held-out repressor-promoter pairs that never appear during training. A model that has memorized specific part behaviors will fail; success requires applying abstract regulatory principles to novel repressor systems. While naming conventions provide a modest signal (~3pp; Appendix F.2), the naming convention ablation demonstrates that RLVF-H-C retains 73% of OOD performance even under adversarial naming conditions, confirming that the model acquires regulatory principles beyond surface-level pattern matching. On this split, RLVF-H-C achieves 52.6% TSR, compared to 32.4% for

SFT, demonstrating compositional generalization. Performance on the full Literature-91 set (44.9% for RLVF-H-C) is substantially lower than on the original 50 core circuits (60.8%; see Table 54 for the original/extended breakdown), reflecting the increased complexity and novel regulatory mechanisms (quorum sensing, optogenetics) present in the extended 41 circuits. Masked component prediction on Literature-91 circuits further confirms this pattern: RLVF-H-C achieves 74.6% function-level accuracy versus 52.4% for SFT, a 22.2pp improvement concentrated at the regulatory reasoning level rather than at structural slot-filling (Appendix F.4.3).

Beyond part-level generalization, Appendix F.1 analyzes topology-level and complexity generalization. Models retain 73.3% of in-distribution performance when half of parts are novel and exhibit degradation with circuit complexity, maintaining 67% retention on circuits exceeding training complexity by 4+ cassettes.

**Comparison with Cello.** Cello (Nielsen et al., 2016b; Jones et al., 2022), the most prominent existing tool for automated genetic circuit design, takes a Boolean truth table and a characterized parts library as input and uses simulated annealing to assign biological gates to a pre-synthesized NOR topology, whereas GenCircuit-RL accepts a natural language specification and generates complete SBOL code without access to quantitative part characterization data (Hill function parameters). On the gate assignment task (T8), which most closely matches Cello's technology mapping stage, Cello achieves **88.3%** topological correctness on the shared 111-circuit evaluation set using its own characterized response functions, while GenCircuit-RL achieves 52.6% without access to those parameters. Cello's advantage on this matched task is expected because its solver uses quantitative response function data unavailable to GenCircuit-RL. GenCircuit-RL addresses a broader design setting that includes novel topologies, open-vocabulary part specifications, and circuits outside Cello's library coverage. As complementary approaches, Cello excels at constrained gate optimization when characterized parts are available, while GenCircuit-RL enables generative design from high-level intent. Appendix F.3 provides a per-library breakdown.

### 5.4. Ablation Studies

**Reward component ablation.** Table 6 examines the contribution of individual verification levels to the reward signal. Removing $r_{exec}$ causes complete failure (0% TSR) because no outputs pass the execution prerequisite for higher-level verification. Removing $r_{struct}$ has the largest impact on functional tasks (T6–T7), reducing TSR by 14.9 percentage points, confirming that structural understanding is a prerequisite for functional reasoning.

*Table 6.* Reward component ablation: TSR (%) when individual reward levels are removed.

| Configuration | T1–T2 | T3–T5 | T6–T7 | Avg |
|---|---|---|---|---|
| Full hierarchical | 90.3 | 73.9 | 53.3 | 72.5 |
| Remove $r_{struct}$ | 87.4 | 60.2 | 38.4 | 62.0 |
| Remove $r_{sem}$ | 88.6 | 66.4 | 45.2 | 66.7 |
| Remove $r_{func}$ | 89.8 | 71.6 | 41.8 | 67.7 |

**Parameter-efficient training.** LoRA (rank 64) achieves 83.2% of full fine-tuning performance on code fundamentals (T1–T2) but only 59.8% on functional tasks (T6–T7). This disparity suggests that functional reasoning may require learning biological domain-specific representations spanning multiple layers, which the low-rank constraint cannot fully express.

### 5.5. Verification Level Analysis

We decompose distinct failure patterns across methods. For SFT, the largest drop occurs between structure and semantics (62.8% → 48.2%), indicating that models produce circuits with correct part composition but incorrect ontology annotations or regulatory relationships. RLVF-H-C shows more gradual degradation across levels, with the largest remaining gap between semantics and function (74.2% → 62.8%), indicating that the hierarchical reward addresses earlier failure modes. Qualitative analysis of failures reveal incomplete regulatory graphs (circuits contain correct parts but miss regulatory interactions) and part compatibility errors (models substitute biologically incompatible parts) as common problems, with a detailed error taxonomy provided in Appendix E.12.

These failures exhibit structured, circuit-type-specific patterns rather than random degradation. On oscillators (T6, Literature-91 O1–O8), the dominant failure is incomplete feedback topology: the model typically generates 2 of 3 required repression interactions in repressilator variants, suggesting it acquires the mutual-repression motif but struggles with three-node ring closure. Among Literature-91 rediscovery failures, 26.8% involve missing regulatory edges (Table 56). On cascaded circuits (T8–T9), failures concentrate at inter-module wiring: individual gates pass structural verification, but cascade connections (output of gate $i$ → input of gate $i+1$) are missing or mis-specified, consistent with the 17.6 pp degradation for cascades under leave-one-topology-out evaluation (Appendix F.1). Crucially, these are *diagnosable* failures amenable to practitioner repair: even among T6–T7 circuits that fail overall, 71% pass Levels 1–3 (executable, valid, structurally correct code), compared to only 34% for SFT. Appendix E.13 provides a detailed breakdown.

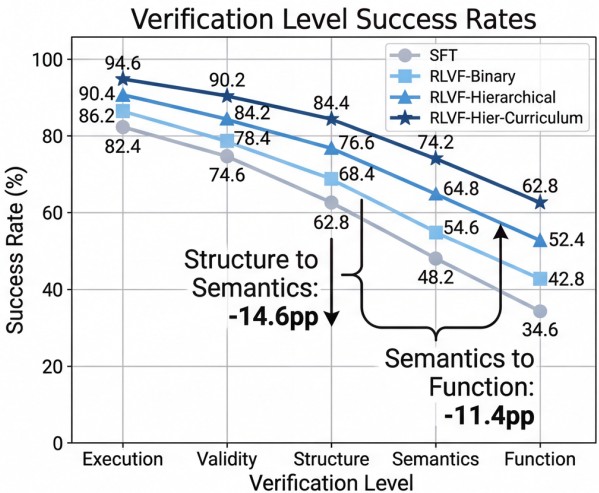

*Figure 2.* **Verification level success rates on the Procedural-Test split.** All methods show monotonically decreasing success from Execution to Function, but RLVF-Hier-Curriculum (RLVF-H-C) degrades most gracefully. The two largest drops—Structure→Semantics ($-14.6$ pp) and Semantics→Function ($-11.4$ pp)—identify ontology annotation and regulatory reasoning as the principal remaining bottlenecks. *Bottom:* the gain of RLVF-H-C over SFT compounds at higher verification levels ($+12.2$ pp at Execution, $+28.2$ pp at Function), confirming that hierarchical rewards and curriculum learning disproportionately benefit the reasoning capabilities the reward structure was designed to develop.

### 5.6. Pass@$k$ Analysis: RLVF Improves Ranking

RLVF-H-C improves Pass@1 by 18.4 percentage points but Pass@10 by only 8.2 points on functional tasks (T6, T7). The ratio of Pass@1 to Pass@10 increases from 0.42 (SFT) to 0.68 (RLVF), indicating that RLVF primarily improves the model's ability to rank correct solutions rather than expanding the space of solutions it can generate. Appendix E.10 provides complete Pass@$k$ breakdowns by task category and analysis of how solution ranking evolves during training.

### 5.7. Model Scale Analysis

To understand the capacity requirements of genetic circuit reasoning, we evaluate RLVF-H-C across five model scales from 0.6B to 30B parameters (Table 7). The resulting scaling pattern differs sharply between code-centric tasks and functional reasoning tasks.

Tasks T1–T2 show gradual improvement across all scales, from 46.4% at 0.6B to 93.7% at 30B. In contrast, tasks requiring biological reasoning (T6–T7) remain near floor for models $\leq 1.7$B parameters, then rise sharply to 53.3% at 8B. Above 8B, gains taper to $1.16\times$ from 8B to 30B. The result is not a sharp phase transition, but it does indicate

*Table 7.* Model scaling: TSR (%) by task category.

| MODEL | T1–T2 | T3–T5 | T6–T7 | AVG |
|---|---|---|---|---|
| QWEN3-0.6B | 46.4 | 18.5 | 5.2 | 23.4 |
| QWEN3-1.7B | 62.8 | 30.4 | 12.6 | 35.3 |
| QWEN3-4B | 75.4 | 45.7 | 27.0 | 49.4 |
| QWEN3-8B | 90.3 | 73.9 | 53.3 | 72.5 |
| QWEN3-30B | 93.7 | 79.3 | 61.6 | 78.2 |

that functional reasoning benefits disproportionately from additional capacity relative to syntax-heavy tasks.

Improvements from curriculum learning vary non-monotonically with model scale. The benefit peaks at 8B ($5.1\times$), where curriculum staging enables functional reasoning that direct training fails to achieve. Direct training at 8B achieves worse functional-task performance (10.5%) than at 4B (14.8%), despite better performance on code tasks. This reversal suggests that the larger model more readily collapses to solutions that satisfy lower verification levels without reaching topological correctness. Appendix E.8 provides full per-task scaling and curriculum-by-scale results.

## 6. Conclusion

We introduce GenCircuit-RL, a framework for training language models to reason about genetic circuit design through verifiable feedback.

The accompanying SynBio-Reason specification provides 4,753 circuits across six canonical types and nine task categories, including held-out repressor-promoter pairs for rigorous evaluation of generalization. Our results show that hierarchical verification improves functional reasoning and that curriculum staging is required for strong design performance. Models trained with this objective generalize to novel biological parts and rediscover canonical circuit topologies from the literature, which indicates that RLVF can instill transferable domain knowledge in this setting.

Our current verifier establishes topological correctness: it checks whether circuits contain the required parts, connections, and regulatory motifs. It does not yet verify quantitative dynamics. Extending the reward with kinetic simulation would move evaluation from topology to behavior and enable targets such as oscillation period, switching threshold, or response time (Appendix G.5). More broadly, the framework applies to scientific domains where correctness can be decomposed into executable intermediate criteria.

## Impact Statement

This work studies whether language models can assist with genetic circuit design, a task that remains labor-intensive and concentrated in a small number of expert groups. GenCircuit-RL shows that compact models can acquire ver-

ifiable design capabilities when training signal is tied to executable intermediate criteria.

Genetic circuits underlie engineered systems used in therapeutic delivery, biosensing, biomanufacturing, and environmental monitoring. GenCircuit-RL therefore contributes both domain-specific infrastructure and a training recipe that may transfer to other scientific problems with formal multi-stage verification.

Current genetic circuit design requires expertise in molecular biology, regulatory dynamics, and specialized software, which concentrates capability in well-resourced institutions. Tools based on standardized SBOL representations may broaden access and improve reproducibility, but they do not remove the need for expert review. For these reasons, data are made available through a request-based process consistent with responsible access practices for dual-use biological design tools and to prevent test set contamination of future models.

Our system reasons about circuit topology and regulatory logic, but it does not model quantitative dynamics, metabolic burden, or evolutionary stability, all of which matter for deployment. Appendix H reports preliminary experiments that use ML surrogates trained on CLASSIC data (Rai et al., 2025) as reward signals for quantitative refinement, but those results remain exploratory. Procedural generation also limits structural diversity. We partially address that limitation with Cello and Literature-91 evaluations on human-designed circuits, yet outputs should still be treated as design hypotheses that require experimental validation, especially in therapeutic or environmental settings.

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

*Table 8.* Comparison of genetic circuit design tools. Circuit type support indicates whether the tool can design each category. Osc = oscillators, TS = toggle switches, FFL = feed-forward loops, Casc = cascaded multi-gate circuits. Only GenCircuit-RL supports feedback circuit types and circuit debugging.

| TOOL | INPUT | GATES | CASC | TS | OSC | FFL | DEBUG |
|---|---|---|---|---|---|---|---|
| CELLO | VERILOG | ✓ | ✓ | – | – | – | – |
| CELLM | NL→VERILOG | ✓ | ✓ | – | – | – | – |
| EUGENE | RULES | CONSTRAINT SPEC. ONLY | | | | | – |
| GEN. CONST. | GUI | ASSEMBLY ONLY | | | | | – |
| GENCIRCUIT-RL | NL | ✓ | ✓ | ✓ | ✓ | ✓ | ✓ |

# A. Related Work

**Automated Genetic Circuit Design.** Early work in genetic design automation adapted electronic design automation ideas to biology. Cello established a workflow in which high-level logic specifications are compiled into DNA sequences, and tools such as Eugene (Bilitchenko et al., 2011) and Genetic Constructor (Bates et al., 2017) focused on constraint specification and assembly. CELLM (Abello Castillo & Gutierrez Pescarmona, 2025) adds a natural-language interface that translates prompts into Verilog for execution by Cello. GenCircuit-RL addresses a different objective: it trains a model to generate executable pysbol3 code directly, so the learned policy must represent circuit construction and regulatory logic rather than dispatch to an external solver. Table 8 summarizes the resulting capability differences, especially for feedback motifs such as oscillators, toggle switches, and feed-forward loops.

**RLVF for Scientific Domains.** Standard code generation methods rely on binary unit-test feedback (Le et al., 2022; Chen et al., 2022), which is too sparse for biological circuit design because a minor syntax error prevents downstream verification. We extend dense and hierarchical reward ideas from VeRPO (Weng et al., 2026) and RL-Struct (Hu & Wu, 2025) to a biological setting with five verification levels. Relative to RewardBridge (Su et al., 2025), our approach emphasizes sequential task decomposition; relative to SynLogic (Liu et al., 2025a), it uses synthetic data with guaranteed ground truth while validating on real circuit corpora. Our curriculum result also sharpens the picture from ProRL (Liu et al., 2025b): training duration matters, but the ordering of objectives matters as well. This formulation brings strict engineering constraints into the reward itself by checking execution, validity, compatibility, and function at progressively more demanding levels.

# B. Domain Background

## B.1. Fundamental Circuit Types

We produce a dataset of six circuit types selected to span fundamental synthetic biology motifs while enabling verifiable evaluation.

**Expression Cassettes.** The simplest circuit type consists of a single transcription unit: promoter, RBS, CDS, and terminator in canonical order. Variations include constitutive versus inducible promoters, weak/medium/strong RBS, and different reporter genes. Ground truth specifies part ordering, predicted expression level (derived from promoter and RBS strength), and inducibility. Expression cassettes serve as the foundation for curriculum learning, establishing basic SBOL comprehension before introducing regulatory complexity.

**Logic Gates.** Genetic implementations of Boolean functions including NOT, AND, OR, NOR, and NAND gates. Each gate comprises multiple cassettes with regulatory interactions implementing the logical function. For example, a NOT gate uses an inducible promoter driving a repressor that inhibits an output promoter (when the inducer is present, the output protein is repressed; when absent, the output is expressed). Two-input gates like AND, OR, NOR, and NAND require coordinated regulation from multiple pathways. Ground truth includes the gate type, input/output mapping, and complete truth table. Logic gates enable evaluation of functional reasoning without requiring dynamic simulation.

**Feed-Forward Loops.** Three-node regulatory motifs where gene A regulates both gene B and gene C, while B also regulates C. We implement coherent type 1 (C1-FFL, all activating edges) and incoherent type 1 (I1-FFL, mixed activation and repression). FFLs represent intermediate complexity between gates and memory circuits, testing understanding of regulatory path interactions. Ground truth specifies the FFL type, node identities, and expected dynamic behavior (delayed response for C1-FFL, pulse generation for I1-FFL).

**Toggle Switches.** Bistable circuits implementing biological memory through mutual repression. The canonical toggle switch comprises two repressors, each inhibiting expression of the other, creating two stable states that persist even after the inducing signal is removed. This genetic memory forms the basis of many cellular computation systems. Ground truth includes the bistability motif (mutual repression edges), stable state identities, and switching inputs. Toggle switches require understanding of feedback topology and its functional consequences.

**Oscillators.** Repressilator-style circuits that generate periodic gene expression through negative feedback loops of odd length (e.g., $n = 3$ or $n = 5$ genes). The canonical repressilator comprises three repressors arranged in a ring, where each represses the next, creating oscillations in protein levels over time. Ground truth specifies cycle length, feedback polarity, and whether oscillation is expected based on topology. Oscillators represent the most topologically complex circuits in the benchmark.

**Cascaded Circuits.** Multi-gate circuits that chain logic gates to implement Boolean functions requiring signal integration across multiple regulatory layers. These circuits, which match the complexity of designs produced by the Cello genetic circuit design tool (Nielsen et al., 2016b), include two-layer cascades (e.g., NAND implemented as AND followed by NOT), three-layer cascades (e.g., multiplexers, majority gates), and four-layer cascades representing the complexity frontier demonstrated by Cello. Ground truth includes the Boolean function as a truth table, the optimized gate topology, and predicted output states. Appendix C.3.2 provides detailed specifications.

## B.2. Applications of Genetic Circuits

### B.2.1. LIVING THERAPEUTICS

Engineered living cells offer advantages over traditional pharmaceuticals due to their capacity for environment-responsive behavior. A cell equipped with appropriate genetic circuits can sense its environment, distinguish between healthy and diseased states, and activate therapeutic programs accordingly, functioning as an autonomous decision-making agent (Kitada et al., 2018).

**Bacterial Cancer Therapy.** Certain bacteria naturally colonize solid tumors, making them attractive vehicles for targeted drug delivery (Zhou et al., 2018). The synchronized lysis circuit (SLC) exploits quorum sensing, a bacterial cell-to-cell communication mechanism, to coordinate therapeutic release (Din et al., 2016b). At low population density, cells grow normally; as density increases, signaling molecules accumulate until reaching a threshold that triggers coordinated cell lysis and therapeutic release. The surviving cells then regrow, creating periodic bursts of drug delivery.

**Engineered T-Cell Therapy.** Chimeric antigen receptor (CAR) T-cells are patient-derived immune cells modified to target cancer (Garfall et al., 2015). Synthetic Notch (synNotch) receptors enable sophisticated genetic circuits in these cells (Morsut et al., 2016). SynNotch receptors can be programmed to recognize specific cell-surface antigens and, upon binding, activate transcription of user-defined genes. Roybal et al. (Roybal et al., 2016) demonstrated AND-gate circuits where T-cells only activate when both of two tumor antigens are present, improving specificity and reducing off-target effects.

**miRNA-Based Classifiers.** Cancer cells exhibit characteristic microRNA (miRNA) expression profiles. Genetic classifier circuits can detect these signatures and trigger cell death selectively in cancer cells (Xie et al., 2011). Computational methods have been developed to design optimal classifier circuits from miRNA expression data, including approaches using two-step optimization (Mohammadi et al., 2017) and answer set programming (Becker et al., 2018).

### B.2.2. WHOLE-CELL BIOSENSORS

Whole-cell biosensors are engineered microorganisms that detect specific analytes and produce measurable outputs (fluorescence, luminescence, or electrical signals) (Gui et al., 2017). Compared to *in vitro* diagnostics, they offer low production costs, self-replication, and the ability to perform continuous monitoring.

**Gut Health Monitoring.** The probiotic strain *E. coli* Nissle has been engineered for various gastrointestinal applications. Mimee et al. (Mimee et al., 2018) developed an ingestible biosensor that detects gastrointestinal bleeding via heme sensing, packaged with miniaturized electronics for wireless data transmission. For gut inflammation detection, Woo et al. (Woo et al.,

2020) implemented an AND-gate circuit that simultaneously detects thiosulfate and nitrate biomarkers, improving diagnostic specificity through multi-input logic. Isabella et al. (Isabella et al., 2018) engineered bacteria to treat phenylketonuria (PKU), a metabolic disorder. The circuit activates phenylalanine-metabolizing enzymes specifically under the anoxic conditions of the mammalian gut, demonstrating how environmental sensing can control therapeutic function.

**Environmental Monitoring.** Heavy metal contamination is a widespread environmental concern. Genetic circuits for metal detection typically couple metal-responsive promoters to reporter genes, though achieving adequate signal-to-noise ratios remains challenging (Jung & Lee, 2019). Strategies include signal amplification cascades, toggle switches for digitized outputs, and alternative readout mechanisms. Din et al. (Din et al., 2020) developed an arsenic sensor using population-level dynamics: arsenic triggers cell lysis, causing a measurable drop in culture impedance and removing the need for optical detection equipment. Multiplexed biosensor arrays have been developed for detecting aromatic pollutants (phenols, benzene, toluene) at parts-per-billion concentrations (Roy et al., 2021), leveraging natural degradation pathways as sensing elements.

### B.2.3. BIOMANUFACTURING

Industrial biotechnology employs engineered microbial consortia (communities of multiple engineered strains) rather than single populations (McCarty & Ledesma-Amaro, 2019). Distributing complex metabolic pathways across multiple strains reduces the metabolic burden on individual cells and enables combinations of organisms with complementary capabilities.

**Distributed Biosynthesis.** Zhou et al. (Zhou et al., 2015) demonstrated distributed production of oxygenated taxanes (precursors to the cancer drug paclitaxel) using a two-species consortium of *E. coli* and *S. cerevisiae*. The system exploits cross-feeding: *E. coli* metabolizes xylose and excretes acetate, which *S. cerevisiae* consumes, creating a stable mutualistic interaction that maintains population balance without external control.

**Programmable Biomaterials.** Cell-to-cell communication enables spatial patterning of engineered materials. Chen et al. (Chen et al., 2014) created an *E. coli* consortium that produces patterned curli fiber biofilms under quorum sensing control. A "sender" population produces signaling molecules in response to an external inducer, while a "receiver" population detects these signals and produces the structural protein. This system generated environmentally switchable conductive biofilms and enabled assembly of quantum dot-decorated fibers.

## C. Benchmark Specification

### C.1. Training Parts Library and Coverage

#### C.1.1. PARTS LIBRARY COMPOSITION

See Table 9 for the complete parts library.

#### C.1.2. CELLO CIRCUIT CORPUS

Cello circuits represent high-quality evaluation data that have been experimentally validated and characterized with measured transfer functions, were designed by domain experts using principled engineering approaches, and span a range of complexity from simple NOT gates to multi-input NOR gate cascades. Cello 2.0 (Jones et al., 2022) includes five characterized libraries across three organisms. We select three libraries for our evaluation corpus based on coverage of architectural patterns, availability of characterization data, and alignment with our TetR-family repressor focus.

Table 10 summarizes the Cello circuit corpus. The Eco1C1G1T1 library contains the original 52 circuits designed for E. coli DH10$\beta$ using a tandem promoter architecture where multiple input promoters are arranged in sequence upstream of a single coding region. The Eco2C1G3T1 library contains 24 circuits designed for genomic integration in E. coli MG1655 using a split transcriptional unit architecture where each input promoter drives its own copy of the repressor gene. The SC1C1G1T1 library contains 35 circuits designed for S. cerevisiae BY4741, also using split transcriptional units, enabling evaluation of cross-organism generalization.

Each circuit was converted to SBOL3 format and paired with canonical pysbol3 construction code. Ground truth includes the circuit's Boolean function (derived from continuous response functions by thresholding at 50% of maximum output), the regulatory topology, and characterization data including Hill function parameters for each gate. We exclude the Bth1C1G1T1 library (B. thetaiotaomicron) because it uses CRISPRi-based regulation, which operates through a fundamentally different mechanism than the TetR-family repressors used in our other libraries.

To enable evaluation of compositional generalization, we partition the Cello circuits based on the repressor systems they use. A circuit is classified as in-distribution if all its gates use training-tier repressors (LacI, TetR, cI, PhlF, SrpR), even if the circuit topology differs from procedurally generated circuits. This tests generalization to novel topologies with familiar parts. A circuit is classified as out-of-distribution if it contains at least one gate using a held-out repressor (BM3R1, AmtR, QacR, BetI, AmeR).

**Cello UCF Coverage.** See Table 11 for parts library coverage of Cello User Constraints Files (UCFs).

**Eco1C1G1T1** is the original Cello library containing 12 TetR-family repressors characterized in E. coli DH10$\beta$ on plasmids. Gates use the tandem promoter architecture with an additive input model. This library was used to design and validate 60 circuits with up to 10 regulators and 55 parts (Nielsen et al., 2016b). We include 52 of these circuits as our primary E. coli evaluation set.

**Eco2C1G3T1** contains 6 repressors characterized for genomic integration in E. coli MG1655 using split transcriptional units. This library matches our procedural generation architecture and provides a direct comparison between training-distribution circuits (procedurally generated) and evaluation-distribution circuits (Cello-designed) using the same gate architecture. Our parts library provides 100% coverage of this UCF.

**SC1C1G1T1** contains 9 repressors characterized in S. cerevisiae BY4741 with split transcriptional units integrated at genomic loci. Including this library tests cross-organism generalization: the model must apply regulatory principles learned from E. coli circuits to yeast circuits that use different promoters, terminators, and cellular context. Our parts library provides 56% coverage (5 of 9 repressors).

We exclude **Bth1C1G1T1** (B. thetaiotaomicron) because it uses CRISPRi-based regulation through dCas9 and single guide RNAs, which operates through a fundamentally different mechanism than the TetR-family DNA-binding repression used in our other libraries. We also exclude **Eco1C2G2T2** to avoid redundancy with Eco1C1G1T1, as both use E. coli DH10$\beta$ with overlapping repressor sets.

*Table 9.* Complete Parts Library (48 parts). Parts are organized by functional category with associated properties and characterization tier. Training-tier parts are used during model training; held-out parts are reserved for out-of-distribution evaluation on Cello circuits.

| ID | Name | Type | Properties | Tier | Source |
|---|---|---|---|---|---|
| *Promoters (17)* | | | | | |
| BBa_J23100 | J23100 | promoter | constitutive, strong | training | iGEM |
| BBa_J23106 | J23106 | promoter | constitutive, medium | training | iGEM |
| BBa_J23117 | J23117 | promoter | constitutive, weak | training | iGEM |
| BBa_J23105 | J23105 | promoter | constitutive, med-weak | training | iGEM |
| BBa_J23114 | J23114 | promoter | constitutive, med-weak | training | iGEM |
| BBa_I0500 | pBad | promoter | inducible (arabinose) | training | iGEM |
| BBa_R0010 | pLac | promoter | repressible (LacI) | training | iGEM |
| BBa_R0040 | pTet | promoter | repressible (TetR) | training | iGEM |
| BBa_R0051 | pLambda | promoter | repressible (cI) | training | iGEM |
| BBa_K914003 | pRha | promoter | inducible (rhamnose) | training | iGEM |
| pPhlF | pPhlF | promoter | repressible (PhlF) | training | Cello |
| pSrpR | pSrpR | promoter | repressible (SrpR) | training | Cello |
| pBM3R1 | pBM3R1 | promoter | repressible (BM3R1) | held-out | Cello |
| pAmtR | pAmtR | promoter | repressible (AmtR) | held-out | Cello |
| pQacR | pQacR | promoter | repressible (QacR) | held-out | Cello |
| pBetI | pBetI | promoter | repressible (BetI) | held-out | Cello |
| pAmeR | pAmeR | promoter | repressible (AmeR) | held-out | Cello |
| *Ribosome Binding Sites (4)* | | | | | |
| BBa_B0030 | B0030 | rbs | strong | training | iGEM |
| BBa_B0032 | B0032 | rbs | medium | training | iGEM |
| BBa_B0033 | B0033 | rbs | weak | training | iGEM |
| BBa_B0034 | B0034 | rbs | very strong | training | iGEM |
| *Coding Sequences (14)* | | | | | |
| BBa_E0040 | GFP | cds | reporter (green) | training | iGEM |
| BBa_E1010 | mCherry | cds | reporter (red) | training | iGEM |
| BBa_I732005 | LacZ | cds | reporter (enzymatic) | training | iGEM |
| BBa_C0012 | LacI | cds | repressor ⊣ pLac | training | iGEM |
| BBa_C0040 | TetR | cds | repressor ⊣ pTet | training | iGEM |
| BBa_C0051 | cI | cds | repressor ⊣ pLambda | training | iGEM |
| PhlF | PhlF | cds | repressor ⊣ pPhlF | training | Cello |
| SrpR | SrpR | cds | repressor ⊣ pSrpR | training | Cello |
| BM3R1 | BM3R1 | cds | repressor ⊣ pBM3R1 | held-out | Cello |
| AmtR | AmtR | cds | repressor ⊣ pAmtR | held-out | Cello |
| QacR | QacR | cds | repressor ⊣ pQacR | held-out | Cello |
| BetI | BetI | cds | repressor ⊣ pBetI | held-out | Cello |
| AmeR | AmeR | cds | repressor ⊣ pAmeR | held-out | Cello |
| BBa_K206000 | AraC | cds | activator → pBad | training | iGEM |
| *Terminators (3)* | | | | | |
| BBa_B0010 | T1 | terminator | strong | training | iGEM |
| BBa_B0012 | TE | terminator | strong | training | iGEM |
| BBa_B0015 | DT | terminator | very strong (double) | training | iGEM |
| *Operators (10)* | | | | | |
| lacO | lacO | operator | cognate: LacI | training | — |
| tetO | tetO | operator | cognate: TetR | training | — |
| lambdaO | λO | operator | cognate: cI | training | — |
| phlO | phlO | operator | cognate: PhlF | training | Cello |
| srpO | srpO | operator | cognate: SrpR | training | Cello |
| bm3r1O | bm3r1O | operator | cognate: BM3R1 | held-out | Cello |
| amtO | amtO | operator | cognate: AmtR | held-out | Cello |
| qacO | qacO | operator | cognate: QacR | held-out | Cello |
| betO | betO | operator | cognate: BetI | held-out | Cello |
| ameO | ameO | operator | cognate: AmeR | held-out | Cello |

*Table 10.* Cello circuit corpus for out-of-distribution evaluation. Circuits are partitioned by whether they use only training-tier repressors (in-distribution) or include at least one held-out repressor (OOD).

| Library | Organism | Architecture | Total | In-Dist | OOD |
|---------|----------|--------------|-------|---------|-----|
| Eco1C1G1T1 | E. coli DH10$\beta$ | Tandem | 52 | 31 | 21 |
| Eco2C1G3T1 | E. coli MG1655 | Split | 24 | 18 | 6 |
| SC1C1G1T1 | S. cerevisiae BY4741 | Split | 35 | 22 | 13 |
| **Total** | | | **111** | **71** | **40** |

*Table 11.* Parts library coverage of Cello User Constraints Files (UCFs). Primary evaluation target is Eco2C1G3T1, which uses the same split transcriptional unit architecture as our procedurally generated circuits.

| UCF | Organism | Repressors | Coverage | Gate Model | Notes |
|-----|----------|------------|----------|------------|-------|
| Eco2C1G3T1 | E. coli MG1655 | 6 | 6/6 (100%) | Additive | Primary target; split architecture |
| Eco1C1G1T1 | E. coli DH10$\beta$ | 12 | 7/12 (58%) | Additive | Original Cello; tandem architecture |
| Eco1C2G2T2 | E. coli DH10$\beta$ | 9 | 6/9 (67%) | Non-additive | Roadblocking model; tandem |
| SC1C1G1T1 | S. cerevisiae | 9 | 5/9 (56%) | Additive | Yeast; split architecture |
| Bth1C1G1T1 | B. thetaiotaomicron | 7 | 0/7 (0%) | Additive | CRISPRi-based; different technology |

**Response Function Parameters.** Each gate in Cello libraries is equipped with a Hill function response mapping input transcriptional activity (in relative promoter units, RPU) to output activity:

$$y = y_{\min} + \frac{(y_{\max} - y_{\min}) \cdot K^n}{K^n + x^n} \tag{10}$$

where $y_{\min}$ is the minimum output (fully repressed), $y_{\max}$ is the maximum output (fully active), $K$ is the half-maximal input concentration, and $n$ is the Hill coefficient.

Hill function parameters from Cello characterization data for all ten repressor systems vary. These differences mean that gate assignment in cascaded circuits is non-trivial: the output dynamic range of an upstream gate must span the input threshold of the downstream gate for proper signal propagation. A model that has learned only the topology of circuits without understanding response function matching will fail on the gate assignment task (T8).

**Gate Architecture.** Genetic logic gates can be implemented using several architectural patterns that differ in how input signals are integrated and how transcriptional interference is managed.

Each gate in our library consists of two functional modules: input devices and output devices. An input device comprises an input promoter driving expression of the gate's regulatory protein through an RBS and coding sequence, terminated by a transcriptional terminator. For gates with multiple inputs, each input has its own complete input device producing the same regulatory protein from sequence-variant genes to avoid recombination. The output device consists of the gate's output promoter, which is regulated by the protein produced from the input devices. For repressor-based gates, the output promoter contains operator sequences recognized by the repressor protein; when repressor concentration is high, transcription from the output promoter is inhibited.

**Tandem versus Split Architectures.** In the tandem promoter architecture, multiple input promoters are arranged in sequence upstream of a single coding region. For a two-input NOR gate, both input promoters $P_1$ and $P_2$ are placed in tandem, each capable of driving transcription of the downstream repressor gene. This architecture is compact but introduces roadblocking effects: when RNA polymerase initiates from the upstream promoter, it can interfere with transcription initiation from the downstream promoter. This is addressed through a nonadditive input composition model that accounts for promoter interference, but the resulting transfer functions are more complex and position-dependent.

The split transcriptional unit architecture separates input promoters onto distinct gene copies. For a two-input NOR gate, input promoter $P_1$ drives one copy of the repressor gene while $P_2$ drives a second copy. Both gene copies produce the same repressor protein, which then inhibits the output promoter. This architecture eliminates roadblocking effects entirely,

allowing a simple additive input model where total repressor production equals the sum of contributions from each input. The cost is increased DNA length due to gene duplication.

We adopt the split architecture for procedural circuit generation because it simplifies the input composition model to pure addition:

$$x_{\text{total}} = \sum_{i=1}^{n} x_i \tag{11}$$

where $x_i$ is the transcriptional activity from input promoter $i$ in relative promoter units (RPU). This simplification enables straightforward topological verification: the output of a gate depends only on the sum of its inputs, not on their spatial arrangement. The split architecture also aligns with recent Cello libraries (Eco2C1G3T1, SC1C1G1T1) designed for genomic integration, where modular gene placement is essential.

---

**Algorithm 1** GenerateExpressionCassette

---

**Require:** Parts library $\mathcal{L}$, random seed $s$, constraints $\mathcal{C}$
**Ensure:** CircuitSpec containing SBOL document, pysbol3 code, and ground truth
    Initialize random state with seed $s$
    **Part Selection:**
    **if** $\mathcal{C}$ specifies promoter type **then**
        $P \leftarrow$ sample from $\mathcal{L}$.promoters matching $\mathcal{C}$
    **else**
        $P \leftarrow$ sample from $\mathcal{L}$.promoters (70% constitutive, 30% inducible)
    **end if**
    $R \leftarrow$ sample from $\mathcal{L}$.rbs (uniform over 4 training-tier RBS)
    $G \leftarrow$ sample from $\mathcal{L}$.cds.reporters (uniform over 3 reporters)
    $T \leftarrow$ sample from $\mathcal{L}$.terminators (50% B0015, 25% each other)

    **SBOL Construction:**
    Create Document $D$ with namespace `https://synbio-bench.org/`
    Create Component $C_{\text{cassette}}$ with role SO:0000804 (engineered_region)
    Create SubComponents $S_P, S_R, S_G, S_T$ referencing $P, R, G, T$
    Add SubComponents to $C_{\text{cassette}}$
    Add Constraint: $S_P$ `precedes` $S_R$
    Add Constraint: $S_R$ `precedes` $S_G$
    Add Constraint: $S_G$ `precedes` $S_T$
    {Constraints use SBOL3 restriction vocabulary (`sbol:precedes`)}

    **Ground Truth Computation:**
    expr_level $\leftarrow P$.strength $\times R$.strength
    inducible $\leftarrow (P$.regulation $=$ "inducible")
    repressible $\leftarrow (P$.regulation $=$ "repressible")

    **Code Generation:**
    Generate pysbol3 code that reconstructs $D$
    Generate natural language description
    **return** CircuitSpec($D$, code, ground_truth, description)

---

## C.2. Circuit Generation Algorithms

This appendix provides detailed algorithms for procedural generation of several circuit types in the benchmark. All algorithms use the 48-part library, drawing only from training-tier parts during generation. The algorithms produce both SBOL documents and canonical pysbol3 construction code.

### C.2.1. EXPRESSION CASSETTE GENERATION

Algorithm 1 describes the generation procedure for single expression cassettes. The algorithm samples parts according to specified constraints or defaults, then constructs the SBOL representation with appropriate ontology annotations.

### C.2.2. LOGIC GATE GENERATION

Logic gates are constructed by composing expression cassettes with regulatory interactions. Algorithm 2 describes NOT gate generation; two-input gates follow analogous structure with additional input cassettes.

### C.2.3. TWO-INPUT GATE GENERATION

Two-input gates are implemented using layered repressor architectures following established genetic logic design principles. NOR gates serve as the universal primitive: both inputs drive repressors targeting a common output promoter. AND, OR,

---

**Algorithm 2** GenerateNOTGate

---

**Require:** Parts library $\mathcal{L}$, random seed $s$
**Ensure:** CircuitSpec for NOT gate
  Initialize random state with seed $s$

  **Topology Selection:**
  Select repressor $Rep$ from {LacI, TetR, cI, PhlF, SrpR} uniformly
  Determine cognate promoter $P_{\text{rep}}$ for $Rep$

  **Input Cassette Construction:**
  $P_{\text{in}} \leftarrow$ sample constitutive or inducible promoter (excluding $P_{\text{rep}}$)
  $R_{\text{in}} \leftarrow$ sample RBS
  $G_{\text{in}} \leftarrow Rep$ (the repressor CDS)
  $T_{\text{in}} \leftarrow$ sample terminator
  Construct input cassette Component $C_{\text{in}}$

  **Output Cassette Construction:**
  $P_{\text{out}} \leftarrow P_{\text{rep}}$ (the repressible promoter)
  $R_{\text{out}} \leftarrow$ sample RBS
  $G_{\text{out}} \leftarrow$ sample reporter CDS
  $T_{\text{out}} \leftarrow$ sample terminator
  Construct output cassette Component $C_{\text{out}}$

  **Regulatory Interaction:**
  Create Interaction $I$ with type SBO:0000169 (inhibition)
  Add Participation: $Rep$ as inhibitor (SBO:0000020)
  Add Participation: $P_{\text{out}}$ as inhibited (SBO:0000642)

  **Ground Truth:**
  truth_table $\leftarrow \{(0, 1), (1, 0)\}$ {Input OFF $\rightarrow$ Output ON, etc.}
  gate_type $\leftarrow$ "NOT"
  **return** CircuitSpec with topology, code, and truth table

---

and NAND gates are constructed by composing NOR and NOT operations.

### C.2.4. TOGGLE SWITCH GENERATION

Toggle switches require mutual repression between two cassettes. Algorithm 4 ensures proper wiring of the bistable motif.

### C.2.5. BRANCHED ACTIVATION MOTIF GENERATION

Branched activation motifs feature a single regulator driving multiple downstream targets in parallel without regulatory interaction between targets. This topology is useful for coordinated gene expression but lacks the temporal filtering properties of true feed-forward loops.

### C.2.6. FEED-FORWARD LOOP GENERATION

True feed-forward loops (FFLs) comprise three nodes where node A regulates both B and C, and B additionally regulates C, creating parallel direct and indirect paths to the output. Coherent type-1 FFLs (C1-FFLs) use consistent regulatory signs (all activation or all repression) and function as sign-sensitive delays and noise filters.

C.2.7. OSCILLATOR GENERATION

Repressilator-style oscillators are generated as rings of $n$ repression interactions. The algorithm verifies that cycle length is odd (required for sustained oscillation in the deterministic limit).

## C.3. Cascaded Circuits

### C.3.1. CIRCUIT COMPLEXITY DEFINITIONS

To enable systematic analysis of model performance across the difficulty spectrum, we formalize circuit complexity through four structural parameters. Given a circuit $C = (V, E, \tau, \rho)$, we define the complexity tuple $\kappa(C) = (d, f, b, n)$ where each component captures a distinct aspect of regulatory structure: regulatory depth ($d$), maximum fan-in ($f$), feedback indication ($b$), and expression cassette count ($n$).

**Regulatory depth** $d$ measures the longest directed path in the regulatory graph from any input to any output. Formally, let $G_R = (V_R, E_R)$ be the subgraph containing only regulatory edges where $V_R \subseteq V$ contains cassettes and $E_R \subseteq E$ contains activation or repression relationships. Then $d = \max_{u,v \in V_R} \text{dist}(u, v)$ where $\text{dist}(u, v)$ is the length of the longest path from $u$ to $v$. Expression cassettes have $d = 0$; NOT gates have $d = 1$; cascaded gates have $d \geq 2$.

**Maximum fan-in** $f$ measures the greatest number of regulatory inputs to any single promoter. For each promoter $p \in V$ with $\tau(p) = \text{promoter}$, let $\text{in}(p) = |\{(u, p) \in E : \rho((u, p)) \in \{\text{activation}, \text{repression}\}\}|$. Then $f = \max_p \text{in}(p)$. Single-input gates have $f = 1$; AND and OR gates have $f = 2$; complex logic may have $f \geq 3$.

**Feedback indicator** $b$ is a binary variable indicating whether the regulatory graph contains any directed cycle. Formally, $b = 1$ if $G_R$ contains a cycle and $b = 0$ otherwise. Feed-forward circuits have $b = 0$; toggle switches and oscillators have $b = 1$. When finer granularity is needed, we extend this to a feedback count $b' = $ number of independent cycles.

**Node count** $n$ is simply the number of expression cassettes, that is, $n = |\{v \in V : v \text{ is a cassette}\}|$.

### C.3.2. EXTENSION TO CASCADED CIRCUITS

Cascaded circuits chain multiple logic gates to implement Boolean functions that cannot be realized with a single gate. The Cello design tool demonstrated that circuits containing up to 10 regulatory proteins and 55 genetic parts can function reliably in E. coli when gates are properly insulated. We include cascaded circuits at three depth levels to span this complexity range.

**Two-layer cascades** connect two gates in series, where the output promoter of the first gate serves as an input to the second gate. The simplest example is a double inversion (NOT followed by NOT), which implements a buffer. More useful two-layer circuits include NAND gates implemented as AND followed by NOT, and IMPLIES gates combining NOR with signal routing. Two-layer cascades require 2 regulatory proteins and 10 to 14 parts depending on the fan-in of each gate.

**Three-layer cascades** enable implementation of functions requiring signal integration across multiple paths. The multiplexer circuit, which selects between two data inputs based on a selector signal, requires three layers in NOR-only logic. Priority encoders and majority gates also fall into this category. Three-layer cascades typically require 4 to 6 regulatory proteins and 20 to 35 parts.

**Four-layer cascades** represent the complexity frontier demonstrated by Cello. The 0x3D circuit (XOR of three inputs) and 0x8E circuit contain 7 to 8 gates arranged in four layers, requiring 40 to 46 parts. At this depth, signal propagation delays become observable: transient incorrect output states (faults) can occur when inputs change, as signals propagate through the circuit at different rates. Four-layer cascades require 6 to 10 regulatory proteins and 35 to 55 parts.

Table 12 extends our complexity classification to include cascaded circuits.

### C.3.3. CASCADED CIRCUIT GENERATION

Generating cascaded circuits requires specifying a target Boolean function and synthesizing a gate network that implements it. We adopt the approach used by Cello: starting from a truth table specification, we apply logic synthesis to produce a network of NOR gates (which are functionally complete), then map biological gates to each node.

For procedural generation, we sample Boolean functions of 2, 3, or 4 variables uniformly at random from the space of all such functions. We exclude degenerate functions (constant 0, constant 1, or functions depending on fewer variables than specified). Each function is converted to a NOR-only network using the Yosys logic synthesis tool with optimization passes that minimize gate count. The resulting network defines the circuit topology; biological gates are then assigned to each node.

The gate assignment problem is NP-complete because different gates have different response functions, and the output of one gate must span the input threshold of the next gate for proper signal propagation. We implement a Monte Carlo simulated annealing algorithm following Nielsen et al. to find gate assignments that maximize the predicted ON/OFF ratio

*Table 12.* Extended circuit complexity classification including cascaded multi-gate circuits. Layers indicates the longest path from any input to the output. Parts count includes sensor cassettes (8 parts for 3 sensors) and output reporter (2 parts).

| Circuit Type | $d$ | $f$ | $b$ | $n$ | Parts | Examples |
|---|---|---|---|---|---|---|
| *Single-gate circuits* | | | | | | |
| Expression cassette | 0 | 0 | 0 | 1 | 4 | Constitutive reporter |
| NOT gate | 1 | 1 | 0 | 2 | 15 | Inverter |
| 2-input NOR/AND/OR | 1 | 2 | 0 | 3 | 19–21 | Basic 2-input logic |
| 3-input NOR | 1 | 3 | 0 | 4 | 23–25 | 3-input logic |
| *Feedback circuits* | | | | | | |
| Toggle switch | 2 | 1 | 1 | 2 | 18 | Bistable memory |
| 3-node oscillator | 3 | 1 | 1 | 3 | 22 | Repressilator |
| 5-node oscillator | 5 | 1 | 1 | 5 | 30 | Extended oscillator |
| *Feed-forward circuits* | | | | | | |
| C1-FFL | 2 | 2 | 0 | 3 | 22 | Coherent FFL |
| I1-FFL | 2 | 2 | 0 | 3 | 22 | Incoherent FFL |
| *Cascaded circuits* | | | | | | |
| 2-layer cascade | 2 | $\leq 2$ | 0 | 3–4 | 20–28 | NAND, IMPLIES, buffer |
| 3-layer cascade | 3 | $\leq 2$ | 0 | 4–6 | 28–38 | Multiplexer, majority |
| 4-layer cascade | 4 | $\leq 2$ | 0 | 6–10 | 38–55 | 0x3D, 0x8E, Consensus |

*Table 13.* Circuit complexity classes defined by the tuple $\kappa = (d, f, b, n)$. Classes progress from minimal complexity (single cassettes) through high complexity (multi-node oscillators with feedback). For high complexity circuits, we define a "Cascaded" class for deep feed-forward logic ($d \geq 3, b = 0$) and a "Feedback" class for cyclic graphs ($b = 1$)

| CLASS | $d$ | $f$ | $b$ | $n$ | TGT | EXAMPLES |
|---|---|---|---|---|---|---|
| *Tier 1: Basic* | | | | | | |
| MINIMAL | 0 | 0 | 0 | 1 | 10% | REPORTER |
| SIMPLE | 1 | 1 | 0 | 2 | 15% | NOT, BUFFER |
| *Tier 2: Logic* | | | | | | |
| MODERATE | $\leq 2$ | $\leq 2$ | 0 | 3–4 | 30% | NOR, FFLs |
| CASCADED | 3–4 | $\leq 3$ | 0 | 4–10 | 20% | MUX, 0x3D |
| *Tier 3: Dynamic* | | | | | | |
| FEEDBACK | $\geq 2$ | $\leq 2$ | 1 | $\geq 2$ | 25% | TOGGLE, OSC. |

across all output states. The algorithm iteratively swaps gate assignments and accepts improvements deterministically while accepting degradations with probability proportional to a temperature parameter that decreases over iterations.

Ground truth for cascaded circuits includes the Boolean function (as a truth table), the optimized gate topology, and the predicted output state for each input combination. Functional verification checks that the topology implements the specified truth table when signal propagation is computed through the gate response functions.

### C.4. Task Specifications

We organize benchmark tasks into three groups aligned with their role in the paper: procedural training tasks, Cello evaluation tasks, and Literature-91 evaluation tasks. Table 14 summarizes task applicability across circuit types and data sources.

**Tier 1: Procedural Training Tasks (T1–T7).** These tasks use procedurally generated circuits for model training and in-distribution evaluation. They are organized into four difficulty levels forming a curriculum: code fundamentals (T1 code repair, T2 code completion), translation (T3 part substitution, T4 natural language to code), functional reasoning (T5 logic prediction, T6 circuit debugging), and design (T7 de novo design). Each task provides a prompt and expects executable pysbol3 code as output, evaluated by executing the code and verifying the resulting SBOL document. Appendix C.4.1 provides detailed task specifications.

**Tier 2: Cello Evaluation Tasks (T8–T9).** These tasks specifically target the cascaded multi-gate circuits characteristic of Cello designs, testing capabilities that emerge only at higher circuit complexity. Task T8 (gate assignment optimization) provides a Boolean function specification and fixed gate topology, requiring the model to assign biological gates from the library such that the circuit produces correct output states. Task T9 (cascaded circuit debugging) extends T6 to multi-layer circuits where errors propagate through multiple gates, requiring signal flow analysis to localize faults. Appendix C.4.2 provides detailed specifications.

**Tier 3: Literature-91 Evaluation Tasks.** These tasks evaluate design capability on canonical circuits. *Masked component prediction* presents a circuit with one component replaced by a mask token, requiring prediction at three granularity levels: the specific part (part-level), the part type (type-level), or the functional role (function-level). *De novo design evaluation* assesses whether generated circuits match reference topologies using graph isomorphism, allowing different part choices while requiring identical regulatory structure. Appendix C.4.3 provides detailed evaluation methodology.

*Table 14.* Task applicability across circuit types and data sources. ✓indicates the task applies; – indicates not applicable. Logic prediction (T5) applies to circuits with deterministic steady-state input-output mappings; for feed-forward loops, T5 evaluates steady-state rather than transient behavior. Toggle switches and oscillators are excluded from T5 due to their bistable or oscillatory dynamics. Cello tasks (T8–T9) apply only to cascaded circuits. Literature-91 tasks apply to all circuit types in that corpus.

| Data Source | Task | Circuit Type | | | | | |
|---|---|---|---|---|---|---|---|
| | | Cassette | Gate | FFL | Toggle | Oscillator | Cascade |
| Procedural | T1: Code repair | ✓ | ✓ | ✓ | ✓ | ✓ | ✓ |
| | T2: Code completion | ✓ | ✓ | ✓ | ✓ | ✓ | ✓ |
| | T3: Part substitution | ✓ | ✓ | ✓ | ✓ | ✓ | ✓ |
| | T4: NL to code | ✓ | ✓ | ✓ | ✓ | ✓ | ✓ |
| | T5: Logic prediction | – | ✓ | ✓ | – | – | ✓ |
| | T6: Circuit debugging | ✓ | ✓ | ✓ | ✓ | ✓ | ✓ |
| | T7: De novo design | ✓ | ✓ | ✓ | ✓ | ✓ | ✓ |
| Cello | T8: Gate assignment | – | – | – | – | – | ✓ |
| | T9: Cascade debugging | – | – | – | – | – | ✓ |
| Literature-91 | Masked prediction | ✓ | ✓ | ✓ | ✓ | ✓ | – |
| | De novo (graph iso) | ✓ | ✓ | ✓ | ✓ | ✓ | – |

### C.4.1. TRAINING TASKS

We define seven task types organized into four curriculum tiers. Each task provides a prompt $x$ and expects executable pysbol3 code $\hat{c}$ as output. Evaluation executes $\hat{c}$ to obtain document $\hat{d}$ and then applies task-specific verification. The summaries below emphasize the distinction between tasks; the corresponding prompt templates appear later in Appendix G.4.

**Tier 1: Code Fundamentals**

**Task T1 (Code Repair).** T1 presents pysbol3 code with errors that prevent execution or cause SBOL validation failure. The model receives the failing code together with the corresponding traceback or validation report and must return a corrected program. Verification checks only whether the repaired code executes and produces a valid SBOL document. We generate these instances from Level 1–2 flaw injections in Table 15; Level 3–4 biological flaws are reserved for T6.

**Task T2 (Code Completion).** T2 presents partial pysbol3 code with missing regions marked by comments. Missing spans range from single statements to full blocks such as ordering constraints or interaction definitions. The model must complete the program so that it integrates with the preserved context and passes execution and validation.

**Tier 2: Translation**

**Task T3 (Part Substitution).** T3 provides working pysbol3 code and an instruction to make a local change, such as swapping an RBS strength or replacing one repressor-promoter pair with another. Success requires implementing exactly the requested modification while preserving the remaining structure.

**Task T4 (Natural Language to Code).** T4 provides a natural-language circuit specification and requires complete pysbol3 code for the described design. The task tests translation from an informal description to a formally valid implementation.

## Tier 3: Functional Reasoning

**Task T5 (Logic Prediction).** T5 provides pysbol3 code for a circuit and a set of input conditions, and asks the model to predict the output state. It is a functional inference task rather than a code-generation task. We apply it to gates, feed-forward loops, and cascaded circuits. Toggle switches and oscillators are excluded because their behavior is not determined by a single steady-state input-output mapping. For feed-forward loops, evaluation uses sustained-input steady state rather than transient pulse behavior.

**Task T6 (Circuit Debugging).** T6 provides code that executes successfully but yields the wrong biological behavior, together with a symptom phrased as an experimental observation. The model must infer the underlying design flaw from the regulatory topology and return corrected code. Unlike T1, the failure is silent at the code level and appears only in circuit behavior. We generate T6 instances from Level 1–4 biological flaw injections in Table 15, ranging from simple assembly defects to subtle feedback errors that preserve execution while breaking function.

## Tier 4: Design

**Task T7 (De Novo Design).** T7 provides a functional specification and access to the parts library, and requires complete pysbol3 code for a circuit that satisfies the requested behavior. This is the most difficult training task because it requires topology selection, compatible-part assignment, and correct SBOL construction in a single generation pass.

*Table 15.* Flaw injection specifications. Level indicates relative difficulty from trivial (1) to subtle (4). Levels 1–2 cause execution failures or SBOL validation errors and are used for T1 (Code Repair). Levels 3–4 produce syntactically valid code with biological/functional defects and are used exclusively for T6 (Circuit Debugging).

| Flaw Type | Level | Procedure | Symptom (T6 only) |
|---|---|---|---|
| missing_terminator | 1 | Remove terminator SubComponent | Transcriptional read-through |
| duplicate_component | 1 | Duplicate random SubComponent | Validation/parsing error |
| empty_feature | 1 | Add Feature with no data | Validation failure |
| wrong_part_order | 2 | Swap adjacent SubComponents | Circuit malfunction |
| missing_constraint | 2 | Delete one Constraint | Ambiguous assembly order |
| orphan_component | 2 | Remove references to part | Disconnected component |
| wrong_orientation | 2 | Flip orientation property | Circuit malfunction |
| mismatched_pair | 3 | Replace repressor with non-cognate | No repression observed |
| wrong_inducer | 3 | Change inducer in description | No response to inducer |
| inverted_logic | 3 | Swap activation/repression | Inverted output behavior |
| missing_interaction | 3 | Delete Interaction object | No regulation observed |
| incomplete_feedback | 4 | Remove one edge in feedback loop | No oscillation / no bistability |
| promoter_leak | 4 | Add constitutive baseline | Always-on expression |
| extra_regulation | 4 | Add spurious Interaction | Unexpected interference |

### C.4.2. CELLO EVALUATION TASKS

The introduction of cascaded circuits from Cello motivates two additional task types that test multi-gate reasoning capabilities.

**Task T8: Gate Assignment Optimization.** Given a Boolean function specification (truth table) and a fixed gate topology (NOR network), the model must assign biological gates from the library to each node such that the circuit produces correct output states. This task isolates the gate assignment problem from topology synthesis. The challenge arises because different gates have different response functions (Hill function parameters). The output of gate A must span the input threshold of

gate B for signal propagation to work correctly. A naive random assignment will likely fail; the model must reason about response function compatibility.

**Input:** Truth table specifying desired Boolean function; gate topology as a directed acyclic graph of NOR operations; gate library with Hill function parameters for each available gate.

**Output:** Assignment of biological gates to topology nodes, represented as a mapping from node identifiers to gate identifiers.

**Verification:** We simulate signal propagation using the assigned gates' response functions. For each input state in the truth table, we compute the predicted output by propagating RPU values through the network. The assignment is correct if all output states match the truth table within a tolerance threshold (ON states $> 0.5$ RPU, OFF states $< 0.1$ RPU).

This task directly mirrors Cello's technology mapping stage, enabling comparison between learned assignment strategies and Cello's simulated annealing approach.

**Task T9: Cascaded Circuit Debugging with Propagation Analysis.** Given pysbol3 code for a cascaded circuit that produces incorrect output in one or more states, along with the observed incorrect behavior, the model must identify which gate(s) are responsible and propose corrections.

This task is more challenging than T6 (single-gate debugging) because errors can propagate through multiple layers. An incorrect output might result from a faulty gate at layer 1, layer 2, or layer 3, and the model must trace signal flow to localize the fault.

**Input:** pysbol3 code implementing a cascaded circuit; truth table specification; description of observed incorrect output states (e.g., "output is ON for input state 010 when it should be OFF").

**Output:** Identification of faulty gate(s); explanation of how the fault propagates through the circuit to produce observed behavior; corrected pysbol3 code.

**Verification:** We execute the corrected code and verify that the resulting circuit passes all output state checks.

Ground truth for T9 is generated by introducing controlled faults into working cascaded circuits: swapping one gate for another with incompatible response function, removing one regulatory edge, or inverting one edge polarity.

### C.4.3. LITERATURE-91 EVALUATION TASKS

The Literature-91 corpus supports two evaluation tasks that assess design capability on canonical circuits.

**Masked Component Prediction.** To evaluate circuit understanding at a fine-grained level, we introduce a cloze-style task. Given a circuit with one component replaced by a mask token, the model must predict the masked component. We generate masks at three levels of granularity.

**Part-level masking** removes a specific part and requires prediction of that exact part. For example, given a circuit with the strong RBS B0030 masked, the model must predict "B0030" from the parts library. This tests memorization of common design patterns and part preferences.

**Type-level masking** removes a part and requires prediction of the part type. For example, given a masked position between a promoter and CDS, the model must predict that an RBS is required. This tests understanding of the canonical cassette structure and SBOL composition conventions.

**Function-level masking** removes a part and requires prediction of its functional role. For example, given a masked CDS in a toggle switch, the model must predict that a repressor is needed even if the specific repressor identity is ambiguous from context. This tests comprehension of regulatory logic and the relationship between circuit topology and component function.

Formally, let $C = (V, E, \tau, \rho)$ be a circuit and let $v^* \in V$ be the masked component. The task is to predict $v^*$ (part-level), $\tau(v^*)$ (type-level), or the functional role of $v^*$ derived from $E$ and $\rho$ (function-level). We evaluate using top-1 and top-5 accuracy, measuring whether the correct answer appears in the model's highest-ranked predictions.

**De Novo Design with Graph Isomorphism Evaluation.** Evaluating de novo design requires comparing generated circuits to reference topologies without requiring exact part matches. Two circuits implementing the same function may use different parts while sharing identical regulatory structure. We therefore evaluate topological correctness via labeled graph

isomorphism.

Let $G = (V, E, \ell)$ be the ground-truth regulatory graph where vertices $V$ represent expression cassettes, edges $E$ represent regulatory relationships, and the labeling function $\ell : E \to \{+, -\}$ assigns each edge a polarity indicating activation $(+)$ or repression $(-)$. Let $\hat{G} = (\hat{V}, \hat{E}, \hat{\ell})$ be the generated graph. The isomorphism score is defined as:

$$\text{Iso}(G, \hat{G}) = \mathbf{1}\left[\exists \phi : V \to \hat{V} \text{ bijective s.t. } (u, v) \in E \Leftrightarrow (\phi(u), \phi(v)) \in \hat{E} \text{ and } \ell(u, v) = \hat{\ell}(\phi(u), \phi(v))\right] \quad (12)$$

### C.5. Task-Specific Function Rewards

This appendix provides complete specifications for the function reward $r_{\text{func}}$ across all training tasks.

**Relationship to Hierarchical Rewards.** The definitions below specify the task-specific function reward component $\tilde{r}_{\text{func}}$ before hierarchical dependencies are applied. The final function reward incorporates prerequisite verification:

$$r_{\text{func}} = r_{\text{struct}} \cdot r_{\text{sem}} \cdot \tilde{r}_{\text{func}} \quad (13)$$

where $r_{\text{struct}}$ and $r_{\text{sem}}$ enforce that structural and semantic verification must pass before task-specific correctness contributes to the reward. The equations below define $\tilde{r}_{\text{func}}$ for each task type.

#### C.5.1. CODE REPAIR (T1)

Code repair tasks present pysbol3 code containing errors that prevent execution or cause validation failure. The function reward decomposes into error identification and repair quality:

$$\tilde{r}_{\text{func}}^{\text{T1}}(\hat{c}, s) = 0.3 \cdot \mathbf{1}[\text{error type correct}] + 0.3 \cdot \mathbf{1}[\text{error location correct}] + 0.4 \cdot \mathbf{1}[\text{repaired code executes and validates}] \quad (14)$$

Error types include API misuse (incorrect method signatures, missing arguments), reference errors (undefined variables, wrong variable names), and ontology errors (invalid SO/SBO terms). Location correctness requires identifying the specific line or code block containing the error.

#### C.5.2. CODE COMPLETION (T2)

Code completion tasks provide partial pysbol3 code with masked sections. The function reward measures whether the completed code produces the intended circuit:

$$\tilde{r}_{\text{func}}^{\text{T2}}(\hat{c}, s) = \mathbf{1}[\texttt{exec}(\hat{c}) \text{ produces document isomorphic to target}] \quad (15)$$

Isomorphism is evaluated using the graph matching procedure described in Appendix D.1.

#### C.5.3. PART SUBSTITUTION (T3)

Part substitution tasks require modifying a circuit by replacing specified parts. The function reward checks that exactly the specified modification was made:

$$\tilde{r}_{\text{func}}^{\text{T3}}(\hat{d}, s) = \mathbf{1}[\text{target part replaced}] \cdot \mathbf{1}[\text{replacement part correct}] \cdot \mathbf{1}[\text{no other changes}] \quad (16)$$

The "no other changes" criterion verifies that parts not mentioned in the substitution instruction remain identical to the original circuit.

#### C.5.4. NATURAL LANGUAGE TO CODE (T4)

Translation tasks provide natural language circuit descriptions and require complete pysbol3 implementations. The function reward is based on specification coverage:

$$\tilde{r}_{\text{func}}^{\text{T4}}(\hat{d}, s) = \frac{1}{|S|} \sum_{i \in S} \mathbf{1}[\text{specification element } i \text{ satisfied}] \quad (17)$$

Specification elements include required parts (by type and properties), required interactions (by type and participants), and any explicitly stated constraints.

### C.5.5. LOGIC PREDICTION (T5)

Logic prediction tasks provide circuit code and input conditions, requiring prediction of output state. The function reward is truth table accuracy:

$$\tilde{r}_{\text{func}}^{\text{T5}}(\hat{d}, s) = \frac{1}{|T|} \sum_{(i,o) \in T} \mathbf{1}[\texttt{eval}(\hat{d}, i) = o] \tag{18}$$

Symbolic evaluation proceeds by: (1) setting input promoter states based on inducer conditions, (2) propagating activation signals (conjunctive semantics for multiple activators), (3) applying repression (any active repressor blocks its target), and (4) reading output reporter state.

### C.5.6. CIRCUIT DEBUGGING (T6)

Debugging tasks present circuits with biological flaws and observed malfunction symptoms. The function reward requires both correct diagnosis and effective repair:

$$\tilde{r}_{\text{func}}^{\text{T6}}(\hat{c}, s) = 0.3 \cdot \mathbf{1}[\text{flaw type}] + 0.2 \cdot \mathbf{1}[\text{flaw location}] + 0.2 \cdot \mathbf{1}[\text{fix valid}] + 0.3 \cdot \mathbf{1}[\text{fix functional}] \tag{19}$$

Flaw types are categorized according to Table 15. "Fix valid" requires the repaired code to execute and produce a valid SBOL document. "Fix functional" requires the repaired circuit to pass the specific verification check that the original circuit failed.

### C.5.7. DE NOVO DESIGN (T7)

Design tasks provide functional specifications and require complete circuit implementations. The function reward combines topological correctness with specification adherence:

$$r\tilde{r}_{\text{func}}^{\text{T7}}(\hat{d}, s) = 0.4 \cdot \mathbf{1}[\text{topology correct}] + 0.3 \cdot r_{\text{struct}}^{\text{spec}} + 0.3 \cdot r_{\text{sem}}^{\text{spec}} \tag{20}$$

Topological correctness is evaluated via labeled graph isomorphism between the generated regulatory graph and the reference topology. $r_{\text{struct}}^{\text{spec}}$ and $r_{\text{sem}}^{\text{spec}}$ evaluate structure and semantic constraints derived from the design specification rather than from ground-truth code.

## C.6. Dataset Statistics and Deduplication

### C.6.1. SPLIT STATISTICS

Table 16 provides detailed statistics for each data source and split.

*Table 16.* Detailed dataset split statistics by circuit type and data source.

| Source | Circuit Type | Train | Val | Test | Total |
|---|---|---|---|---|---|
| | Expression cassettes | 240 | 30 | 30 | 300 |
| | Logic gates (NOT) | 120 | 15 | 15 | 150 |
| | Logic gates (2-input) | 320 | 40 | 40 | 400 |
| Procedural (seed) | Toggle switches | 80 | 10 | 10 | 100 |
| | Feed-forward loops | 56 | 7 | 7 | 70 |
| | Oscillators | 56 | 7 | 7 | 70 |
| | Cascaded circuits | 126 | 15 | 16 | 157 |
| | *Subtotal* | 998 | 124 | 125 | 1,247 |
| Procedural (perturbed) | All types | 1,995 | 250 | 249 | 2,494 |
| | In-distribution (seed) | – | – | 71 | 71 |
| Cello | In-distribution (perturbed) | – | – | 355 | 355 |
| | Out-of-distribution | – | – | 40 | 40 |
| Literature-91 | Seed circuits | – | – | 91 | 91 |
| | Perturbed variants | – | – | 455 | 455 |
| **Total** | | **2,993** | **374** | **1,166** | **4,753** |

C.6.2. DEDUPLICATION METHODOLOGY

**Procedural-to-procedural deduplication** uses the abstract topology matching described below to prevent data leakage. Any duplicates within a split are removed by retaining only the first occurrence.

Stage 1: Hash-based filtering. For computational efficiency, we first compute a canonical fingerprint from the sorted list of (part_type, part_role) tuples and the sorted list of (source_type, target_type, edge_polarity) tuples for regulatory edges. Circuits with matching fingerprints are candidate duplicates requiring further verification.

Stage 2: Graph isomorphism verification. For candidate duplicates and for all cross-split comparisons of procedural circuits, we extract regulatory graphs and verify non-isomorphism using the VF2 algorithm with attribute matching on node roles (cassette type) and edge polarities (activation/repression).

**Cello circuit deduplication** against the procedural training set uses part-aware graph isomorphism that considers specific part identities as node labels, not just functional roles. A Cello circuit using BM3R1-pBM3R1 is considered distinct from a procedural circuit using LacI-pLac even if both implement the same NOT gate topology, because the node labels differ. This preserves Cello circuits that test generalization to held-out parts while excluding exact duplicates.

## C.7. Perturbation Specifications

We define four perturbation operators that modify circuits while preserving varying degrees of functional equivalence.

**Iso-functional part substitution** replaces a part with another of the same type and similar properties. For example, a strong RBS may be replaced with a medium-strength RBS, or one repressor-promoter pair may be substituted for another orthogonal pair. These perturbations preserve circuit topology and function while altering quantitative behavior. They generate training examples for the part substitution task (T3) and test robustness to part choice variation.

**Class-preserving substitution** replaces a part with another of the same type but different properties that may alter function. For example, replacing a constitutive promoter with an inducible promoter changes the circuit from always-on to controllable. These perturbations preserve structural validity while potentially changing behavior, creating examples where the model must recognize functional consequences of part changes.

**Topology augmentation** adds regulatory edges consistent with available parts. For example, adding a redundant repression edge to a toggle switch creates a circuit with the same steady-state behavior but different dynamic properties. These perturbations test whether models can distinguish functionally equivalent topologies.

**Topology ablation** removes regulatory edges from a circuit. Unlike the other operators, ablation typically breaks circuit function: removing one repression edge from a toggle switch destroys bistability. We annotate ablated circuits with the specific functional defect introduced, creating training examples for the circuit debugging task (T6).

Given a seed circuit $C$, the perturbation pipeline proceeds as follows. First, a perturbation operator is selected according to a task-specific distribution. Second, the operator is applied to a randomly selected target within the circuit, subject to biological validity constraints: substitutions must respect part compatibility, and topology modifications must maintain a connected regulatory graph. Third, ground truth is updated to reflect the perturbation.

We generate perturbation chains of length one to three, where each step applies one operator to the result of the previous step. This creates a spectrum from near-seed circuits (one small change) to substantially modified designs (three compounding changes). The perturbation metadata, including operator sequence, targets, and functional annotations, is retained for all generated circuits.

# D. Verification and Training

## D.1. Verification Implementation Details

D.1.1. REGULATORY GRAPH EXTRACTION

We extract regulatory graphs from SBOL documents using the following procedure:

1. Parse the SBOL document using pysbol3.

2. Create graph nodes for each Component with role in {promoter, CDS, reporter}.

3. For each Interaction object in the document:

    (a) Identify participant roles from Participation objects (inhibitor/inhibited for repression, stimulator/stimulated for activation).

    (b) Create a directed edge from regulator (CDS producing the regulatory protein) to target (regulated promoter).

    (c) Label the edge with interaction polarity (repression or activation).

4. Validate graph connectivity and return the labeled directed graph.

### D.1.2. STRUCTURAL AND SEMANTIC CHECK ENUMERATION

Levels 3 and 4 of the hierarchical reward are computed as averages over fixed check sets $C_{\text{struct}}$ and $C_{\text{sem}}$, each containing five equally-weighted pass/fail checks ($|C_{\text{struct}}| = |C_{\text{sem}}| = 5$; each check contributes $1/5$ to the corresponding level score). The check set is identical for all task types T1–T7; only the expected component counts (c1, c2 in $C_{\text{struct}}$) vary by `circuit_type`, and checks c4–c5 in $C_{\text{sem}}$ are vacuously satisfied for expression cassettes that contain no interactions.

*Table 17.* Structural checks $C_{\text{struct}}$ (Level 3). Each check contributes $1/5$ to $r_{\text{struct}}$.

| ID | Description | Circuit Types | Implementation Note |
|---|---|---|---|
| c1 | Correct total count of `SO_ENGINEERED_REGION` components | All (expected count varies: 1, 3, or 4 cassettes depending on type) | Compares against `_EXPECTED_REGION_COUNT[circuit_type]` |
| c2 | Correct number of leaf cassettes, each containing exactly 4 `SubComponents` | All (expected leaf count varies by type) | Uses `_EXPECTED_LEAF_COUNT`; iterates over leaf SubComponents |
| c3 | Parts in each leaf cassette follow $P \to \text{RBS} \to \text{CDS} \to T$ ordering enforced by `SBOL_PRECEDES` constraint chain | All | Reconstructs total ordering from `Constraint` edges and checks canonical 4-step sequence |
| c4 | Each positional `SubComponent` references a `Component` carrying the correct Sequence Ontology role at that position | All | Checks SO role at each of the 4 canonical positions in the cassette |
| c5 | All DNA parts are drawn from the allowed parts library (identified by `Component.name`) | All | Skips protein and `SO_ENGINEERED_REGION` components; validates leaf parts against `PartsLibrary` |

### D.1.3. MOTIF DETECTION

Ground truth for toggle switches and oscillators requires detecting specific regulatory motifs.

**Bistability motif.** A circuit exhibits the bistability motif if its regulatory graph contains a cycle of length 2 where both edges are repression type. Formally, nodes $A$ and $B$ satisfy the bistability motif if edges $(A, B)$ and $(B, A)$ both exist with polarity "repression."

**Oscillator motif.** A circuit exhibits the oscillator motif if its regulatory graph contains a cycle where the number of repression edges is odd. For a cycle $v_1 \to v_2 \to \cdots \to v_n \to v_1$, let $n_{\text{rep}}$ be the count of repression edges. Oscillation is expected when $n_{\text{rep}} \mod 2 = 1$.

**Feed-forward loop motif.** A circuit contains a feed-forward loop if nodes $A$, $B$, $C$ exist such that edges $(A, B)$, $(B, C)$, and $(A, C)$ are all present. The FFL is coherent type-1 if all three edges are activation; incoherent type-1 if $(A, B)$ and $(A, C)$ are activation while $(B, C)$ is repression.

### D.1.4. GRAPH ISOMORPHISM FOR EVALUATION

De novo design evaluation requires comparing generated circuits to reference topologies without requiring exact part matches. We evaluate topological correctness via labeled graph isomorphism.

*Table 18.* Semantic checks $C_{\text{sem}}$ (Level 4). Each check contributes $1/5$ to $r_{\text{sem}}$.

| ID | Description | Circuit Types | Implementation Note |
|---|---|---|---|
| c1 | Every DNA `Component` carries at least one valid Sequence Ontology role | All | Validates against `_VALID_COMPONENT_ROLES` (5 SO terms: SO_PROMOTER, SO_RBS, SO_CDS, SO_TERMINATOR, SO_OPERATOR) |
| c2 | Every `Interaction.types` entry is a recognised SBO interaction term | All (vacuously true for expression cassettes with no Interactions) | Validates against `_VALID_INTERACTION_TYPES`: {SBO_INHIBITION, SBO_STIMULATION, SBO_GENETIC_PRODUCTION} |
| c3 | Every `Participation.roles` entry is a recognised SBO participation term | All (vacuously true for expression cassettes) | Validates against `_VALID_PARTICIPATION_ROLES` (6 SBO terms: inhibitor, inhibited, stimulator, stimulated, template, product) |
| c4 | Each inhibited promoter has at least one cognate repressor among its inhibitors | NOT / NOR gates; skipped (vacuously true) for expression cassettes | Wired-OR aware: non-cognate co-inhibitors are permitted provided at least one cognate inhibitor is present |
| c5 | Reporter CDS components do not appear as `TEMPLATE` in production interactions; repressor CDS components do not occupy the output cassette CDS position | NOT / NOR gates; vacuously true for expression cassettes | Checks both directions of the reporter/repressor role mismatch |

Let $G = (V, E, \ell)$ be the ground-truth regulatory graph where $V$ contains expression cassettes, $E$ contains regulatory edges, and $\ell : E \to \{+, -\}$ assigns polarity. Let $\hat{G} = (\hat{V}, \hat{E}, \hat{\ell})$ be the generated graph. The graphs are isomorphic if there exists a bijection $\phi : V \to \hat{V}$ such that $(u, v) \in E \Leftrightarrow (\phi(u), \phi(v)) \in \hat{E}$ and $\ell(u, v) = \hat{\ell}(\phi(u), \phi(v))$ for all edges.

We implement isomorphism checking using the VF2 algorithm with attribute matching on edge polarities.

### D.1.5. SIGNAL PROPAGATION VERIFICATION FOR CASCADED CIRCUITS

Cascaded circuits require verification that signals propagate correctly through all layers. Let $G = (V, E)$ be the gate topology where $V$ contains gate nodes and $E$ contains signal edges. For each gate $v \in V$, let $f_v : \mathbb{R}^+ \to \mathbb{R}^+$ be its response function mapping input RPU to output RPU.

Given input state $\mathbf{i}$ (a vector of sensor ON/OFF states), we compute the steady-state output by iterative propagation:

$$x_v^{(t+1)} = f_v \left( \sum_{u \in \text{parents}(v)} x_u^{(t)} \right) \tag{21}$$

where $x_v^{(0)} = 0$ for all non-input nodes. Iteration continues until convergence (change $< 10^{-6}$ RPU) or a maximum iteration count is reached.

The circuit passes signal propagation verification if, for all input states, the output node's steady-state value correctly distinguishes ON versus OFF according to the truth table.

### D.1.6. GATE COMPATIBILITY VERIFICATION

We verify that each gate connection is functionally compatible: the output dynamic range of the upstream gate must span the input threshold of the downstream gate. Formally, for edge $(u, v) \in E$:

$$[y_{\min,u}, y_{\max,u}] \cap [K_v/10, K_v \times 10] \neq \emptyset \tag{22}$$

where $K_v$ is the half-maximal concentration of gate $v$. This check identifies assignments where signal matching fails even though the topology is correct.

## D.2. Training Hyperparameters

### D.2.1. SUPERVISED FINE-TUNING

Table 19 lists best performing hyperparameters for the SFT phase.

*Table 19.* Supervised fine-tuning hyperparameters.

| Parameter | Value |
|---|---|
| Learning rate | $2 \times 10^{-5}$ |
| Batch size | 32 |
| Gradient accumulation steps | 4 |
| Epochs | 3 |
| Warmup ratio | 0.03 |
| Weight decay | 0.1 |
| Maximum sequence length | 8192 |
| Truncation strategy | Right truncation |
| Optimizer | AdamW |

### D.2.2. GRPO REINFORCEMENT LEARNING

Table 20 summarizes best performing hyperparameters for the RL phase. Learning rate $1 \times 10^{-6}$ provided stable training; higher rates caused instability after curriculum transitions. We empirically estimate KL divergence using the per-token log-ratio.

*Table 20.* GRPO reinforcement learning hyperparameters.

| Parameter | Value |
|---|---|
| Prompts per batch ($N$) | 64 |
| Completions per prompt ($K$) | 8 |
| Learning rate | $1 \times 10^{-6}$ |
| Clip ratio ($\epsilon$) | 0.2 |
| KL coefficient ($\beta$) | 0.05 |
| Advantage normalization $\epsilon_0$ | $10^{-8}$ |
| Maximum sequence length | 8192 |
| Evaluation frequency | 500 steps |
| Temperature (sampling) | 0.7 |
| Temperature (evaluation) | 0.0 |

### D.2.3. TASK SAMPLING DISTRIBUTION

Table 21 specifies the probability of sampling each task type within each curriculum stage.

*Table 21.* Task sampling probability by curriculum stage. Stage-appropriate tasks receive highest weight while earlier tasks maintain coverage to prevent forgetting.

| Task | Stage 1 | Stage 2 | Stage 3 | Stage 4 |
|---|---|---|---|---|
| T1: Code repair | 0.40 | 0.10 | 0.05 | 0.05 |
| T2: Code completion | 0.40 | 0.10 | 0.05 | 0.05 |
| T3: Part substitution | 0.10 | 0.30 | 0.10 | 0.05 |
| T4: NL to code | 0.10 | 0.30 | 0.15 | 0.10 |
| T5: Logic prediction | 0.00 | 0.10 | 0.30 | 0.15 |
| T6: Circuit debugging | 0.00 | 0.10 | 0.25 | 0.15 |
| T7: De novo design | 0.00 | 0.00 | 0.10 | 0.45 |

# E. Extended Experimental Results

This appendix provides additional experimental details, extended results tables, and supplementary analyses.

## E.1. Per-Task Performance Breakdown

Table 22 provides complete per-task TSR breakdowns for all methods on the Procedural-Test split.

*Table 22.* Per-task TSR (%) on Procedural-Test split. Best performance in bold.

| Method | T1 | T2 | T3 | T4 | T5 | T6 | T7 | Avg |
|---|---|---|---|---|---|---|---|---|
| Claude Opus 4.5 (0-shot) | 52.4 | 48.6 | 34.2 | 28.4 | 22.6 | 16.8 | 12.4 | 30.8 |
| Claude Opus 4.5 (5-shot) | 64.2 | 60.8 | 46.4 | 40.2 | 34.6 | 26.2 | 20.4 | 41.8 |
| Qwen3-8B (0-shot) | 22.4 | 18.6 | 10.2 | 8.4 | 6.2 | 4.4 | 2.8 | 10.4 |
| SFT | 78.4 | 76.2 | 62.4 | 58.6 | 44.8 | 32.6 | 24.2 | 53.9 |
| RLVF-Binary | 83.4 | 80.8 | 68.2 | 64.4 | 48.6 | 36.4 | 28.2 | 58.6 |
| RLVF-Hierarchical | 87.2 | 85.4 | 74.6 | 70.2 | 62.8 | 52.0 | 41.8 | 67.7 |
| RLVF-Hier-Curriculum | **91.2** | **89.4** | **80.6** | **76.8** | **64.2** | **56.4** | **50.2** | **72.7** |

## E.2. Performance by Circuit Type

Table 23 shows performance stratified by circuit type on T7 (de novo design), revealing how capabilities vary with topological complexity. Performance decreases with circuit complexity for both methods. Improvements are largest for logic gates (+30.0pp), where hierarchical rewards most effectively teach functional reasoning about truth tables and regulatory logic. Improvements are smaller for expression cassettes (+22.0pp), where SFT already achieves reasonable performance on structurally simple circuits, and for oscillators (+19.0pp), where the complexity of cyclic feedback topology limits the benefit of hierarchical verification alone. This non-uniform pattern is consistent with the hypothesis that hierarchical rewards primarily improve intermediate-difficulty reasoning rather than providing uniform gains.

*Table 23.* TSR (%) by circuit type on T7 (de novo design). Complexity increases left to right.

| Method | Cassette | Gate | FFL | Toggle | Oscillator |
|---|---|---|---|---|---|
| SFT | 48.6 | 32.4 | 22.8 | 18.4 | 12.6 |
| RLVF-Hier-Curriculum | 70.6 | 62.4 | 49.8 | 42.4 | 31.6 |
| Δ | +22.0 | +30.0 | +27.0 | +24.0 | +19.0 |

## E.3. Curriculum Stage Transition Analysis

Table 24 records the training steps at which each curriculum stage transition occurred.

*Table 24.* Curriculum stage transitions across 5 training seeds.

| Transition | Steps (mean $\pm$ std) | TSR at Promotion | Threshold |
|---|---|---|---|
| S1 → S2 | 2,850 $\pm$ 340 | 82.4% | 80% |
| S2 → S3 | 11,400 $\pm$ 1,280 | 72.6% | 70% |
| S3 → S4 | 24,600 $\pm$ 2,450 | 63.8% | 60% |

Stage 1 (code fundamentals) is mastered quickly. Stage 3 (functional reasoning) represents the most challenging transition, requiring the most additional training before achieving sufficient performance to advance.

## E.4. Curriculum Ablations

This section provides extended ablation studies investigating curriculum design choices: the number of stages, promotion thresholds, and reward weight sensitivity.

### E.4.1. Curriculum Granularity

Table 25 evaluates different curriculum granularities, from no curriculum through 2-stage and 3-stage variants to the full 4-stage curriculum.

*Table 25.* Curriculum granularity ablation: TSR (%) across all tasks. Configurations vary in how stages are grouped. "2-stage" merges S1+S2 (code) and S3+S4 (function); "3-stage" separates S1, merges S2+S3, and separates S4.

| Configuration | T1 | T2 | T3 | T4 | T5 | T6 | T7 | T6–T7 | Avg |
|---|---|---|---|---|---|---|---|---|---|
| No curriculum | 82.4 | 80.2 | 54.6 | 48.2 | 22.4 | 10.6 | 8.2 | 9.4 | 43.8 |
| 2-stage | 86.8 | 85.4 | 70.2 | 65.8 | 52.6 | 42.8 | 33.4 | 38.1 | 62.4 |
| 3-stage | 89.4 | 87.8 | 76.4 | 72.2 | 60.4 | 50.2 | 44.6 | 47.4 | 68.7 |
| 4-stage (full) | 91.2 | 89.4 | 80.6 | 76.8 | 64.2 | 56.4 | 50.2 | 53.3 | 72.7 |
| Reverse (4→1) | 78.6 | 75.4 | 48.2 | 42.4 | 32.6 | 19.4 | 15.2 | 17.3 | 44.5 |

The results demonstrate that curriculum granularity matters substantially for functional task performance. Moving from no curriculum to 2-stage yields the largest single improvement (+28.7 pp on T6–T7), establishing the importance of separating code-focused from function-focused training. The 3-stage and 4-stage configurations provide incremental gains (+9.3 pp and +5.9 pp respectively), with diminishing returns suggesting that the coarse separation between prerequisite and target capabilities matters most.

Unlike anti-curriculum approaches that expose models to hard examples early (i.e., the reverse curriculum), our results demonstrate that for hierarchical verification tasks, the forward curriculum is essential. The reverse curriculum performs worse than no curriculum on T1–T2, indicating that attempting functional tasks before establishing code competence interferes with learning basic API usage.

### E.4.2. Promotion Threshold Sensitivity

Table 26 examines sensitivity to the TSR thresholds that trigger curriculum stage advancement. Lower thresholds enable faster training but achieve lower final performance (48.2% vs 53.3%), suggesting that advancing before capabilities are consolidated leads to suboptimal learning. Higher thresholds marginally improve final performance (+1.5 pp) at the cost of 50% more training steps, yielding worse efficiency (TSR per training step). The current thresholds represent a reasonable trade-off. Fixed step-based advancement underperforms adaptive thresholds despite similar training budgets, confirming the value of performance-based curriculum progression that adapts to learning dynamics.

*Table 26.* Promotion threshold sensitivity. "Lower" advances at 70%/60%/50%; "Higher" at 90%/80%/70%; "Fixed" uses predetermined step counts (3K, 12K, 25K) rather than performance-based advancement.

| Threshold Config | S1→S2 | S2→S3 | S3→S4 | T6–T7 | Total Steps | TSR/Step |
|---|---|---|---|---|---|---|
| Lower (70/60/50) | 70% | 60% | 50% | 48.2±2.4 | 22,400 | 2.15 |
| Current (80/70/60) | 80% | 70% | 60% | 53.3±1.6 | 30,200 | 1.77 |
| Higher (90/80/70) | 90% | 80% | 70% | 54.8±1.8 | 45,600 | 1.20 |
| Fixed (step-based) | @3K | @12K | @25K | 50.6±2.2 | 40,000 | 1.27 |

### E.4.3. Reward Weight Sensitivity

Table 27 investigates sensitivity to the Stage 4 reward weights $\mathbf{w} = (w_{\text{exec}}, w_{\text{valid}}, w_{\text{struct}}, w_{\text{sem}}, w_{\text{func}})$.

*Table 27.* Reward weight sensitivity at Stage 4. Weights sum to 1.0 and determine the relative contribution of each verification level to the total reward.

| Configuration | $w_{\text{exec}}$ | $w_{\text{valid}}$ | $w_{\text{struct}}$ | $w_{\text{sem}}$ | $w_{\text{func}}$ | T6–T7 | $\Delta$ |
|---|---|---|---|---|---|---|---|
| Current | 0.05 | 0.05 | 0.15 | 0.15 | 0.60 | 53.3 | — |
| Uniform | 0.20 | 0.20 | 0.20 | 0.20 | 0.20 | 45.8 | −7.5 |
| Function-heavy | 0.05 | 0.05 | 0.10 | 0.10 | 0.70 | 52.1 | −1.2 |
| Function-only | 0.00 | 0.00 | 0.00 | 0.00 | 1.00 | 38.4 | −14.9 |
| Structure-heavy | 0.05 | 0.05 | 0.35 | 0.35 | 0.20 | 47.2 | −6.1 |

The current weight configuration performs best, but the method is not overly sensitive to moderate variations: increasing $w_{func}$ from 0.60 to 0.70 yields only a 1.2 pp decrease. However, extreme configurations cause substantial degradation. Function-only weights ($w_{func} = 1.0$) reduce performance by 14.9 pp, demonstrating that intermediate verification levels provide essential gradient signal. Uniform weights reduce performance by 7.5 pp, confirming that emphasizing task-specific correctness—the ultimate training objective—yields better results than treating all verification levels equally.

The structure-heavy configuration ($w_{struct} + w_{sem} = 0.70$) underperforms by 6.1 pp, indicating that while structural correctness enables task-specific correctness, over-emphasizing intermediate objectives can distract from the final goal.

### E.4.4. Principled Justification for Hierarchical Verification

Our five-level verification hierarchy aligns with prior work on structured verification for code generation. Hierarchical reward decomposition has proven effective in code generation. VeRPO (Weng et al., 2026) introduced weighted test case rewards, and RL-Struct (Hu & Wu, 2025) proposed hierarchical verification for structured data generation. Our approach extends these ideas to the biological domain, where the verification hierarchy has natural correspondence to the layers of biological abstraction: sequence → parts → devices → systems.

The granularity of five levels emerged from the structure of SBOL verification itself. Coarser decompositions (e.g., syntax vs semantics) lose the distinction between structural composition (SubComponents, Constraints) and semantic annotation (SO/SBO terms, Interactions), which represent distinct failure modes in our error analysis (Appendix E.12). Finer decompositions would increase complexity without corresponding gains, as our ablation on reward weights suggests moderate sensitivity to the exact decomposition.

## E.5. Training Dynamics and Stability Analysis

We analyze training dynamics to understand how curriculum staging interacts with policy optimization. Figure 4 shows TSR evolution across task categories, mean reward, and KL divergence over 30,200 training steps for GRPO with curriculum staging. Three key patterns are visible. First, T1–T2 performance (code fundamentals) converges rapidly during Stage 1, reaching 85% TSR within 2,800 steps, and remains stable through subsequent stages with only minor transient dips at transitions. Second, T6–T7 performance (functional reasoning) remains near zero throughout Stages 1 and 2 despite exposure to functional tasks at low sampling rates (Table 21), confirming that functional reasoning requires established structural capabilities as prerequisites. Third, each curriculum transition produces a transient reward dip (Table 28) as the policy adapts to shifted task distributions and reward weights, followed by recovery within 600–1,500 steps.

*Table 28.* Curriculum transition dynamics for GRPO. Reward dip is the maximum decrease in mean reward within 500 steps of the transition. Recovery is the number of steps until reward returns to pre-transition levels. KL spike is the maximum KL divergence observed within 500 steps of the transition. Values are means across 5 seeds.

| Transition | Reward Dip | Recovery (steps) | KL Spike |
|---|---|---|---|
| S1→S2 | −0.08 | 600 | 0.06 |
| S2→S3 | −0.12 | 1,500 | 0.10 |
| S3→S4 | −0.10 | 1,500 | 0.08 |

The S2→S3 transition produces the largest reward dip (−0.12) and KL spike (0.10), reflecting the substantial shift from structural tasks to functional reasoning. The S3→S4 transition is smoother despite the high $w_{func}$ weight, because Stage 3 has already established functional reasoning foundations. This asymmetry provides further evidence that the curriculum ordering (structure before function) is essential.

**Implications for compute allocation.** The rapid convergence of T1–T2 within Stage 1 (∼2,850 steps, 9.4% of total training) and the extended plateau on T6–T7 through Stages 1–2 together imply that over one-third of the training budget is spent on prerequisites that are necessary but individually insufficient for functional reasoning. The adaptive promotion thresholds ensure that compute shifts to harder objectives as soon as prerequisites are met, avoiding wasted steps on already-mastered capabilities. We quantify the full per-stage compute allocation in Appendix E.14.

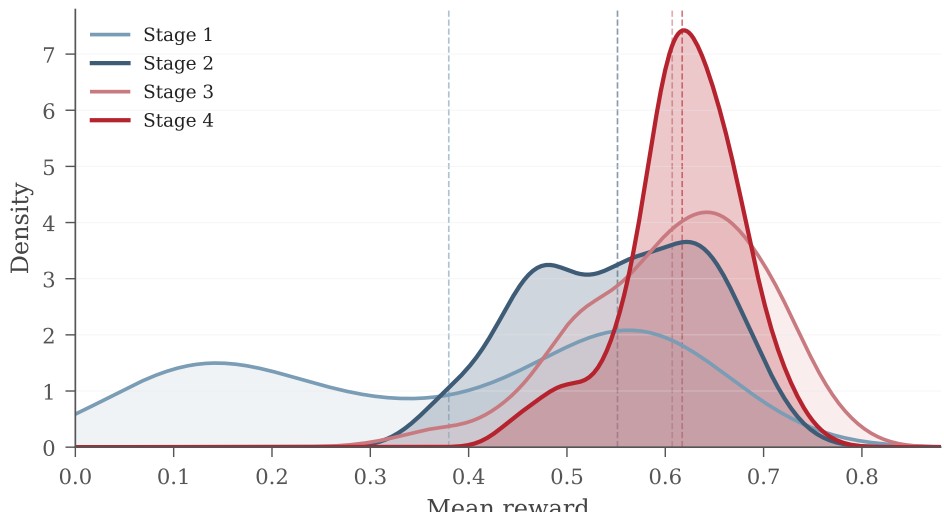

*Figure 3.* **Reward distribution evolution across curriculum stages.** Stage 1 (light blue) shows a bimodal distribution with substantial mass near zero, reflecting the mixture of executable and non-executable code. Successive stages shift the distribution rightward: Stage 2 (dark blue) eliminates most zero-reward outputs, Stage 3 (light red) concentrates mass above 0.5, and Stage 4 (dark red) produces a tight, high-reward distribution. Dashed vertical lines indicate stage means. The progressive concentration of reward mass demonstrates that each curriculum stage builds on the previous stage's competence.

## E.6. Policy Optimization Algorithm Comparison

To isolate the contribution of the optimization algorithm from the curriculum structure, we compare GRPO and PPO under identical curriculum configurations (Table 29). Curriculum learning provides the dominant contribution: it improves T6–T7 performance by +42.8pp for GRPO and +33.8pp for PPO, confirming that hierarchical curriculum staging benefits both algorithms. GRPO provides an additional +10.7pp over PPO when both use curriculum (53.3% vs 42.6%), primarily attributable to stability at curriculum transitions (Table 30). PPO exhibits approximately $2\times$ deeper reward dips and $2\times$ longer recovery at each transition, because its learned value function becomes stale when the reward distribution shifts. GRPO's within-group advantage normalization $\hat{A}_{i,k} = (R_{i,k} - \mu_i)/(\sigma_i + \epsilon_0)$ is computed per-prompt, making it naturally robust to non-stationary reward distributions induced by curriculum transitions. Without curriculum, the GRPO advantage shrinks to just 1.7pp (10.5% vs 8.8%), confirming that the contributions of curriculum structure and optimization algorithm are largely orthogonal.

*Table 29.* Policy optimization algorithm comparison. GRPO and PPO are evaluated with and without curriculum staging. Both use hierarchical rewards. Results show mean $\pm$ std over 5 seeds.

| Method | T1–T2 | T3–T5 | T6–T7 | Avg |
|---|---|---|---|---|
| GRPO + Curriculum | 90.3±2.8 | 73.9±2.0 | 53.3±2.3 | 72.5 |
| GRPO + No Curriculum | 82.4±2.8 | 54.6±3.5 | 10.5±3.4 | 49.2 |
| PPO + Curriculum | 86.8±3.9 | 66.4±3.1 | 42.6±3.4 | 65.3 |
| PPO + No Curriculum | 80.6±3.2 | 50.2±3.5 | 8.8±3.9 | 46.5 |

## E.7. Model Scale Comparison

## E.8. Detailed Scaling Analysis

Table 32 provides per-task performance across all model scales evaluated.

Table 33 highlights the superlinear scaling pattern by computing ratios between consecutive scales.

Table 34 shows that the curriculum benefit is non-monotonic with scale, peaking at 8B.

*Table 30.* Transition instability comparison between GRPO and PPO. PPO exhibits approximately $2\times$ deeper reward dips and $2\times$ longer recovery at each curriculum transition due to value function staleness.

| Transition | Algo | Reward Dip | Recovery (steps) | KL Spike |
|---|---|---|---|---|
| S1→S2 | GRPO | −0.08 | 600 | 0.06 |
| | PPO | −0.14 | 1,200 | 0.09 |
| S2→S3 | GRPO | −0.12 | 1,500 | 0.10 |
| | PPO | −0.22 | 3,200 | 0.18 |
| S3→S4 | GRPO | −0.10 | 1,500 | 0.08 |
| | PPO | −0.18 | 2,800 | 0.15 |

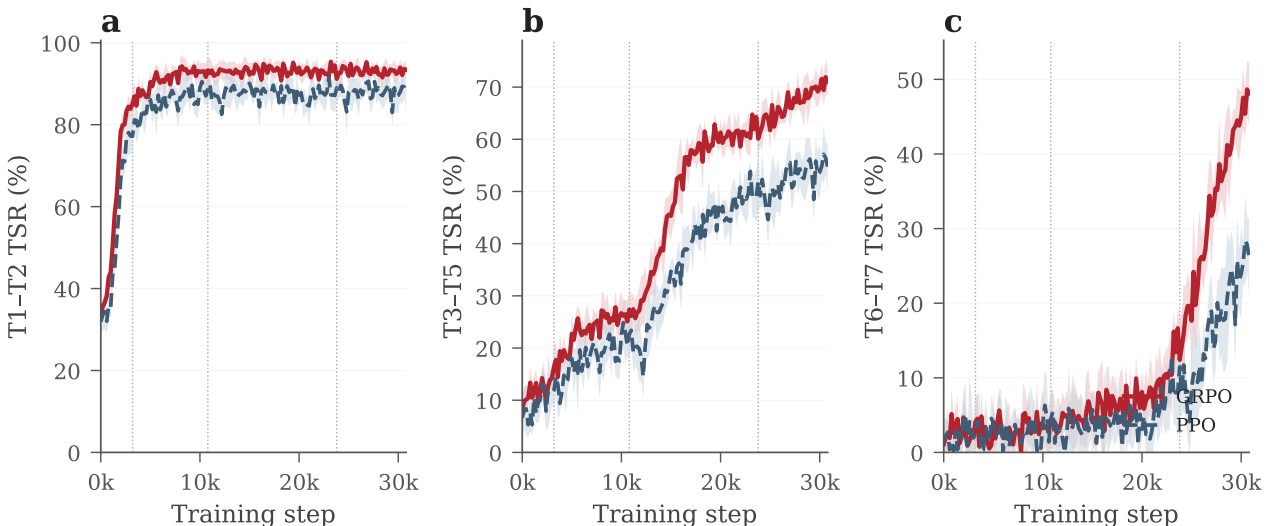

*Figure 4.* **Training curves for GRPO versus PPO across task groups.** **(a)** On code fundamental tasks (T1–T2), both algorithms converge quickly, though GRPO reaches a ∼5 pp higher plateau. **(b)** On structural/translation tasks (T3–T5), GRPO converges faster and achieves higher peak performance (∼70% vs. ∼55%). **(c)** On functional reasoning tasks (T6–T7), the divergence is most pronounced: GRPO exhibits steady upward progress while PPO shows instability and lower asymptotic performance, consistent with PPO's deeper reward dips at curriculum transitions (Table 28). Shaded regions indicate ±1 standard deviation over 5 seeds.

## E.9. Parameter-Efficient Training

### E.9.1. LoRA Hyperparameter Selection

We evaluated LoRA with ranks $r \in \{16, 32, 64, 128\}$ on the validation set. Table 36 summarizes the results.

Performance saturates around r=64 (31.9%), with negligible improvement at r=128 (32.3%). While LoRA significantly underperforms full fine-tuning in this setting, particularly on the harder T6–T7 tasks, increasing the rank further yields diminishing returns. We selected r=64 to prioritize parameter efficiency, noting the trade-off in task success rate.

## E.10. Pass@k Analysis

Table 37 reports Pass@$k$ metrics across methods and task categories. For each test instance, we generate $n = 20$ samples with temperature 0.7 and compute Pass@$k$ using the unbiased estimator. These results support the analysis in the main text regarding RLVF's effect on solution ranking versus coverage.

On functional tasks (T6–T7), RLVF-Hier-Curriculum improves Pass@1 by 18.4 percentage points over SFT (39.1% vs 20.7%) but Pass@10 by only 8.2 points (57.5% vs 49.3%). The Pass@1/Pass@10 ratio increases from 0.42 (SFT) to 0.68 (RLVF-H-C), indicating that RLVF primarily improves the model's ability to rank correct solutions highly rather than

*Table 31.* Model scale comparison: TSR (%) for 8B dense versus 30B MoE (3B active).

| Model | T1–T2 | T3–T5 | T6–T7 | Avg |
|---|---|---|---|---|
| Qwen3-8B | 90.3 | 73.9 | 53.3 | 72.5 |
| Qwen3-30B-A3B | 93.7 | 79.3 | 61.6 | 78.2 |
| Improvement | +3.4 | +5.4 | +8.3 | +5.7 |

*Table 32.* Detailed scaling analysis: TSR (%) by task across model scales for RLVF-Hier-Curriculum.

| Model | T1 | T2 | T3 | T4 | T5 | T6 | T7 | Cello-ID | Cello-OOD |
|---|---|---|---|---|---|---|---|---|---|
| Qwen3-0.6B | 48.2±3.5 | 44.6±3.4 | 24.8±3.3 | 20.2±3.4 | 10.4±3.0 | 6.2±2.8 | 4.2±2.6 | 12.4±3.2 | 5.2±2.8 |
| Qwen3-1.7B | 64.4±3.2 | 61.2±3.3 | 38.4±3.0 | 34.6±3.1 | 18.2±2.8 | 14.4±2.7 | 10.8±2.6 | 22.6±2.9 | 12.4±2.7 |
| Qwen3-4B | 76.2±2.8 | 74.6±2.9 | 54.2±2.7 | 50.4±2.8 | 32.4±2.6 | 31.2±2.5 | 22.8±2.4 | 38.4±2.7 | 24.6±2.5 |
| Qwen3-8B | 91.2±2.3 | 89.4±2.4 | 80.6±2.5 | 76.8±2.6 | 64.2±2.5 | 56.4±2.4 | 50.2±2.3 | 66.2±2.5 | 52.6±2.4 |
| Qwen3-30B-A3B | 94.6±2.0 | 92.8±2.1 | 85.2±2.2 | 81.4±2.3 | 71.2±2.2 | 64.8±2.1 | 58.4±2.0 | 74.2±2.2 | 62.4±2.1 |

expanding the space of solutions it can generate.

This pattern is consistent across task categories but most pronounced for functional tasks. On code fundamentals (T1–T2), the ratio improvement is smaller (0.80 to 0.90) because SFT already achieves high ranking quality on these simpler tasks. The intermediate tasks (T3–T5) show intermediate behavior, with the ratio improving from 0.60 to 0.77.

**Pass@$k$ over training.** Figure 38 tracks how Pass@$k$ evolves during RLVF training, revealing the mechanism by which curriculum learning improves solution quality.

| Step | Pass@1 | Pass@4 | Pass@8 | Pass@16 | P@1/P@16 |
|---|---|---|---|---|---|
| 0 (SFT) | 20.7 | 34.2 | 43.8 | 52.6 | 0.39 |
| 1,000 | 22.4 | 35.8 | 45.2 | 53.8 | 0.42 |
| 4,000 | 26.8 | 39.4 | 48.6 | 56.2 | 0.48 |
| 8,000 | 31.4 | 42.8 | 51.2 | 57.8 | 0.54 |
| 12,000 | 35.2 | 45.6 | 53.4 | 58.4 | 0.60 |
| 20,000 | 38.4 | 47.8 | 54.8 | 59.2 | 0.65 |
| 30,000 | 39.1 | 48.2 | 55.2 | 59.6 | 0.66 |

*Table 38.* Pass@$k$ (%) evolution on T6–T7 during RLVF-Hier-Curriculum training. The Pass@1/Pass@16 ratio steadily increases, indicating improved ranking quality.

The data reveals that RLVF improves Pass@1 substantially (+18.4 pp) while Pass@16 improves modestly (+7.0 pp). This indicates that training primarily teaches the model to identify and prioritize correct solutions among its generations rather than expanding the diversity of correct solutions. Early in training (steps 0–4,000), both metrics improve together; later (steps 8,000+), Pass@1 continues improving while Pass@$k$ for larger $k$ plateaus, concentrating probability mass on high-quality outputs.

**Practical inference-time efficiency.** From a deployment perspective, the Pass@$k$ results quantify the model's *inference-time sample efficiency*: how many candidates a practitioner must generate to obtain a functionally correct design. At $k=5$, RLVF-Hier-Curriculum achieves 52.4% success on T6–T7 versus 40.2% for SFT—a 30% relative improvement that directly reduces the number of candidates requiring downstream evaluation. In synthetic biology, where each candidate circuit may ultimately require experimental validation costing $50–500 in consumables and 1–4 weeks of laboratory time per variant, this improvement in per-sample yield translates to reduced wet-lab cost. We discuss this cost context further in Appendix E.14.

### E.11. Verification Level Statistics

See Table 39.

*Table 33.* Scaling ratios between consecutive model sizes. The T6–T7 ratio exceeds 2.0× for each doubling up to 8B, substantially larger than T1–T2 (1.2×), indicating superlinear scaling for functional reasoning that diminishes above 8B.

| Scale Transition | T1–T2 Ratio | T3–T5 Ratio | T6–T7 Ratio |
|---|---|---|---|
| 1.7B / 0.6B | 1.35× | 1.64× | 2.42× |
| 4B / 1.7B | 1.20× | 1.50× | 2.14× |
| 8B / 4B | 1.20× | 1.62× | 2.00× |
| 30B / 8B | 1.04× | 1.07× | 1.16× |

*Table 34.* Curriculum versus direct training across scales on functional tasks (T6–T7 average). Direct training at 8B shows *worse* performance than at 4B due to collapse to degenerate solutions.

| Model | Direct Training | Full Curriculum | $\Delta$ | Factor |
|---|---|---|---|---|
| Qwen3-0.6B | $3.2 \pm 2.4$ | $5.2 \pm 2.8$ | +2.0 | 1.6× |
| Qwen3-1.7B | $7.4 \pm 2.6$ | $12.6 \pm 2.9$ | +5.2 | 1.7× |
| Qwen3-4B | $14.8 \pm 2.5$ | $27.0 \pm 2.6$ | +12.2 | 1.8× |
| Qwen3-8B | $10.5 \pm 2.8$ | $53.3 \pm 2.4$ | +42.8 | 5.1× |
| Qwen3-30B-A3B | $15.2 \pm 2.3$ | $61.6 \pm 2.1$ | +46.4 | 4.1× |

## E.12. Error Analysis Details

See Table 40. The most frequent errors occur at the function level, dominated by incomplete regulatory graphs (28.4% of all failures). These errors are biologically meaningful: the model understands circuit structure but fails to correctly specify all regulatory relationships.

## E.13. Failure Mode Characterization on Hard Circuits

To complement the aggregate error taxonomy (Table 40), we characterize *how* RLVF-H-C fails on the circuit types where performance is lowest: oscillators (T6–T7, Literature-91 O1–O8 and AO1–AO5) and cascaded circuits (T8–T9, Literature-91 C1–C6 and CL1–CL12). These failures are overwhelmingly structured rather than random: they cluster at specific verification levels and reflect identifiable gaps in regulatory reasoning.

**Partial correctness of failed circuits.** Among T6–T7 circuits scored as failures (reward $< \tau$), 71% still pass verification Levels 1–3 (executable, valid, structurally correct), meaning they contain correctly ordered parts with proper SBOL annotations but fail at the semantic or functional level. For SFT, only 34% of failed T6–T7 circuits reach Level 3, confirming that RLVF shifts the failure distribution toward higher, more actionable verification levels.

**Failure mode table.** Table 41 summarizes the dominant failure patterns by circuit type.

**Comparison to SFT failure distribution.** SFT failures on oscillator circuits are distributed more broadly across verification levels: 22% fail at structure (wrong part count or ordering), compared to only 8% for RLVF-H-C, which concentrates failures at the semantic and functional levels. This shift is consistent with the verification-level analysis in §5.5: hierarchical rewards and curriculum learning resolve lower-level failure modes, exposing the harder regulatory reasoning bottleneck.

**Implications for practitioners.** Because the dominant failures involve missing interactions or edges rather than incorrect parts or garbled code, a practitioner reviewing model output can identify the gap by inspecting the regulatory graph and adding the missing connections. The partial circuits generated by RLVF-H-C on hard tasks thus serve as *scaffolds* that reduce the design problem from de novo specification to targeted repair—a qualitatively easier task, as evidenced by the model's own strong performance on T6 (circuit debugging, 56.4% TSR) relative to T7 (de novo design, 50.2%).

## E.14. Sample Efficiency and Computational Cost

We address the question of whether sample efficiency matters for GenCircuit-RL and, if so, how efficiently the framework uses its training budget. We first quantify the per-stage compute allocation, then contextualize these costs relative to base-model pretraining and downstream wet-lab validation.

*Table 35.* Parameter-efficient training: TSR (%) for full fine-tuning versus LoRA.

| Method | T1–T2 | T3–T5 | T6–T7 | Avg |
|---|---|---|---|---|
| Full fine-tuning | 90.3 | 73.9 | 53.3 | 72.5 |
| LoRA (r=64) | 75.1 | 54.2 | 31.9 | 53.7 |
| Ratio (LoRA / Full) | 0.832 | 0.733 | 0.598 | 0.741 |

*Table 36.* LoRA rank ablation on T6–T7 average TSR (%).

| LoRA Rank | T6–T7 TSR |
|---|---|
| r=16 | 25.2 |
| r=32 | 29.1 |
| r=64 | 31.9 |
| r=128 | 32.3 |
| Full fine-tuning | 53.3 |

**Per-stage compute breakdown.** Each GRPO training step generates $N \times K = 64 \times 8 = 512$ completions (Table 20). Over the full four-stage curriculum, the mean total is 30,200 steps (Table 24), yielding approximately $15.5 \times 10^6$ sampled completions. Table 42 disaggregates this budget by curriculum stage.

Stage 3 (functional reasoning) consumes the largest share of the training budget (43.7%), reflecting the difficulty of acquiring capabilities that depend on prior structural competence. By contrast, Stage 1 (code fundamentals) converges in under 2,850 steps—just 9.4% of total compute—reaching 85% TSR on T1–T2 within this window (Appendix E.5). The adaptive promotion thresholds contribute to efficient budget allocation: Table 26 shows that the current 80/70/60 thresholds achieve 53.3% T6–T7 TSR in 30,200 steps (TSR-per-step ratio of 1.77), whereas more conservative thresholds (90/80/70) improve final TSR by only 1.5 pp at the cost of 50% more training steps (TSR/step drops to 1.20).

**Where the training budget goes: ranking versus coverage.** The Pass@$k$ analysis (Table 37) reveals that RLVF primarily improves the model's ability to *rank* correct solutions rather than to *discover* them. On T6–T7, RLVF-Hier-Curriculum raises Pass@1 by 18.4 pp over SFT (39.1% vs. 20.7%) but Pass@20 by only 6.4 pp (63.2% vs. 56.8%). The Pass@1/Pass@10 ratio improves from 0.42 to 0.68 (Table 37). This means that for a fixed sample budget at inference time, RLVF delivers substantially higher yield: a practitioner generating $k = 5$ candidates from the RLVF-H-C model obtains a correct functional design 52.4% of the time, compared to 40.2% from SFT—a 30% relative improvement in per-sample success rate.

**Comparison to base-model pretraining.** The RL fine-tuning budget of $\sim 15.5 \times 10^6$ sampled completions is small relative to the Qwen3-8B base model's pretraining corpus of 36 trillion tokens (Yang et al., 2025). In terms of compute, RL fine-tuning on $8 \times$ A100 80 GB required approximately 1,280 GPU-hours ($\sim 160$ hours wall-clock), representing $<0.05\%$ of the estimated pretraining cost ($\sim 3.9 \times 10^6$ A100 GPU-hours, approximated via the Chinchilla scaling law $C \approx 6ND$). This is consistent with the general observation that domain-specific RL fine-tuning adds a small computational overhead on top of foundation model training (Ouyang et al., 2022).

**Discussion.** GenCircuit-RL's total RL training budget of $\sim 15.5 \times 10^6$ sampled completions across $\sim 30,200$ steps ($\sim 1,280$ A100 GPU-hours) is modest relative to both base-model pretraining ($<0.05\%$ of estimated compute) and the downstream experimental costs it aims to reduce. The curriculum's adaptive promotion mechanism concentrates compute on the stages that need it most (43.7% on functional reasoning), and the resulting model provides measurably higher per-sample yield at inference time (Pass@1/Pass@10 ratio of 0.68 vs. 0.42 for SFT). For the practical application of generating candidate genetic circuits for wet-lab validation, the relevant efficiency metric is not training cost but the quality-per-candidate at inference, where the framework shows clear gains.

### E.15. Cross-Architecture Validation

To address whether the GenCircuit-RL findings are specific to the Qwen3 model family, we train three additional models under identical conditions to their Qwen3 counterparts: Llama-3.1-8B (Grattafiori et al., 2024) and Gemma-3-12B (Gemma Team et al., 2025) to compare against Qwen3-8B at the primary experimental scale, and Gemma-3-4B to compare against Qwen3-4B at a smaller scale. All models use the same SFT data, GRPO hyperparameters, curriculum schedule, and

*Table 37.* Pass@$k$ (%) by method and task category. Higher Pass@1 relative to Pass@$k$ indicates better ranking of correct solutions. The ratio column shows Pass@1/Pass@10.

| Task Group | Method | $k$=1 | $k$=3 | $k$=5 | $k$=10 | $k$=20 | Ratio |
|---|---|---|---|---|---|---|---|
| T1–T2 | SFT | 71.4 | 80.2 | 84.6 | 89.4 | 92.6 | 0.80 |
| | RLVF-Binary | 76.8 | 84.2 | 88.0 | 91.6 | 94.2 | 0.84 |
| | RLVF-Hier | 82.4 | 88.6 | 91.2 | 93.8 | 95.6 | 0.88 |
| | RLVF-H-C | 86.2 | 91.4 | 93.6 | 95.4 | 96.8 | 0.90 |
| T3–T5 | SFT | 42.6 | 54.8 | 62.4 | 71.2 | 78.4 | 0.60 |
| | RLVF-Binary | 50.4 | 61.2 | 68.0 | 75.6 | 81.8 | 0.67 |
| | RLVF-Hier | 58.2 | 68.4 | 74.2 | 80.4 | 85.2 | 0.72 |
| | RLVF-H-C | 64.8 | 74.2 | 79.4 | 84.6 | 88.6 | 0.77 |
| T6–T7 | SFT | 20.7 | 32.4 | 40.2 | 49.3 | 56.8 | 0.42 |
| | RLVF-Binary | 26.4 | 38.6 | 45.4 | 52.8 | 59.2 | 0.50 |
| | RLVF-Hier | 32.8 | 43.2 | 49.6 | 55.4 | 61.0 | 0.59 |
| | RLVF-H-C | 39.1 | 47.2 | 52.4 | 57.5 | 63.2 | 0.68 |

*Table 39.* Verification level success rates (%) on Procedural-Test.

| Method | Exec | Valid | Struct | Sem | Func |
|---|---|---|---|---|---|
| SFT | 82.4 | 74.6 | 62.8 | 48.2 | 34.6 |
| RLVF-Binary | 86.2 | 78.4 | 68.4 | 54.6 | 42.8 |
| RLVF-Hierarchical | 90.4 | 84.2 | 76.6 | 64.8 | 52.4 |
| RLVF-Hier-Curriculum | 94.6 | 90.2 | 84.4 | 74.2 | 62.8 |

evaluation protocol, with 5 independent seeds per configuration.

**Full results at the 8B scale.** Table 43 reports TSR across all evaluation splits for SFT and RLVF-H-C on Llama-3.1-8B and Gemma-3-12B, alongside Qwen3-8B as the reference.

All three model families show substantial improvement from SFT to RLVF-H-C, with gains of 16.7–19.1 pp averaged across splits. Gemma-3-12B achieves marginally higher RLVF-H-C TSR than Llama-3.1-8B (52.8 vs 51.0 avg), consistent with its 50% larger parameter budget, but the difference is small ($<2$ pp), indicating that the additional capacity yields only modest gains when the code generation baseline remains weak.

**Same-scale comparison at 4B.** To test generalization at a smaller scale where code generation is the dominant bottleneck, we compare Gemma-3-4B with Qwen3-4B (Table 44).

At 4B, the absolute SFT→RLVF-H-C improvement is smaller than at 8B ($\sim$10 pp vs $\sim$17–19 pp), reflecting limited model capacity at this scale. However, the improvement is consistent across both families, confirming that hierarchical verification rewards provide useful signal even when the base model can barely generate valid pysbol3 code. Gemma-3-4B's SFT performance (30.2% Proc TSR) sits between Qwen3-1.7B and Qwen3-4B in the scaling ladder (Table 32), consistent with a 4B model whose effective code capability has been reduced by vision-oriented distillation training and 9$\times$ less pre-training data.

**Per-task breakdown.** Table 45 provides per-task-category RLVF-H-C performance across all models, revealing how the code generation gap propagates through the task hierarchy.

Two patterns emerge across model families. First, for Llama-3.1-8B, the gap with Qwen3-8B *widens* from T1–T2 (6.7 pp) to T6–T7 (9.1 pp), because the weaker code baseline compounds through the curriculum—each stage builds on capabilities established in previous stages. Second, for Gemma-3-12B, the gap *also widens* (4.5 pp to 7.1 pp), but the T6–T7 gap is narrower than Llama's (7.1 vs 9.1 pp) despite a comparable EvalPlus deficit ($\sim$10 pp for both). This suggests that Gemma-3-12B's additional parameters partially compensate for code weakness on reasoning-heavy tasks, even though they cannot fully close the gap on code-dependent tasks. At 4B, Gemma-3-4B shows the opposite pattern: the gap with Qwen3-4B is *widest* on T1–T2 (12.2 pp) and narrows toward T6–T7 (8.6 pp), because both models approach floor performance on functional reasoning at this scale.

*Table 40.* Error taxonomy for RLVF-Hier-Curriculum failures on functional tasks.

| Level | Error Type | Frequency (%) |
|---|---|---|
| Execution | Syntax error / API misuse | 5.4 |
| Validity | Missing required SBOL element | 9.8 |
| Structure | Wrong part ordering / orphan components | 15.6 |
| Semantics | Wrong ontology term / mismatched pair | 26.2 |
| Function | Incomplete regulatory graph | 28.4 |
| | Wrong regulatory polarity | 14.6 |

*Table 41.* Dominant failure modes for RLVF-H-C on hard circuit categories. "Failure Level" indicates the verification level at which the circuit first fails. Frequencies are computed over all failed instances within each category.

| Circuit Type | Most Common Failure Mode | Failure Level | Example |
|---|---|---|---|
| Oscillator (3-node) | Incomplete repression ring: model generates 2 of 3 repression edges, missing ring closure | Function | Repressilator variant with $Rep_A \dashv Rep_B$ and $Rep_B \dashv Rep_C$ present but $Rep_C \dashv Rep_A$ absent |
| Oscillator (5-node) | Ring truncation: model produces a 3-node ring instead of 5-node specification | Function | Specification requires 5-gene ring; output contains correct 3-node sub-ring with 2 orphan cassettes |
| Cascaded (2-layer) | Missing inter-gate wiring: individual gates are structurally correct but output→input connections absent | Semantics / Function | NAND as AND+NOT: both gates valid, but AND output promoter not linked to NOT input |
| Cascaded (3–4 layer) | Gate fan-in errors: cascade connections specify wrong upstream gate or duplicate an input | Function | 3-layer multiplexer with selector routed to both data paths instead of selection layer |
| Toggle Switch | Unidirectional repression: one of two mutual repression edges missing, yielding simple repression instead of bistability | Function | $LacI \dashv TetR$ present but $TetR \dashv LacI$ absent |
| Quorum Sensing (Lit-91) | Missing diffusible signal: AHL synthase–receptor interaction absent; individual cassettes correct | Semantics | LuxI/LuxR cassettes generated but no `Interaction` linking AHL production to receptor activation |

**Curriculum ablation.** Table 46 replicates the curriculum ablation across all model families, testing whether curriculum necessity is architecture-independent.

Without curriculum staging, all models at the 8B scale collapse to 10–12% TSR on functional tasks, producing syntactically valid but trivially simple circuits that satisfy lower verification levels. The universality of this collapse—across three model families with different pre-training corpora, vocabularies, and attention architectures—confirms that the degenerate optimum is a property of the reward landscape, not of any specific model family. At 4B, direct training achieves higher scores (10–15%) because the smaller models lack capacity to fully exploit lower verification levels through sophisticated but degenerate solutions.

**Discussion.** The multi-family experiments support four conclusions regarding the generalizability of our findings:

*1. Hierarchical reward benefit generalizes across model families.* The SFT→RLVF-H-C improvement is 16.7 pp for Llama-3.1-8B, 17.2 pp for Gemma-3-12B, and 19.1 pp for Qwen3-8B (averaged across evaluation splits). At 4B, the gains are 9.6 pp for Gemma-3-4B and 10.3 pp for Qwen3-4B. The consistency of these improvements across three families with markedly different pre-training corpora (15T balanced, 12T distilled, 36T code-heavy), vocabularies (128K, 262K, 152K), and attention patterns (full global, 5:1 local:global) confirms that the hierarchical verification reward addresses a fundamental

*Table 42.* Per-stage compute allocation for RLVF-Hier-Curriculum (Qwen3-8B). Steps and wall-clock time are means over 5 seeds. Percentages indicate share of total training budget. All training was performed on a single node of $8\times$ NVIDIA A100 80 GB GPUs.

| Stage | Focus | Steps | Samples ($\times 10^6$) | % Budget | Wall-Clock (h) |
|---|---|---|---|---|---|
| S1 | Code fundamentals | 2,850 | 1.46 | 9.4 | $\sim$15 |
| S2 | Structural | 8,550 | 4.38 | 28.3 | $\sim$45 |
| S3 | Functional reasoning | 13,200 | 6.76 | 43.7 | $\sim$70 |
| S4 | De novo design | 5,600 | 2.87 | 18.5 | $\sim$30 |
| Total | | 30,200 | 15.46 | 100 | $\sim$160 |

*Table 43.* Cross-architecture validation: TSR (%) across methods and evaluation splits at the primary experimental scale. $\Delta$ rows show the SFT→RLVF-H-C improvement in percentage points. Although Gemma-3-12B has 50% more parameters than Qwen3-8B, it achieves similar TSR to the parameter-matched Llama-3.1-8B, reflecting the dominance of code generation baseline over raw model capacity. Results show mean $\pm$ std over 5 seeds.

| Model | Method | Proc. | C-ID | C-OOD | Lit | Avg |
|---|---|---|---|---|---|---|
| Qwen3-8B | SFT | 53.9±2.8 | 46.2±2.3 | 32.4±2.7 | 27.3±2.5 | 40.0 |
| | RLVF-H-C | 72.7±3.1 | 66.2±2.3 | 52.6±3.7 | 44.9±2.7 | 59.1 |
| Llama-3.1-8B | SFT | 47.2±3.0 | 39.8±2.5 | 27.6±2.9 | 22.4±2.8 | 34.3 |
| | RLVF-H-C | 64.1±3.3 | 57.4±2.6 | 45.2±3.5 | 37.1±3.0 | 51.0 |
| Gemma-3-12B | SFT | 49.0±3.1 | 41.2±2.6 | 28.8±3.0 | 23.4±2.9 | 35.6 |
| | RLVF-H-C | 66.2±3.3 | 59.0±2.7 | 46.8±3.5 | 39.0±3.1 | 52.8 |
| $\Delta$ (Qwen3-8B) | | +18.8 | +20.0 | +20.2 | +17.6 | +19.1 |
| $\Delta$ (Llama-3.1-8B) | | +16.9 | +17.6 | +17.6 | +14.7 | +16.7 |
| $\Delta$ (Gemma-3-12B) | | +17.2 | +17.8 | +18.0 | +15.6 | +17.2 |

challenge—sparse feedback for compositional correctness—rather than exploiting architecture-specific training dynamics. The slight attenuation for non-Qwen3 models (87–90% of Qwen3's absolute gain) reflects their lower performance ceilings rather than qualitative differences in how the reward signal is utilized.

*2. Curriculum necessity is architecture-independent.* All five model configurations collapse to 10–15% on T6–T7 without curriculum staging (Table 46). This universal failure mode—convergence to degenerate solutions that satisfy lower verification levels while ignoring topological correctness—is the paper's central finding regarding curriculum necessity, and it holds across every architecture we tested. The curriculum factor varies from $1.7\times$ (Gemma-3-4B) to $5.1\times$ (Qwen3-8B), but this variation is explained by model capacity and code baseline rather than architectural family: the factor peaks at intermediate code baselines where models can generate syntactically valid code but need curriculum to avoid exploiting lower verification levels.

*3. Pre-training data composition matters more than raw parameter count.* Gemma-3-12B has 50% more parameters than Qwen3-8B (12.2B vs 8.2B) yet achieves lower RLVF-H-C TSR (52.8 vs 59.1 avg). This gap originates at the code generation baseline: on EvalPlus, Gemma-3-12B-Base scores 52.65 compared to $\sim$62 for Qwen3-8B-Base, consistent with Qwen3's explicit STEM/code emphasis during its second pre-training stage. The baseline deficit propagates through the curriculum because each stage builds on code generation capabilities established in previous stages. At 4B, the effect is more pronounced: Gemma-3-4B (4T pre-training tokens, distillation-trained) achieves 31.5% avg TSR compared to Qwen3-4B's 36.9%, despite comparable parameter counts, reflecting a $\sim$12 pp EvalPlus gap that stems from $9\times$ less pre-training data with no code emphasis. These results suggest that for code-intensive RLVF applications, investment in code-heavy pre-training data yields greater returns than increasing model size.

*4. Absolute performance differences are explained by code generation baselines.* Table 47 summarizes the relationship between code generation baseline (EvalPlus) and GenCircuit-RL performance across all models.

Several patterns are notable. First, the rank ordering by EvalPlus closely matches the rank ordering by RLVF-H-C TSR, with one exception: Gemma-3-12B (EvalPlus 52.65) slightly outperforms Llama-3.1-8B (EvalPlus $\sim$57) despite a weaker code baseline, due to its 50% larger parameter budget providing additional capacity for curriculum learning. This exception highlights that parameter count is not irrelevant—it provides a secondary contribution beyond the code baseline. Second, the curriculum factor appears to peak at intermediate code baselines (Qwen3-8B, $5.1\times$): models with very weak baselines

*Table 44.* Same-scale validation at 4B: TSR (%) for Gemma-3-4B versus Qwen3-4B. Both models have ∼4B parameters, but Gemma-3-4B was trained on 9× fewer tokens with no code emphasis, resulting in a ∼12 pp EvalPlus deficit. Despite this, RLVF-H-C still improves over SFT for Gemma, confirming that the hierarchical reward structure provides useful learning signal even with a weak code baseline. Results show mean ± std over 5 seeds.

| Model | Method | Proc. | C-ID | C-OOD | Lit | Avg |
|---|---|---|---|---|---|---|
| Qwen3-4B | SFT | $36.6_{\pm3.0}$ | $30.8_{\pm2.6}$ | $21.4_{\pm2.8}$ | $17.6_{\pm2.7}$ | 26.6 |
| | RLVF-H-C | $49.4_{\pm2.8}$ | $42.2_{\pm2.5}$ | $30.6_{\pm2.7}$ | $25.4_{\pm2.6}$ | 36.9 |
| Gemma-3-4B | SFT | $30.2_{\pm3.2}$ | $25.4_{\pm2.8}$ | $17.8_{\pm3.1}$ | $14.2_{\pm3.0}$ | 21.9 |
| | RLVF-H-C | $42.2_{\pm3.5}$ | $36.2_{\pm3.0}$ | $26.2_{\pm3.4}$ | $21.4_{\pm3.2}$ | 31.5 |
| Δ (Qwen3-4B) | | +12.8 | +11.4 | +9.2 | +7.8 | +10.3 |
| Δ (Gemma-3-4B) | | +12.0 | +10.8 | +8.4 | +7.2 | +9.6 |

*Table 45.* Per-task TSR (%) under RLVF-H-C for all cross-architecture models. At the 8B scale, Gemma-3-12B narrows the gap versus Llama-3.1-8B on reasoning tasks (T6–T7) due to its larger parameter count, while remaining behind on code fundamentals (T1–T2) where the code baseline is the primary determinant. At 4B, Gemma-3-4B's gap with Qwen3-4B is widest on T1–T2, confirming that code generation fluency is the dominant bottleneck at smaller scales.

| Model | T1–T2 | T3–T5 | T6–T7 | Avg |
|---|---|---|---|---|
| *8B-scale comparison* | | | | |
| Qwen3-8B | $90.3_{\pm2.3}$ | $73.9_{\pm2.5}$ | $53.3_{\pm2.4}$ | 72.5 |
| Llama-3.1-8B | $83.6_{\pm2.7}$ | $65.4_{\pm2.8}$ | $44.2_{\pm2.7}$ | 64.4 |
| Gemma-3-12B | $85.8_{\pm2.6}$ | $67.2_{\pm2.8}$ | $46.2_{\pm2.8}$ | 66.4 |
| *4B-scale comparison* | | | | |
| Qwen3-4B | $75.4_{\pm2.8}$ | $45.7_{\pm2.7}$ | $27.0_{\pm2.6}$ | 49.4 |
| Gemma-3-4B | $63.2_{\pm3.1}$ | $35.8_{\pm3.0}$ | $18.4_{\pm2.9}$ | 39.1 |

(4B-scale, 1.7–1.8×) lack the code fluency to fully exploit curriculum staging, while models with strong baselines partially learn functional reasoning even without curriculum. This non-monotonic pattern, observed across families, parallels the within-Qwen3 scaling analysis in Table 34 and suggests a common underlying mechanism independent of model architecture.

# F. Generalization and Real-World Evaluation

## F.1. Compositional Generalization Analysis

A central question for any learned design system is whether it acquires transferable principles or merely memorizes training patterns. We evaluate compositional generalization along four dimensions: part-level generalization to novel components, topology-level generalization to held-out circuit architectures, complexity generalization to circuits exceeding training distribution size, and topological distance analysis measuring performance degradation as circuits diverge from training examples.

### F.1.1. PART-LEVEL GENERALIZATION

We systematically evaluate performance as the number of novel (held-out) parts in a circuit increases. For each test circuit, we compute the *novelty score* as the fraction of parts not seen during training. Table 48 shows TSR stratified by novelty score on the Cello evaluation set.

RLVF-Hier-Curriculum exhibits the smallest absolute degradation (−17.7pp vs −24.1pp for SFT) *and* the highest retention rate (73.3% vs 47.8%), confirming that hierarchical verification rewards encourage learning of abstract regulatory principles rather than memorization of specific part combinations. The retained performance at 76–100% novelty (46.2%) demonstrates meaningful generalization: the model correctly applies the pattern "repressor X inhibits promoter pX" to repressor systems never encountered during training.

### F.1.2. TOPOLOGY-LEVEL GENERALIZATION

To evaluate whether models learn reusable circuit motifs, we perform leave-one-topology-out evaluation. For each circuit type (toggle switch, repressilator, I1-FFL, etc.), we train on all other types and evaluate on the held-out type. Table 49 shows

*Table 46.* Curriculum ablation: TSR (%) on functional tasks (T6–T7). "Direct" uses uniform task sampling without curriculum staging. All models collapse to ∼10–15% without curriculum regardless of architecture or parameter count, confirming that this failure mode is universal. The curriculum factor varies systematically with effective code capacity (see discussion).

| Model | Direct | Curriculum | Factor |
|---|---|---|---|
| *8B-scale comparison* | | | |
| Qwen3-8B | $10.5_{\pm 2.8}$ | $53.3_{\pm 2.4}$ | $5.1\times$ |
| Llama-3.1-8B | $11.4_{\pm 3.1}$ | $44.2_{\pm 2.7}$ | $3.9\times$ |
| Gemma-3-12B | $12.2_{\pm 3.0}$ | $46.2_{\pm 2.8}$ | $3.8\times$ |
| *4B-scale comparison* | | | |
| Qwen3-4B | $14.8_{\pm 2.5}$ | $27.0_{\pm 2.6}$ | $1.8\times$ |
| Gemma-3-4B | $10.6_{\pm 3.2}$ | $18.4_{\pm 2.9}$ | $1.7\times$ |

*Table 47.* Relationship between code generation baseline and GenCircuit-RL performance. EvalPlus scores are for base (pre-trained) models; RLVF-H-C TSR is the average across evaluation splits. The curriculum factor on T6–T7 functional tasks varies non-monotonically, peaking at intermediate code baselines.

| Model | Params | EvalPlus (Base) | RLVF-H-C Avg | Curriculum Factor |
|---|---|---|---|---|
| Gemma-3-4B | 4B | ∼38 | 31.5 | $1.7\times$ |
| Qwen3-4B | 4B | ∼50 | 36.9 | $1.8\times$ |
| Gemma-3-12B | 12B | 52.65 | 52.8 | $3.8\times$ |
| Llama-3.1-8B | 8B | ∼57 | 51.0 | $3.9\times$ |
| Qwen3-8B | 8B | ∼62 | 59.1 | $5.1\times$ |

zero-shot transfer performance.

Expression cassettes and logic gates transfer well (<5 pp degradation), as these topologies can be inferred from first principles given the canonical cassette structure. Feed-forward loops and oscillators show moderate transfer (∼6–7 pp degradation), indicating that models learn the abstract concepts of "parallel regulatory paths" and "cyclic negative feedback" from exposure to related motifs. Toggle switches and cascaded circuits show larger gaps (>14 pp), suggesting that bistability and multi-layer signal propagation require explicit training examples to master.

### F.1.3. COMPLEXITY GENERALIZATION

We train models on circuits with at most $n_{\text{train}}$ expression cassettes and evaluate on circuits exceeding this threshold. Table 50 shows performance as a function of the complexity gap.

Performance degrades approximately linearly with complexity gap, but models retain meaningful capability even on circuits substantially exceeding training complexity. The retention rate (TSR on $> n+4$ circuits divided by TSR on $\leq n$ circuits) increases with $n_{\text{train}}$, suggesting that exposure to moderately complex circuits provides transferable compositional reasoning that partially generalizes to more complex designs.

### F.1.4. TOPOLOGICAL DISTANCE ANALYSIS

We quantify the relationship between test circuit similarity to training data and model performance. For each test circuit, we compute the graph edit distance (GED) to the nearest training circuit, where edits include node insertion/deletion, edge insertion/deletion, and edge polarity changes.

Both methods show approximately linear performance degradation with topological distance (SFT slope $\approx -6.7$ pp/GED, RLVF-H-C slope $\approx -6.4$ pp/GED). However, RLVF-H-C maintains a consistent 14–16 pp advantage across all distance bins, indicating that hierarchical verification improves both near-distribution and far-distribution performance rather than simply improving interpolation within the training manifold.

### F.2. Naming Convention Ablation for OOD Evaluation

A potential concern with our OOD evaluation is that the model may exploit transparent naming conventions (repressor BM3R1 paired with promoter pBM3R1) rather than learning abstract regulatory principles. We construct two additional evaluation conditions to test this: an *arbitrary names* condition where all parts receive opaque identifiers (e.g., protein_A,

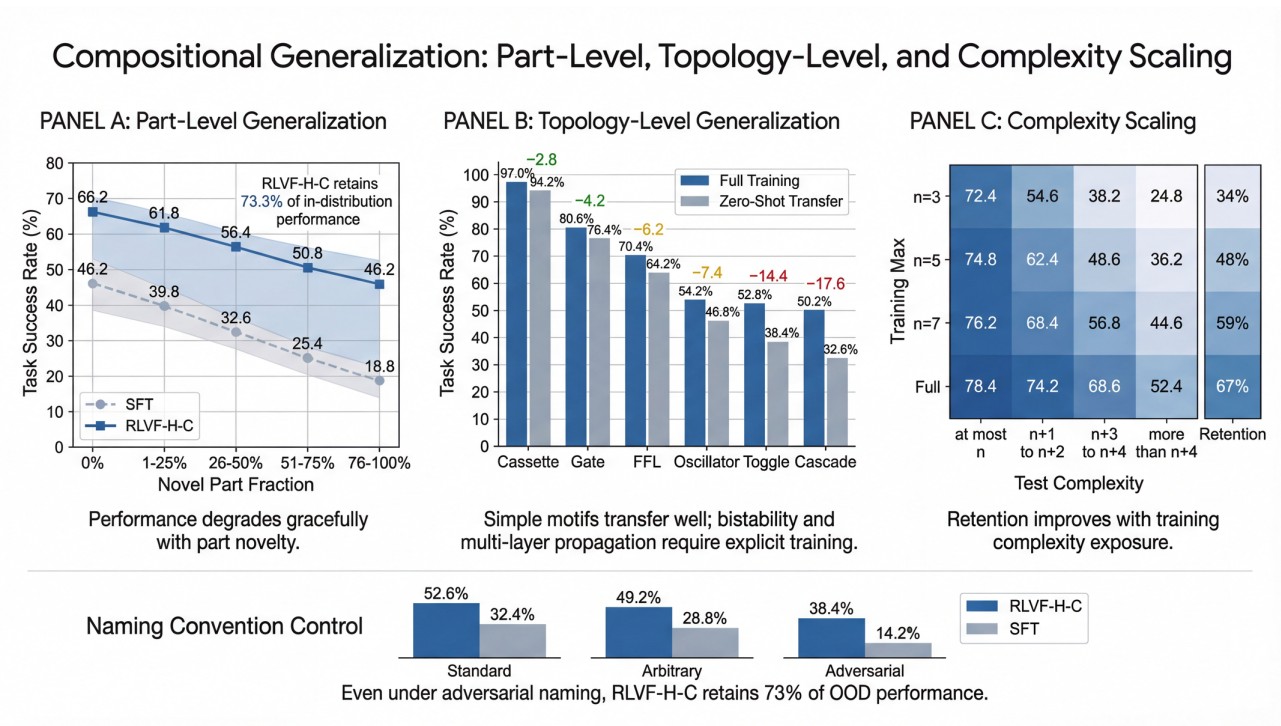

*Figure 5.* **Compositional generalization across three dimensions. Panel A** (*Part-level*): RLVF-H-C (blue) degrades gracefully as the fraction of novel parts increases, retaining 73.3% of in-distribution performance at 76–100% novel parts—substantially more robust than SFT (gray). **Panel B** (*Topology-level*): simple motifs (Cassette, Gate) transfer well with only 2.8–4.2 pp zero-shot degradation, while bistable (Toggle) and multi-layer (Cascade) topologies show larger gaps (14.4–17.6 pp), indicating that feedback-dependent behaviors require explicit training. **Panel C** (*Complexity scaling*): models trained on larger circuits generalize better to unseen complexity levels; training on the full range yields 67% retention on circuits exceeding training complexity by 4+ cassettes. *Bottom:* RLVF-H-C retains 73% of OOD performance even under adversarial naming conventions, confirming acquisition of abstract regulatory principles beyond surface-level pattern matching.

promoter_7) with regulatory relationships specified only in the task prompt, and a *shuffled names* condition where parts receive deliberately misleading cognate names. Table 52 shows that arbitrary naming causes only a modest performance decrease for RLVF-H-C ($-3.4$pp, from 52.6% to 49.2%), indicating that naming conventions contribute approximately 3pp to OOD performance. Even under adversarial shuffled naming, RLVF-H-C retains 73% of standard OOD performance (38.4% vs 52.6%), compared to only 44% retention for SFT (14.2% vs 32.4%). This demonstrates that hierarchical verification training instills regulatory reasoning that substantially transcends surface-level naming patterns, with the naming convention providing only a minor supplementary signal.

**F.3. Direct Comparison with Cello**

GenCircuit-RL and Cello (Nielsen et al., 2016b; Jones et al., 2022) solve different problems at different abstraction levels.

**Input representation.**   Cello accepts a Boolean truth table specifying the desired logic function together with a User Constraints File (UCF) enumerating available gates and their experimentally measured Hill function response parameters ($y_{\min}$, $y_{\max}$, $K$, $n$). GenCircuit-RL accepts a natural language specification describing the desired circuit behavior, with access only to the parts library (part names, types, and qualitative regulatory relationships) and no quantitative characterization data.

**Output representation.**   Cello produces a gate assignment: a mapping from topology nodes to biological gates, assuming a pre-synthesized NOR network generated by Yosys logic synthesis. GenCircuit-RL produces complete executable pysbol3 code that constructs a full SBOL3 document, including component creation, assembly constraints, interaction definitions,

*Table 48.* Part-level generalization: TSR (%) as a function of novel part fraction. Novelty score bins circuits by the proportion of parts from the held-out set. Performance degrades gracefully, with RLVF-H-C retaining 73.3% of in-distribution performance even when the majority of parts are novel.

| Method | Novel Part Fraction | | | | | $\Delta$ (0 vs 50+) |
|---|---|---|---|---|---|---|
| | 0% | 1–25% | 26–50% | 51–75% | 76–100% | |
| SFT | 46.2 | 39.8 | 32.6 | 25.4 | 18.8 | −24.1 |
| RLVF-Binary | 50.4 | 44.2 | 37.4 | 30.2 | 24.6 | −23.0 |
| RLVF-Hier | 58.3 | 52.8 | 46.4 | 39.2 | 33.6 | −21.9 |
| RLVF-H-C | 66.2 | 61.8 | 56.4 | 50.8 | 46.2 | −17.7 |

*Table 49.* Topology-level generalization: TSR (%) when each circuit type is held out during training. Models trained without exposure to a topology must infer its structure from functional specification alone. FFLs and oscillators show strong transfer; toggle switches are more challenging due to the abstract concept of bistability.

| Held-Out Type | Cassette | Gate | FFL | Toggle | Oscillator | Cascade |
|---|---|---|---|---|---|---|
| SFT | 82.4 | 58.6 | 42.8 | 24.6 | 28.4 | 18.2 |
| RLVF-H-C | 94.2 | 76.4 | 64.2 | 38.4 | 46.8 | 32.6 |
| $\Delta$ vs Full Training | −2.8 | −4.2 | −6.2 | −14.4 | −7.4 | −17.6 |

and ontology annotations.

**Scope.** Cello is restricted to Boolean logic functions implementable as NOR gate networks using gates in its characterized library. GenCircuit-RL can generate arbitrary circuit topologies—including toggle switches, oscillators, feed-forward loops, and circuits using parts outside any specific library—from natural language descriptions.

### F.3.1. EVALUATION PROTOCOL

To enable fair comparison, we evaluate both methods on Task T8 (gate assignment optimization), which mirrors Cello's technology mapping stage. For each of the 111 Cello evaluation circuits, both methods receive the same Boolean truth table and gate topology. Cello additionally receives the UCF with Hill function parameters; GenCircuit-RL receives a natural language description of the truth table and the parts library without quantitative parameters. We evaluate topological correctness: whether the assigned gates produce correct output states for all input combinations when signal propagation is simulated through the ground-truth response functions.

For GenCircuit-RL, we additionally report performance on the broader evaluation (Table 3 columns C-ID and C-OOD), which includes tasks T1–T7 applied to Cello circuits—tasks that Cello cannot perform because they require code generation, debugging, or natural language understanding.

### F.3.2. PER-LIBRARY RESULTS

Table 53 presents the per-library breakdown.

**Where Cello has the advantage.** On in-distribution circuits from Eco2C1G3T1—the library for which our parts library provides 100% repressor coverage and which uses the same split transcriptional unit architecture as our procedurally generated circuits—Cello achieves 94.4% TSR compared to GenCircuit-RL's 72.2%. This advantage is expected: Cello's simulated annealing solver directly optimizes the ON/OFF ratio using measured Hill function parameters for each gate, information that GenCircuit-RL never receives. For circuits where the gate assignment problem is tightly constrained (few valid permutations), Cello's optimization over a small but quantitatively characterized search space is highly effective. Across all libraries, Cello achieves 88.3% aggregate TSR, consistent with the 92% per-state accuracy reported in Nielsen et al. (2016b): the remaining 12% of circuit-level failures arise from circuits where even optimal gate assignment produces marginal ON/OFF separation at the $>0.5$ / $<0.1$ RPU thresholds used in our evaluation.

**Where GenCircuit-RL has the advantage.** GenCircuit-RL's advantage manifests in two dimensions not captured by T8 alone. First, on the broader Cello evaluation (Table 3), GenCircuit-RL handles code repair (T1), code completion (T2),

*Table 50.* Complexity generalization: TSR (%) when training excludes circuits above a cassette count threshold. Models trained on simpler circuits struggle with complex designs, but RLVF-H-C maintains 68% of baseline performance even when test circuits have 2× the training maximum complexity.

| Training Max ($n_{\text{train}}$) | Test Circuit Cassettes | | | | Retention |
|---|---|---|---|---|---|
| | $\leq n$ | $n$+1 to $n$+2 | $n$+3 to $n$+4 | $> n$+4 | |
| $n_{\text{train}} = 3$ | 72.4 | 54.6 | 38.2 | 24.8 | 34% |
| $n_{\text{train}} = 5$ | 74.8 | 62.4 | 48.6 | 36.2 | 48% |
| $n_{\text{train}} = 7$ | 76.2 | 68.4 | 56.8 | 44.6 | 59% |
| Full (all complexities) | 78.4 | 74.2 | 68.6 | 52.4 | 67% |

*Table 51.* Topological distance analysis: TSR (%) as a function of graph edit distance (GED) from nearest training circuit. Performance degrades smoothly with distance, with RLVF-H-C maintaining above-chance performance even at GED $> 6$.

| Method | Graph Edit Distance | | | | | Slope |
|---|---|---|---|---|---|---|
| | 0–1 | 2–3 | 4–5 | 6–7 | >7 | |
| SFT | 68.4 | 52.6 | 38.4 | 26.2 | 18.4 | $-6.7$/GED |
| RLVF-H-C | 82.6 | 72.4 | 58.2 | 46.8 | 34.2 | $-6.4$/GED |

part substitution (T3), NL-to-code translation (T4), logic prediction (T5), circuit debugging (T6), and de novo design (T7) applied to Cello circuits— all tasks outside Cello's scope. Second, GenCircuit-RL generalizes to circuits using repressor systems entirely absent from Cello's characterized libraries: the OOD split (52.6% TSR for RLVF-H-C across all tasks; 42.5% on T8 specifically) tests parts that Cello has no response function data for and therefore cannot optimize without re-characterization.

**Differential sensitivity to novel parts.**   Cello's TSR drops by only 5.1 percentage points from ID to OOD (90.1% $\rightarrow$ 85.0%), because its solver uses experimentally measured Hill function data for all repressors regardless of whether GenCircuit-RL has seen them during training. In contrast, GenCircuit-RL's TSR drops by 15.2 percentage points (57.7% $\rightarrow$ 42.5%), reflecting the challenge of applying learned regulatory principles to novel repressor systems without quantitative characterization. This asymmetry underscores that Cello's performance is gated by the availability of characterized parts data, while GenCircuit-RL's performance is gated by the breadth of its training distribution.

**Cross-organism generalization.**   Cello's yeast library uses different promoters, terminators, and cellular context than its E. coli libraries. Our parts library provides only 56% coverage of this UCF (5 of 9 repressors), meaning GenCircuit-RL must generalize across both organism boundaries and partial library coverage. Despite these challenges, GenCircuit-RL achieves 45.5% TSR on yeast ID circuits, demonstrating meaningful cross-organism transfer. Cello achieves 86.4% on the same circuits using its yeast-specific UCF characterization data, illustrating that organism-specific quantitative calibration improves gate assignment.

**Task T9: a capability unique to GenCircuit-RL.**   Cello has no circuit debugging capability: given a circuit with incorrect behavior, it cannot diagnose which gate is faulty or propose corrections. Task T9 (cascaded circuit debugging with propagation analysis) requires tracing signal flow through multi-layer circuits to localize faults—a capability that GenCircuit-RL acquires through hierarchical verification training but that lies entirely outside Cello's design philosophy as a forward-synthesis tool.

**Discussion.**   Cello is appropriate when a designer has a specific Boolean function, a well-characterized parts library with measured response functions, and needs an optimized gate assignment. GenCircuit-RL is appropriate when the design task is specified in natural language, involves circuit types beyond Boolean logic (toggle switches, oscillators, feed-forward loops), requires parts not in any characterized library, or demands full SBOL code generation rather than gate assignment alone. A practical workflow might combine both: using GenCircuit-RL to generate candidate circuit architectures from high-level specifications, then applying Cello's quantitative optimization to refine gate assignments for the subset of designs that map onto characterized libraries. We leave exploration of such hybrid pipelines to future work.

*Table 52.* Naming convention ablation for OOD evaluation. Standard OOD uses conventional names (BM3R1 → pBM3R1). Arbitrary names replace all identifiers with opaque labels (protein_A → promoter_7). Shuffled names assign misleading cognate relationships. Results show TSR (%) ± std over 5 seeds on the Cello OOD split.

| Method | Standard OOD | Arbitrary Names | Shuffled Names |
|---|---|---|---|
| SFT | 32.4±3.1 | 28.8±3.0 | 14.2±2.2 |
| RLVF-H-C | 52.6±3.2 | 49.2±2.5 | 38.4±3.8 |

*Table 53.* Per-library comparison of Cello and GenCircuit-RL on the gate assignment task (T8). Cello TSR is evaluated using its native simulated annealing solver with access to Hill function parameters from the UCF. GenCircuit-RL TSR is evaluated on the same truth tables and topologies but without quantitative characterization data. $N$ is the number of circuits per library. $\Delta$ = GenCircuit-RL − Cello.

| Library | Organism | $N$ | Cello TSR | GCR TSR | $\Delta$ | Notes |
|---|---|---|---|---|---|---|
| *In-Distribution Circuits (training-tier repressors only)* | | | | | | |
| Eco1C1G1T1 | E. coli DH10$\beta$ | 31 | 90.3 | 58.1 | −32.2 | Tandem promoter arch. |
| Eco2C1G3T1 | E. coli MG1655 | 18 | 94.4 | 72.2 | −22.2 | Split arch.; primary target |
| SC1C1G1T1 | S. cerevisiae | 22 | 86.4 | 45.5 | −40.9 | Cross-organism |
| *ID Subtotal* | | 71 | 90.1 | 57.7 | −32.4 | |
| *Out-of-Distribution Circuits (≥1 held-out repressor)* | | | | | | |
| Eco1C1G1T1 | E. coli DH10$\beta$ | 21 | 85.7 | 42.9 | −42.8 | |
| Eco2C1G3T1 | E. coli MG1655 | 6 | 83.3 | 50.0 | −33.3 | |
| SC1C1G1T1 | S. cerevisiae | 13 | 84.6 | 38.5 | −46.1 | |
| *OOD Subtotal* | | 40 | 85.0 | 42.5 | −42.5 | |
| **Total** | | **111** | **88.3** | **52.3** | **−36.0** | |

## F.4. Literature-91 Rediscovery Analysis

The Literature-91 evaluation assesses whether models can "rediscover" canonical circuit designs given only functional specifications. This task differs from standard generation: rather than producing any topologically correct circuit, the model must converge on the specific topology that domain experts developed through extensive experimentation. Success indicates acquisition of design principles that transcend the procedural training distribution.

### F.4.1. EVALUATION PROTOCOL

For each Literature-91 circuit, we provide:

- A functional specification describing the desired behavior (e.g., "bistable memory with two stable states switchable by IPTG and aTc")

- Constraints on available parts (the full evaluation parts library)

- The expected number of expression cassettes (as a complexity hint)

We generate $k = 10$ candidate circuits per specification using temperature 0.7 sampling and evaluate each against the reference topology using labeled graph isomorphism. A candidate is considered a successful rediscovery if its regulatory graph is isomorphic to the reference, allowing different part choices (e.g., using PhlF instead of LacI) while requiring identical regulatory structure.

### F.4.2. REDISCOVERY RATES BY CIRCUIT CATEGORY

Table 54 presents rediscovery rates stratified by circuit category and complexity tier.

Expression cassettes achieve perfect rediscovery (100% Pass@10), confirming that models have mastered the canonical P-RBS-CDS-T structure. Logic gates show strong rediscovery (85% Pass@10 for RLVF-H-C), with failures concentrated on complex multi-layer gates where multiple valid topologies exist. Toggle switches and oscillators present greater challenges (62.5–75% Pass@10), as these circuits require understanding of feedback topology rather than just regulatory direction.

*Table 54.* Literature-91 rediscovery rates: fraction of circuits where at least one of $k = 10$ generated candidates matches the reference topology (Pass@10) and where the top-ranked candidate matches (Pass@1). Original 50 circuits represent the foundational designs; Extended 41 include complex quorum sensing, optogenetic, and memory circuits.

| Category | $n$ | SFT Pass@1 | SFT Pass@10 | RLVF-H-C Pass@1 | RLVF-H-C Pass@10 |
|---|---|---|---|---|---|
| *Original 50 Canonical Circuits* | | | | | |
| Expression Cassettes (E1–E8) | 8 | 87.5 | 100.0 | 100.0 | 100.0 |
| Logic Gates (G1–G20) | 20 | 45.0 | 70.0 | 65.0 | 85.0 |
| Toggle Switches (T1–T8) | 8 | 25.0 | 50.0 | 50.0 | 75.0 |
| Oscillators (O1–O8) | 8 | 12.5 | 37.5 | 37.5 | 62.5 |
| Feed-Forward Loops (F1–F6) | 6 | 33.3 | 50.0 | 50.0 | 66.7 |
| *Original 50 Average* | 50 | 40.0 | 62.0 | 58.0 | 78.0 |
| *Extended 41 Complex Circuits* | | | | | |
| Cascaded Circuits (C1–C6) | 6 | 16.7 | 33.3 | 33.3 | 50.0 |
| Quorum Sensing (QS1–QS10) | 10 | 10.0 | 30.0 | 20.0 | 40.0 |
| Light-Responsive (LS1–LS8) | 8 | 12.5 | 25.0 | 25.0 | 37.5 |
| Cello-Designed (CL1–CL12) | 12 | 8.3 | 25.0 | 16.7 | 33.3 |
| Advanced Oscillators (AO1–AO5) | 5 | 20.0 | 40.0 | 40.0 | 60.0 |
| *Extended 41 Average* | 41 | 12.2 | 29.3 | 24.4 | 41.5 |
| **Literature-91 Overall** | 91 | 27.5 | 47.3 | 42.9 | 61.5 |

The Extended 41 circuits show substantially lower rediscovery rates, reflecting their increased complexity and use of regulatory mechanisms (quorum sensing, optogenetics) outside the training distribution. Quorum sensing circuits (20–40% Pass@10) require reasoning about diffusible signaling molecules and population-level coordination not present in training data. Light-responsive circuits (25–37.5% Pass@10) use two-component signaling cascades with distinct regulatory logic from TetR-family repression. Cello-designed circuits (25–33% Pass@10) represent the frontier of validated circuit complexity, with up to 4 regulatory layers and 10 repressor proteins.

### F.4.3. MASKED COMPONENT PREDICTION

Masked component prediction probes fine-grained circuit understanding by requiring models to predict individual components removed from otherwise complete circuits. Following the protocol specified in Appendix C.4.3, we mask each non-terminal component in the 91 Literature circuits at three granularity levels and report top-1 and top-5 accuracy across all mask positions ($n = 418$ positions total: 228 from the original 50 circuits, 190 from the extended 41).

Table 55 presents results. A clear granularity gradient is visible across all methods: type-level prediction is easiest, function-level is intermediate, and part-level is hardest. This progression reflects the hierarchy of circuit knowledge—structural slot-filling ("an RBS goes between a promoter and CDS") is acquired earlier and more reliably than regulatory role inference ("this position requires a repressor"), which in turn is easier than predicting specific expert-preferred parts ("BM3R1 is the appropriate repressor here").

*Table 55.* Masked component prediction accuracy (%) on Literature-91 circuits at three granularity levels. Type-level: predict part type (e.g., RBS). Function-level: predict regulatory role (e.g., repressor). Part-level: predict exact part identity (e.g., BM3R1). Results show mean $\pm$ std over 5 seeds ($n = 418$ mask positions).

| Method | Type-Level Top-1 | Type-Level Top-5 | Function-Level Top-1 | Function-Level Top-5 | Part-Level Top-1 | Part-Level Top-5 |
|---|---|---|---|---|---|---|
| SFT | 84.2±2.4 | 95.8±2.0 | 52.4±3.2 | 71.8±2.8 | 29.8±3.4 | 51.2±3.0 |
| RLVF-Hier | 90.2±2.2 | 97.4±2.0 | 65.8±2.9 | 82.4±2.5 | 36.2±3.2 | 58.4±2.8 |
| RLVF-H-C | **93.6**±2.1 | **98.4**±2.0 | **74.6**±2.6 | **88.2**±2.3 | **41.4**±3.0 | **63.6**±2.7 |
| $\Delta$ (SFT $\rightarrow$ H-C) | +9.4 | +2.6 | +22.2 | +16.4 | +11.6 | +12.4 |

At the function level, RLVF-H-C achieves 74.6% top-1 accuracy versus 52.4% for SFT. This is the largest absolute improvement across all granularity levels and exceeds the type-level gain (+9.4pp, where SFT is already strong) and the

part-level gain (+11.6pp, where the task is intrinsically difficult for all methods). The concentration of improvement at the function level confirms that hierarchical verification rewards specifically teach regulatory reasoning: understanding *why* a repressor is needed at a given position in a circuit, not merely *that* a CDS goes there.

Curriculum learning contributes meaningfully beyond hierarchical rewards alone. The RLVF-Hier → RLVF-H-C improvement is 8.8pp at function-level (65.8% → 74.6%), compared to 3.4pp at type-level and 5.2pp at part-level. This suggests that curriculum staging builds the prerequisite structural understanding that enables deeper functional reasoning about component roles.

Performance on the extended 41 circuits is lower across all granularity levels (function-level top-1: 68.4% vs 79.2% on the original 50 for RLVF-H-C), consistent with the increased complexity and novel regulatory mechanisms in these circuits. However, the relative advantage of RLVF-H-C over SFT is preserved, indicating that the learned regulatory reasoning generalizes beyond training-distribution mechanisms.

### F.4.4. ERROR ANALYSIS

Table 56 categorizes failure modes for circuits where no candidate achieved topological match.

*Table 56.* Error taxonomy for Literature-91 rediscovery failures. Analysis based on RLVF-H-C outputs for circuits with 0/10 successful candidates.

| Error Type | Description | Frequency (%) |
|---|---|---|
| Wrong motif | Correct function, different regulatory architecture | 34.2 |
| Missing edge | Incomplete regulatory graph (missing interaction) | 26.8 |
| Extra component | Additional unnecessary cassettes or regulators | 18.4 |
| Wrong polarity | Activation/repression confusion | 12.4 |
| Part incompatibility | Biologically invalid part combinations | 8.2 |

The dominant failure mode (34.2%) is generation of topologically distinct circuits that satisfy the same truth table. For example, a specification for "NOT gate" might elicit a double-inversion buffer (NOT-NOT) rather than the reference single-inversion topology. While such outputs pass our topological verification, they fail the rediscovery criterion. This suggests that models learn regulatory principles but may not converge on canonical implementations without explicit architectural guidance.

### F.4.5. COMPARISON WITH FRONTIER MODELS

Table 57 compares rediscovery performance against Claude Opus 4.5 with 5-shot prompting.

*Table 57.* Frontier model comparison on Literature-91 rediscovery. RLVF-H-C (8B parameters) outperforms Claude Opus 4.5 (5-shot) on both original and extended circuit sets despite substantially fewer parameters.

| Method | Original 50 | | Extended 41 | |
|---|---|---|---|---|
| | Pass@1 | Pass@10 | Pass@1 | Pass@10 |
| Claude Opus 4.5 (0-shot) | 22.0 | 38.0 | 7.3 | 17.1 |
| Claude Opus 4.5 (5-shot) | 34.0 | 54.0 | 12.2 | 26.8 |
| Qwen3-8B SFT | 40.0 | 62.0 | 12.2 | 29.3 |
| Qwen3-8B RLVF-H-C | **58.0** | **78.0** | **24.4** | **41.5** |

RLVF-H-C achieves 24 pp higher Pass@10 than Claude Opus 4.5 (5-shot) on original circuits and 15 pp higher on extended circuits. This demonstrates that domain-specific training with hierarchical verification enables compact models to outperform general-purpose frontier models on specialized reasoning tasks, even when the frontier model has access to few-shot examples demonstrating the target output format.

# G. Reference Materials and Reproducibility Assets

## G.1. Literature-91 Canonical Circuit Set

The Literature-91 evaluation set comprises 50 canonical genetic circuits selected from seminal publications spanning 2000–2024, plus 41 additional circuits that broaden coverage to quorum sensing, optogenetics, and recombinase-based memory. Selection required regulatory mechanisms compatible with our repressor framework, well-characterized behavior in the literature, SBOL-representable parts, and distinct topologies.

Table 58 presents the complete Literature-91 set. For each circuit, we provide the regulatory topology, number of nodes (expression cassettes), key components, and the expected functional behavior used for evaluation. All circuits have been converted to SBOL3 representation and paired with canonical pysbol3 construction code.

### G.1.1. CIRCUIT CATEGORIES

**Toggle Switches (T1–T8).**    Bistable memory circuits based on mutual repression between two regulators, including the canonical LacI/TetR design (Gardner et al., 2000a) and later insulated or feedback-augmented variants.

**Oscillators (O1–O8).**    Negative-feedback circuits that generate sustained oscillations in gene expression, including the original repressilator (Elowitz & Leibler, 2000a) and later optimized variants.

**Feed-Forward Loops (F1–F6).**    Three-node motifs in which a master regulator controls both an intermediate node and the output. We include coherent type-1 and incoherent type-1 variants with their characteristic delay and pulse behaviors (Mangan & Alon, 2003).

**Logic Gates (G1–G20).**    Repressor-based Boolean circuits implementing NOT, AND, OR, NOR, and NAND functions, drawn from classic libraries such as Guet (Guet et al., 2002) and Stanton (Stanton et al., 2014).

**Expression Cassettes (E1–E8).**    Well-characterized single transcription units from sources such as the Anderson promoter collection (Anderson et al., 2006) and the Mutalik RBS library (Mutalik et al., 2013).

**Cascaded Circuits (C1–C6).**    Multi-layer regulatory networks that implement signal processing through sequential gate operations.

**Quorum Sensing (QS1–QS10).**    Circuits implementing population-level behaviors via diffusible autoinducers, including multicellular logic, spatial patterning, synchronized oscillation, and density-dependent control.

**Light-Responsive Circuits (LS1–LS8).**    Optogenetic control systems built around light-responsive signaling cascades, including pDawn/pDusk systems and multiplexed color-sensing designs.

**Cello-Designed Logic (CL1–CL12).**    Automated designs from the Cello CAD tool (Nielsen et al., 2016b), including multi-layer NAND/NOR cascades and larger functions such as multiplexers and priority encoders.

**Advanced Oscillators (AO1–AO5).**    Oscillator architectures that extend the standard repressilator with dual feedback, metabolic coupling, or distributed implementations.

**Layered Cascades and Amplifiers (LC1–LC3).**    Deep regulatory hierarchies used to study signal propagation, amplification, and fault tolerance.

**Memory and State Machines (MS1–MS3).**    Systems implementing sequential logic and long-lived state retention, including recombinase-based memory and event counters.

### G.1.2. MAJOR EXCLUSIONS

The following circuit types are underrepresented or excluded due to fundamental architectural incompatibility: (1) recombinase-based circuits (e.g., use serine integrases for state-dependent DNA rearrangement rather than transcription

*Table 58.* Literature-91 canonical circuit set. Columns indicate circuit identifier, source publication, and circuit name.

| ID | Ref | Name | ID | Ref | Name |
|---|---|---|---|---|---|
| *Toggle Switches (Bistable Memory)* | | | *Cascaded Circuits* | | |
| T1 | (Gardner et al., 2000b) | Gardner Toggle 1 (Bistable) | C1 | (Hooshangi et al., 2005) | Hooshangi 2-layer |
| T2 | (Gardner et al., 2000b) | Gardner Toggle 2 (Bistable) | C2 | (Pedraza & van Oudenaarden, 2005) | Pedraza Noise |
| T3 | (Atkinson et al., 2003) | Atkinson Toggle (Pulse-switch) | C3 | (Rosenfeld et al., 2005) | Rosenfeld Single |
| T4 | (Kobayashi et al., 2004) | Kobayashi Memory | C4 | (Ellis et al., 2009) | Ellis Diverse |
| T5 | (Litcofsky et al., 2012) | Litcofsky Toggle (Bistable) | C5 | (Regot et al., 2011) | Regot Distributed |
| T6 | (Wu et al., 2014) | Wu Ultrasensitive (Bistable) | C6 | (Lou et al., 2012) | Lou Insulated |
| T7 | (Shopera et al., 2017) | Shopera Insulated (Bistable) | *Quorum Sensing Circuits* | | |
| T8 | (Cherry & Adler, 2000) | Cherry Bistable (Bistable) | QS1 | (Tamsir et al., 2011) | Tamsir Multicellular (Dist. NOR) |
| *Oscillators* | | | QS2 | (Basu et al., 2005) | Basu Band-Detection |
| O1 | (Elowitz & Leibler, 2000b) | Repressilator (3-ring osc.) | QS3 | (Danino et al., 2010) | Danino Sync. Osc. |
| O2 | (Elowitz & Leibler, 2000b) | 5-node Osc. (5-ring osc.) | QS4 | (Din et al., 2016a) | Din Lysis |
| O3 | (Stricker et al., 2008) | Stricker Osc. (Dual-FB) | QS5 | (Prindle et al., 2012a) | Prindle Biopixel |
| O4 | (Hasty et al., 2002) | Hasty Relaxation | QS6 | (Swofford et al., 2015) | Swofford Tumor (Density Sensor) |
| O5 | (Potvin-Trottier et al., 2016) | Optimized Repr. (3-ring osc.) | QS7 | (Shong et al., 2013) | Shong Tunable (Orthogonal QS) |
| O6 | (Danino et al., 2010) | Danino Synchronized | QS8 | (You et al., 2004) | You Pop. Control (Density Limit) |
| O7 | (Prindle et al., 2012b) | Prindle Coupled Osc. | QS9 | (Balagaddé et al., 2008) | Balagadde Eco (Predator-Prey) |
| O8 | (Tomazou et al., 2018) | Tomazou Dual-phase | QS10 | (Kobayashi et al., 2004) | Kobayashi QS-Toggle |
| *Feed-Forward Loops* | | | *Light-Responsive Circuits* | | |
| F1 | (Mangan & Alon, 2003) | C1-FFL (Delay) | LS1 | (Levskaya et al., 2005) | Levskaya Camera (Red Light Repr.) |
| F2 | (Mangan & Alon, 2003) | I1-FFL (Pulse) | LS2 | (Tabor et al., 2009) | Tabor Edge Detection |
| F3 | (Basu et al., 2004) | Basu Pulse gen. | LS3 | (Tabor et al., 2011) | Tabor 2-Color Control |
| F4 | (Isaacs et al., 2003) | Isaacs Autoreg. | LS4 | (Ohlendorf et al., 2012) | pDawn/pDusk (Blue Light) |
| F5 | (Entus et al., 2007) | Entus Sensor | LS5 | (Ramakrishnan & Tabor, 2016) | Ramakrishnan UV/Green Switch |
| F6 | (Kaplan et al., 2008) | Kaplan Non-monotonic | LS6 | (Ong et al., 2018) | Ong NIR Sensor |
| *Logic Gates* | | | LS7 | (Schmidl et al., 2014) | Schmidl Optimized 2-Comp System |
| G1 | (Guet et al., 2002) | Guet NOT | LS8 | (Fernandez-Rodriguez et al., 2017) | Fernandez RGB (3-Color Logic) |
| G2 | (Guet et al., 2002) | Guet Buffer | *Cello-Designed Logic (Automated)* | | |
| G3 | (Guet et al., 2002) | Guet AND | CL1-4 | (Nielsen et al., 2016a) | Cello NOT/Buffers (1-Layer Logic) |
| G4 | (Guet et al., 2002) | Guet NAND | CL5-8 | (Nielsen et al., 2016a) | Cello NAND/NOR (2-Layer Logic) |
| G5 | (Guet et al., 2002) | Guet NOR | CL9-10 | (Nielsen et al., 2016a) | Cello Mux/Majority (3-Layer Logic) |
| G6 | (Guet et al., 2002) | Guet OR | CL11-12 | (Nielsen et al., 2016a) | Cello XOR/Cons. (4-Layer Logic) |
| G7 | (Stanton et al., 2014) | Stanton PhlF-NOT | *Advanced Oscillators* | | |
| G8 | (Stanton et al., 2014) | Stanton SrpR-NOT | AO1 | (Stricker et al., 2008) | Stricker Relax. (Robust Osc.) |
| G9 | (Stanton et al., 2014) | Stanton PhlF-NOR | AO2 | (Potvin-Trottier et al., 2016) | Optimized Repr. (Precision Osc.) |
| G10 | (Stanton et al., 2014) | Stanton SrpR-NOR | AO3 | (Fung et al., 2005) | Fung Metabolic Flux Sensor |
| G11 | (Stanton et al., 2014) | Stanton Dual-NOT Cascade | AO4 | (Stricker et al., 2008) | Stricker Dual-FB (Fast Osc.) |
| G12 | (Stanton et al., 2014) | Stanton Orth-AND | AO5 | (Chen et al., 2015) | Chen Consortium (Distrib. Osc.) |
| G13 | (Cox et al., 2007) | Cox Inverter 1 | *Cascades & Memory* | | |
| G14 | (Cox et al., 2007) | Cox Inverter 2 | LC1 | (Pedraza & van Oudenaarden, 2005) | Pedraza Noise Prop. |
| G15 | (Yokobayashi et al., 2002) | Yoko Evolved | LC2 | (Hooshangi et al., 2005) | Hooshangi Multi-stage Cascade |
| G16 | (Wang et al., 2011) | Wang Modular-NOT | LC3 | (Moon et al., 2012) | Moon Layered AND |
| G17 | (Wang et al., 2011) | Wang Modular-NOR | MS1 | (Friedland et al., 2009) | Friedland Counter |
| G18 | (Wang et al., 2011) | Wang Modular-AND | MS2 | (Ham et al., 2006) | Ham Switch (Perm. Memory) |
| G19 | (Brophy & Voigt, 2014) | Brophy Review-NOT | MS3 | (Ham et al., 2008) | Ham Sequential (State Machine) |
| G20 | (Brophy & Voigt, 2014) | Brophy Review-NOR | | | |
| *Expression Cassettes* | | | | | |
| E1 | (Anderson et al., 2006) | J23100 Cassette (Const. strong) | | | |
| E2 | (Anderson et al., 2006) | J23106 Cassette (Const. medium) | | | |
| E3 | (Anderson et al., 2006) | J23117 Cassette (Const. weak) | | | |
| E4 | (Kelly et al., 2009) | Kelly Standard | | | |
| E5 | (Mutalik et al., 2013) | Mutalik B0030 (Strong RBS) | | | |
| E6 | (Mutalik et al., 2013) | Mutalik B0032 (Medium RBS) | | | |
| E7 | (Salis et al., 2009) | Salis Designed | | | |
| E8 | (Davis et al., 2011) | Davis Insulated | | | |

factor regulation), (2) CRISPR/dCas9 circuits (e.g., use catalytically dead Cas9 for transcriptional interference (CRISPRi) or activation (CRISPRa), operating through a distinct mechanism from DNA-binding repressors), (3) analog computation circuits (implement continuous mathematical operations (addition, subtraction, logarithms) rather than discrete Boolean logic), (4) RNA-based circuits (use riboswitches, toehold switches, or RNA-RNA interactions for regulation rather than transcription factors).

### G.2. Evaluation Parts Library Extensions

Evaluation circuits require additional biological parts beyond the original 48-part library. Table 59 specifies these required extensions, categorized by functional role. These parts are represented in Table 59.

The evaluation parts library is further extended to incorporate additional parts to simulate the dense search spaces and component diversity encountered in real-world automated design tasks. We summarize these parts below, Which were all drawn from well-characterized collections in the iGEM Registry:

**Extended Constitutive Promoters.** The Anderson promoter collection (BBa_J23100–J23119) provides 20 well-characterized constitutive promoters spanning a 1000-fold expression range in *E. coli* (Anderson et al., 2006). These promoters share a common scaffold derived from the consensus $\sigma^{70}$ promoter sequence, with combinatorial mutations in the $-35$ and $-10$ boxes yielding predictable strength gradations. Our evaluation library includes 12 Anderson promoters not present in the training set: J23101, J23102, J23103, J23104, J23107, J23108, J23109, J23110, J23111, J23112, J23113, and J23119. J23100, J23102, J23104, and J23119 forming a strong cluster and J23114, J23117, J23118 forming a weak cluster, enabling systematic evaluation of expression-level sensitivity in circuit designs.

**Alternative Small-Molecule Inducible Systems.** Beyond the arabinose (pBad/AraC) and IPTG (pLac/LacI) systems in the training library, the evaluation set includes orthogonal inducible promoters: the rhamnose-responsive pRha promoter (BBa_K914003), which exhibits tight repression and gradual dose-response characteristics suitable for fine-tuned expression control; the cumate-inducible pCym system, which uses the non-toxic inducer p-isopropyl benzoate; and the vanillate-inducible PV10 promoter characterized in *Methylorubrum extorquens*.

**Chromoprotein Reporter Collection.** The chromoprotein collection provides visible-light reporters that complement fluorescent proteins in the training set. We include 14 chromoproteins: amilCP (blue, 588nm absorption, *Acropora millepora*), amilGFP (yellow, 458nm, fast-maturing), cjBlue (dark green, *Cnidopus japonicus*, slower maturation), spisPink (pink), eforRed, asPink, scOrange, fwYellow, amajLime, aeBlue, tsPurple, gfasPurple, meffRed, and meffBlue. These reporters enable instrument-free detection under ambient lighting and test whether models correctly handle reporter substitutions that preserve circuit topology while changing output modality.

**Extended Terminator Collection.** The evaluation library includes terminators characterized in Chen et al. (Chen et al., 2013), which quantified termination efficiency for 582 natural and synthetic terminators. We add ECK120029600 (the strongest natural terminator identified, with >99% termination efficiency), the L3S2P21 series of synthetic terminators, and the rrnB T1 terminator. These parts test whether models can substitute terminators of varying strength while maintaining proper transcriptional insulation.

**Extended TetR-Family Repressors.** Beyond the 10 repressor-promoter pairs in the training/held-out split, the evaluation library includes additional characterized TetR homologs: HlyIIR, IcaRA, LitR, PsrA, LmrA, McbR, ButR, and TarA (Stanton et al., 2014). These repressors were characterized for orthogonality in both *E. coli* and mammalian cells, providing parts for cross-organism generalization evaluation. The expanded repressor set enables construction of larger, deeper cascaded circuits that exceed the complexity of the training distribution.

### G.3. Example Circuits

#### G.3.1. CANONICAL CODE FOR EXPRESSION CASSETTE

The following implementation of a canonical expression cassette illustrates the standard construction pattern used throughout the benchmark: namespace configuration, component creation, document assembly, constraint specification, and interaction definition.

*Listing 1.* PySBOL Example

```python
import sbol3
from sbol_helpers import (
    SO_PROMOTER,
    SO_RBS,
    SO_CDS,
    SO_TERMINATOR,
    SO_ENGINEERED_REGION
)

sbol3.set_namespace('https://synthia.org/')
doc = sbol3.Document()

# Create components
promoter = sbol3.Component('J23100_promoter', types=[sbol3.SBO_DNA])
promoter.roles = [SO_PROMOTER]
promoter.name = 'J23100'
doc.add(promoter)

rbs = sbol3.Component('B0032_rbs', types=[sbol3.SBO_DNA])
rbs.roles = [SO_RBS]
rbs.name = 'B0032'
doc.add(rbs)

cds = sbol3.Component('E0040_gfp', types=[sbol3.SBO_DNA])
cds.roles = [SO_CDS]
cds.name = 'GFP'
doc.add(cds)

terminator = sbol3.Component('B0015_term', types=[sbol3.SBO_DNA])
terminator.roles = [SO_TERMINATOR]
terminator.name = 'B0015'
doc.add(terminator)

# Create cassette and assembly
cassette = sbol3.Component('gfp_cassette', types=[sbol3.SBO_DNA])
cassette.roles = [SO_ENGINEERED_REGION]

sc_p = sbol3.SubComponent(promoter)
sc_r = sbol3.SubComponent(rbs)
sc_c = sbol3.SubComponent(cds)
sc_t = sbol3.SubComponent(terminator)
cassette.features = [sc_p, sc_r, sc_c, sc_t]

# Establish ordering constraints
cassette.constraints = [
    sbol3.Constraint(sbol3.SBOL_PRECEDES, sc_p, sc_r),
    sbol3.Constraint(sbol3.SBOL_PRECEDES, sc_r, sc_c),
    sbol3.Constraint(sbol3.SBOL_PRECEDES, sc_c, sc_t)
]

doc.add(cassette)
```

### G.3.2. CANONICAL CODE FOR REGULATED EXPRESSION CIRCUIT

The following implementation demonstrates a regulated expression circuit featuring an inducible promoter system. The circuit consists of a TetR-repressible promoter (pTetR), an operator region for transcription factor binding, a ribosome binding site for translation initiation, and a GFP reporter coding sequence. This pattern illustrates how environmental signals can modulate gene expression through repressor-operator interactions, a common motif in synthetic circuit design.

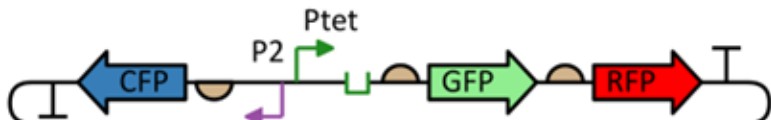

*Figure 6.* Schematic of a regulated genetic circuit (Poole et al., 2022). The pTetR promoter controls expression of downstream coding sequences through operator-mediated regulation.

*Listing 2.* Regulated Expression Circuit in PySBOL

```
import sbol3
from sbol_helpers import (
    SO_PROMOTER,
    SO_OPERATOR,
    SO_RBS,
    SO_CDS,
    SO_ENGINEERED_REGION
)

sbol3.set_namespace('https://synthia.org/')
doc = sbol3.Document()

# Create components
promoter = sbol3.Component('pTetR_promoter', types=[sbol3.SBO_DNA])
promoter.roles = [SO_PROMOTER]
promoter.name = 'pTetR'
promoter.description = 'TetR■repressible■promoter'
doc.add(promoter)

operator = sbol3.Component('tetO_operator', types=[sbol3.SBO_DNA])
operator.roles = [SO_OPERATOR]
operator.name = 'tetO'
doc.add(operator)

rbs = sbol3.Component('UTR1_rbs', types=[sbol3.SBO_DNA])
rbs.roles = [SO_RBS]
rbs.name = 'UTR1'
doc.add(rbs)

cds = sbol3.Component('GFP_cds', types=[sbol3.SBO_DNA])
cds.roles = [SO_CDS]
cds.name = 'GFP'
doc.add(cds)

# Create circuit and assembly
circuit = sbol3.Component('regulated_circuit', types=[sbol3.SBO_DNA])
circuit.roles = [SO_ENGINEERED_REGION]

sc_p = sbol3.SubComponent(promoter)
sc_o = sbol3.SubComponent(operator)
sc_r = sbol3.SubComponent(rbs)
sc_c = sbol3.SubComponent(cds)
circuit.features = [sc_p, sc_o, sc_r, sc_c]

# Establish ordering constraints
circuit.constraints = [
    sbol3.Constraint(sbol3.SBOL_PRECEDES, sc_p, sc_o),
    sbol3.Constraint(sbol3.SBOL_PRECEDES, sc_o, sc_r),
    sbol3.Constraint(sbol3.SBOL_PRECEDES, sc_r, sc_c)
]

doc.add(circuit)
```

### G.3.3. CANONICAL CODE FOR GENETIC MULTIPLEXER

The following implementation demonstrates a 2:1 genetic multiplexer that selects between two input signals based on a selector signal. The circuit implements the Boolean function $Y = (\overline{S} \wedge D_0) \vee (S \wedge D_1)$, where $D_0$ and $D_1$ are data inputs, $S$ is the selector, and $Y$ is the output. Input channel 0 uses an IPTG-inducible promoter (pLac), input channel 1 uses an arabinose-inducible promoter (pAra), and the selector is controlled by aTc via TetR repression.

The selector signal (S) determines whether input $D_0$ or $D_1$ propagates to the GFP output. When the selector is inactive (low aTc), TetR production is suppressed, relieving repression of Channel 0 and allowing IPTG-induced signal to pass. When the selector is active (high aTc), TetR represses both channel operators, but Channel 1 receives stronger arabinose induction that overcomes the gating.

In the implementation, the multiplexer is constructed from four distinct sub-modules (two input channels, selector module, output module) each encapsulated as an SBOL Component with its own internal structure. This modularity enables RL agents to learn reusable design patterns and compose them into higher-order functional units. The code explicitly models TetR-mediated repression of both input channel operators through SBOL Interaction objects with typed participation roles.

The 2:1 multiplexer serves as a canonical building block that can be recursively composed to construct larger $2^n$:1 multiplexers by adding input channels and selector bits. This recursive structure provides a natural curriculum for RL training, where agents can transfer learned policies from simpler circuits to more complex compositions.

*Listing 3.* Genetic Multiplexer in PySBOL

```
import sbol3
from sbol_helpers import (
    SO_PROMOTER,
```

```
        SO_OPERATOR,
        SO_RBS,
        SO_CDS,
        SO_TERMINATOR,
        SO_ENGINEERED_REGION,
        SBO_INHIBITION,
        SBO_GENETIC_PRODUCTION
)

sbol3.set_namespace('https://synthia.org/')
doc = sbol3.Document()

# === Input Channel 0 (D0): IPTG-inducible ===
pLac = sbol3.Component('pLac_promoter', types=[sbol3.SBO_DNA])
pLac.roles = [SO_PROMOTER]
pLac.name = 'pLac'
pLac.description = 'IPTG-inducible■promoter'
doc.add(pLac)

tetO1 = sbol3.Component('tetO1_operator', types=[sbol3.SBO_DNA])
tetO1.roles = [SO_OPERATOR]
tetO1.name = 'tetO1'
tetO1.description = 'TetR■operator■for■channel■gating'
doc.add(tetO1)

rbs_d0 = sbol3.Component('RBS_D0', types=[sbol3.SBO_DNA])
rbs_d0.roles = [SO_RBS]
rbs_d0.name = 'RBS_D0'
doc.add(rbs_d0)

signal0 = sbol3.Component('Signal0_cds', types=[sbol3.SBO_DNA])
signal0.roles = [SO_CDS]
signal0.name = 'Signal0'
signal0.description = 'Channel■0■signaling■protein'
doc.add(signal0)

# === Input Channel 1 (D1): Arabinose-inducible ===
pAra = sbol3.Component('pAra_promoter', types=[sbol3.SBO_DNA])
pAra.roles = [SO_PROMOTER]
pAra.name = 'pAra'
pAra.description = 'Arabinose-inducible■promoter'
doc.add(pAra)

tetO2 = sbol3.Component('tetO2_operator', types=[sbol3.SBO_DNA])
tetO2.roles = [SO_OPERATOR]
tetO2.name = 'tetO2'
tetO2.description = 'TetR■operator■for■channel■gating'
doc.add(tetO2)

rbs_d1 = sbol3.Component('RBS_D1', types=[sbol3.SBO_DNA])
rbs_d1.roles = [SO_RBS]
rbs_d1.name = 'RBS_D1'
doc.add(rbs_d1)

signal1 = sbol3.Component('Signal1_cds', types=[sbol3.SBO_DNA])
signal1.roles = [SO_CDS]
signal1.name = 'Signal1'
signal1.description = 'Channel■1■signaling■protein'
doc.add(signal1)

# === Selector Module (S): aTc-controlled ===
pTet = sbol3.Component('pTet_promoter', types=[sbol3.SBO_DNA])
pTet.roles = [SO_PROMOTER]
pTet.name = 'pTet'
pTet.description = 'aTc-inducible■promoter'
doc.add(pTet)

rbs_sel = sbol3.Component('RBS_Sel', types=[sbol3.SBO_DNA])
rbs_sel.roles = [SO_RBS]
rbs_sel.name = 'RBS_Sel'
doc.add(rbs_sel)

tetR = sbol3.Component('TetR_cds', types=[sbol3.SBO_DNA])
tetR.roles = [SO_CDS]
tetR.name = 'TetR'
tetR.description = 'Tetracycline■repressor'
doc.add(tetR)

tetR_protein = sbol3.Component('TetR_protein', types=[sbol3.SBO_PROTEIN])
tetR_protein.name = 'TetR'
doc.add(tetR_protein)

# === Output Module ===
pOut = sbol3.Component('pOut_promoter', types=[sbol3.SBO_DNA])
pOut.roles = [SO_PROMOTER]
pOut.name = 'pOut'
pOut.description = 'Output■integrating■promoter'
doc.add(pOut)

rbs_out = sbol3.Component('RBS_Out', types=[sbol3.SBO_DNA])
rbs_out.roles = [SO_RBS]
rbs_out.name = 'RBS_Out'
```

```
doc.add(rbs_out)

gfp = sbol3.Component('GFP_cds', types=[sbol3.SBO_DNA])
gfp.roles = [SO_CDS]
gfp.name = 'GFP'
doc.add(gfp)

term = sbol3.Component('B0015_term', types=[sbol3.SBO_DNA])
term.roles = [SO_TERMINATOR]
term.name = 'B0015'
doc.add(term)

# === Assemble Input Channel 0 ===
channel0 = sbol3.Component('channel0', types=[sbol3.SBO_DNA])
channel0.roles = [SO_ENGINEERED_REGION]
channel0.name = 'Input_Channel_0'

sc_pLac = sbol3.SubComponent(pLac)
sc_tetO1 = sbol3.SubComponent(tetO1)
sc_rbs_d0 = sbol3.SubComponent(rbs_d0)
sc_signal0 = sbol3.SubComponent(signal0)
channel0.features = [sc_pLac, sc_tetO1, sc_rbs_d0, sc_signal0]

channel0.constraints = [
    sbol3.Constraint(sbol3.SBOL_PRECEDES, sc_pLac, sc_tetO1),
    sbol3.Constraint(sbol3.SBOL_PRECEDES, sc_tetO1, sc_rbs_d0),
    sbol3.Constraint(sbol3.SBOL_PRECEDES, sc_rbs_d0, sc_signal0)
]
doc.add(channel0)

# === Assemble Input Channel 1 ===
channel1 = sbol3.Component('channel1', types=[sbol3.SBO_DNA])
channel1.roles = [SO_ENGINEERED_REGION]
channel1.name = 'Input_Channel_1'

sc_pAra = sbol3.SubComponent(pAra)
sc_tetO2 = sbol3.SubComponent(tetO2)
sc_rbs_d1 = sbol3.SubComponent(rbs_d1)
sc_signal1 = sbol3.SubComponent(signal1)
channel1.features = [sc_pAra, sc_tetO2, sc_rbs_d1, sc_signal1]

channel1.constraints = [
    sbol3.Constraint(sbol3.SBOL_PRECEDES, sc_pAra, sc_tetO2),
    sbol3.Constraint(sbol3.SBOL_PRECEDES, sc_tetO2, sc_rbs_d1),
    sbol3.Constraint(sbol3.SBOL_PRECEDES, sc_rbs_d1, sc_signal1)
]
doc.add(channel1)

# === Assemble Selector Module ===
selector = sbol3.Component('selector', types=[sbol3.SBO_DNA])
selector.roles = [SO_ENGINEERED_REGION]
selector.name = 'Selector_Module'

sc_pTet = sbol3.SubComponent(pTet)
sc_rbs_sel = sbol3.SubComponent(rbs_sel)
sc_tetR = sbol3.SubComponent(tetR)
selector.features = [sc_pTet, sc_rbs_sel, sc_tetR]

selector.constraints = [
    sbol3.Constraint(sbol3.SBOL_PRECEDES, sc_pTet, sc_rbs_sel),
    sbol3.Constraint(sbol3.SBOL_PRECEDES, sc_rbs_sel, sc_tetR)
]
doc.add(selector)

# === Assemble Output Module ===
output = sbol3.Component('output', types=[sbol3.SBO_DNA])
output.roles = [SO_ENGINEERED_REGION]
output.name = 'Output_Module'

sc_pOut = sbol3.SubComponent(pOut)
sc_rbs_out = sbol3.SubComponent(rbs_out)
sc_gfp = sbol3.SubComponent(gfp)
sc_term = sbol3.SubComponent(term)
output.features = [sc_pOut, sc_rbs_out, sc_gfp, sc_term]

output.constraints = [
    sbol3.Constraint(sbol3.SBOL_PRECEDES, sc_pOut, sc_rbs_out),
    sbol3.Constraint(sbol3.SBOL_PRECEDES, sc_rbs_out, sc_gfp),
    sbol3.Constraint(sbol3.SBOL_PRECEDES, sc_gfp, sc_term)
]
doc.add(output)

# === Assemble Complete Multiplexer (Hierarchical Composition) ===
multiplexer = sbol3.Component('multiplexer', types=[sbol3.SBO_DNA])
multiplexer.roles = [SO_ENGINEERED_REGION]
multiplexer.name = 'Genetic_2to1_MUX'
multiplexer.description = '2:1 genetic multiplexer circuit'

sc_ch0 = sbol3.SubComponent(channel0)
sc_ch1 = sbol3.SubComponent(channel1)
sc_sel = sbol3.SubComponent(selector)
sc_out = sbol3.SubComponent(output)
```

```
multiplexer.features = [sc_ch0, sc_ch1, sc_sel, sc_out]
doc.add(multiplexer)

# === Define Regulatory Interactions ===
# TetR represses both input channels via operator binding
repression_ch0 = sbol3.Interaction(
    'TetR_represses_channel0',
    types=[SBO_INHIBITION]
)
repression_ch0.participations = [
    sbol3.Participation([sbol3.SBO_INHIBITOR], tetR_protein),
    sbol3.Participation([sbol3.SBO_INHIBITED], tetO1)
]
multiplexer.interactions.append(repression_ch0)

repression_ch1 = sbol3.Interaction(
    'TetR_represses_channel1',
    types=[SBO_INHIBITION]
)
repression_ch1.participations = [
    sbol3.Participation([sbol3.SBO_INHIBITOR], tetR_protein),
    sbol3.Participation([sbol3.SBO_INHIBITED], tetO2)
]
multiplexer.interactions.append(repression_ch1)

# TetR production from selector module
tetR_production = sbol3.Interaction(
    'TetR_production',
    types=[SBO_GENETIC_PRODUCTION]
)
tetR_production.participations = [
    sbol3.Participation([sbol3.SBO_TEMPLATE], tetR),
    sbol3.Participation([sbol3.SBO_PRODUCT], tetR_protein)
]
multiplexer.interactions.append(tetR_production)
```

## G.4. Prompt Templates

All tasks share a common prompt structure consisting of four components: (1) a *system preamble* establishing the model's role and providing a condensed pysbol3 API reference with the parts library; (2) a *task instruction* specifying the expected action; (3) *circuit context* providing task-specific inputs (buggy code, natural language descriptions, or functional specifications); and (4) an *output format directive* constraining the response to executable pysbol3 code enclosed in a fenced code block.

The system preamble is shared across all tasks and includes a summary of the pysbol3 API surface (component creation, subcomponent assembly, constraint specification, interaction definition), the available parts library (Table 9), and relevant ontology constants (Sequence Ontology terms for part types, Systems Biology Ontology terms for interaction types). The full preamble is approximately 1,200 tokens; we omit it from the per-task examples below and reproduce it separately in Appendix G.4.5.

Table 60 summarizes the prompt structure across tasks. We provide one complete example per curriculum stage, covering T1 (code fundamentals), T4 (translation), T6 (functional reasoning), and T7 (design).

### G.4.1. T1: CODE REPAIR (STAGE 1—CODE FUNDAMENTALS)

The following shows a T1 prompt for a NOT gate containing a Level 2 structural error (incorrect variable reference in a constraint). The system preamble is omitted.

---

**T1 Prompt: Code Repair (NOT Gate, Level 2 Flaw)**

**Task:** The following pysbol3 code contains one or more errors that prevent correct execution or SBOL validation. Identify each error, explain the cause, and provide corrected code that executes successfully and produces a valid SBOL document.
**Buggy Code:**

```python
import sbol3
from sbol_helpers import (
    SO_PROMOTER, SO_RBS, SO_CDS, SO_TERMINATOR,
    SO_ENGINEERED_REGION, SBO_INHIBITION,
    SBO_GENETIC_PRODUCTION
)

sbol3.set_namespace('https://synthia.org/')
doc = sbol3.Document()

# Input cassette: constitutive TetR expression
```

---

```
p_in = sbol3.Component('J23100_promoter',
    types=[sbol3.SBO_DNA])
p_in.roles = [SO_PROMOTER]
doc.add(p_in)
rbs_in = sbol3.Component('B0034_rbs', types=[sbol3.SBO_DNA])
rbs_in.roles = [SO_RBS]
doc.add(rbs_in)
tetR = sbol3.Component('TetR_cds', types=[sbol3.SBO_DNA])
tetR.roles = [SO_CDS]
doc.add(tetR)
term_in = sbol3.Component('B0015_term1',
    types=[sbol3.SBO_DNA])
term_in.roles = [SO_TERMINATOR]
doc.add(term_in)

# Output cassette: TetR-repressed GFP
p_out = sbol3.Component('pTetR_promoter',
    types=[sbol3.SBO_DNA])
p_out.roles = [SO_PROMOTER]
doc.add(p_out)
rbs_out = sbol3.Component('B0032_rbs',
    types=[sbol3.SBO_DNA])
rbs_out.roles = [SO_RBS]
doc.add(rbs_out)
gfp = sbol3.Component('GFP_cds', types=[sbol3.SBO_DNA])
gfp.roles = [SO_CDS]
doc.add(gfp)
term_out = sbol3.Component('B0015_term2',
    types=[sbol3.SBO_DNA])
term_out.roles = [SO_TERMINATOR]
doc.add(term_out)

# Assemble input cassette
input_cas = sbol3.Component('input_cassette',
    types=[sbol3.SBO_DNA])
input_cas.roles = [SO_ENGINEERED_REGION]
sc_p_in = sbol3.SubComponent(p_in)
sc_rbs_in = sbol3.SubComponent(rbs_in)
sc_tetR = sbol3.SubComponent(tetR)
sc_term_in = sbol3.SubComponent(term_in)
input_cas.features = [
    sc_p_in, sc_rbs_in, sc_tetR, sc_term_in]
input_cas.constraints = [
    sbol3.Constraint(
        sbol3.SBOL_PRECEDES, sc_p_in, sc_rbs_in),
    sbol3.Constraint(
        sbol3.SBOL_PRECEDES, sc_rbs_in, sc_tetR),
    sbol3.Constraint(
        sbol3.SBOL_PRECEDES, sc_tetR, sc_term_in)
]
doc.add(input_cas)

# Assemble output cassette
output_cas = sbol3.Component('output_cassette',
    types=[sbol3.SBO_DNA])
output_cas.roles = [SO_ENGINEERED_REGION]
sc_p_out = sbol3.SubComponent(p_out)
sc_rbs_out = sbol3.SubComponent(rbs_out)
sc_gfp = sbol3.SubComponent(gfp)
sc_term_out = sbol3.SubComponent(term_out)
output_cas.features = [
    sc_p_out, sc_rbs_out, sc_gfp, sc_term_out]
output_cas.constraints = [
    sbol3.Constraint(
        sbol3.SBOL_PRECEDES, sc_p_out, sc_rbs_out),
    sbol3.Constraint(
        sbol3.SBOL_PRECEDES, sc_rbs_out, sc_gfp),
    sbol3.Constraint(
        sbol3.SBOL_PRECEDES, sc_gfp, sc_term_in)
    # ERROR: sc_term_in belongs to input_cassette,
    # not output_cassette
]
doc.add(output_cas)
```

### Error Output:

```
sbol3.ValidationError: SubComponent referenced in
Constraint of 'output_cassette' is not a feature
of 'output_cassette'.
```

> **Instructions:** Provide corrected code in a single fenced code block. The code must execute without errors and pass SBOL validation.

## G.4.2. T4: NATURAL LANGUAGE TO CODE (STAGE 2—TRANSLATION)

The following shows a T4 prompt for a toggle switch. Natural language descriptions are generated procedurally with controlled vocabulary to ensure unambiguous specification while varying syntactic structure to prevent overfitting to fixed phrasings.

---

**T4 Prompt: Natural Language to Code (Toggle Switch)**

**Task:** Generate complete, executable pysbol3 code that constructs the genetic circuit described below. The code must produce a valid SBOL3 document. Use only parts from the provided library.

**Circuit Description:**
Design a toggle switch with two stable states using mutual repression between LacI and TetR. The circuit requires two expression cassettes:

- **Cassette A:** A TetR-repressible promoter (pTetR) drives expression of the LacI repressor through a medium-strength RBS (B0032). Include terminator B0015.

- **Cassette B:** A LacI-repressible promoter (pLac) drives expression of the TetR repressor through a medium-strength RBS (B0034). Include terminator B0015.

The mutual repression creates bistability: LacI represses TetR production (via pLac), while TetR represses LacI production (via pTetR). The circuit should include both repressor proteins as separate protein-type components, and two `Interaction` objects modeling the repression relationships with SBO_INHIBITION type and appropriate SBO_INHIBITOR/SBO_INHIBITED participation roles.

**Instructions:** Provide complete pysbol3 code in a single fenced code block. The code must set the namespace, create a Document, define all components, assemble cassettes with ordering constraints, and define both repression interactions.

---

## G.4.3. T6: CIRCUIT DEBUGGING (STAGE 3—FUNCTIONAL REASONING)

The following shows a T6 prompt for a toggle switch with a Level 4 flaw (incomplete feedback loop destroying bistability). Unlike T1, the code executes and validates; the error manifests only through incorrect circuit behavior.

---

**T6 Prompt: Circuit Debugging (Toggle Switch, Level 4 Flaw)**

**Task:** The following pysbol3 code executes successfully and produces a valid SBOL document, but the resulting circuit does not behave as intended. Given the observed malfunction, identify the design flaw, explain why it causes the observed behavior, and provide corrected code.

**Circuit Code:**

```python
import sbol3
from sbol_helpers import (
    SO_PROMOTER, SO_RBS, SO_CDS, SO_TERMINATOR,
    SO_ENGINEERED_REGION, SBO_INHIBITION,
    SBO_GENETIC_PRODUCTION
)

sbol3.set_namespace('https://synthia.org/')
doc = sbol3.Document()

# --- Parts ---
pTetR = sbol3.Component('pTetR', types=[sbol3.SBO_DNA])
pTetR.roles = [SO_PROMOTER]
doc.add(pTetR)
pLac = sbol3.Component('pLac', types=[sbol3.SBO_DNA])
pLac.roles = [SO_PROMOTER]
doc.add(pLac)
rbs1 = sbol3.Component('B0032_rbs1',
    types=[sbol3.SBO_DNA])
rbs1.roles = [SO_RBS]
doc.add(rbs1)
rbs2 = sbol3.Component('B0034_rbs2',
    types=[sbol3.SBO_DNA])
rbs2.roles = [SO_RBS]
```

---

```
doc.add(rbs2)
lacI_cds = sbol3.Component('LacI_cds',
    types=[sbol3.SBO_DNA])
lacI_cds.roles = [SO_CDS]
doc.add(lacI_cds)
tetR_cds = sbol3.Component('TetR_cds',
    types=[sbol3.SBO_DNA])
tetR_cds.roles = [SO_CDS]
doc.add(tetR_cds)
term1 = sbol3.Component('B0015_t1',
    types=[sbol3.SBO_DNA])
term1.roles = [SO_TERMINATOR]
doc.add(term1)
term2 = sbol3.Component('B0015_t2',
    types=[sbol3.SBO_DNA])
term2.roles = [SO_TERMINATOR]
doc.add(term2)

# --- Proteins ---
lacI_prot = sbol3.Component('LacI_protein',
    types=[sbol3.SBO_PROTEIN])
doc.add(lacI_prot)
tetR_prot = sbol3.Component('TetR_protein',
    types=[sbol3.SBO_PROTEIN])
doc.add(tetR_prot)

# --- Cassette A: pTetR -> LacI ---
casA = sbol3.Component('cassette_A',
    types=[sbol3.SBO_DNA])
casA.roles = [SO_ENGINEERED_REGION]
sc_pTetR = sbol3.SubComponent(pTetR)
sc_rbs1 = sbol3.SubComponent(rbs1)
sc_lacI = sbol3.SubComponent(lacI_cds)
sc_t1 = sbol3.SubComponent(term1)
casA.features = [sc_pTetR, sc_rbs1, sc_lacI, sc_t1]
casA.constraints = [
    sbol3.Constraint(
        sbol3.SBOL_PRECEDES, sc_pTetR, sc_rbs1),
    sbol3.Constraint(
        sbol3.SBOL_PRECEDES, sc_rbs1, sc_lacI),
    sbol3.Constraint(
        sbol3.SBOL_PRECEDES, sc_lacI, sc_t1)
]
doc.add(casA)

# --- Cassette B: pLac -> TetR ---
casB = sbol3.Component('cassette_B',
    types=[sbol3.SBO_DNA])
casB.roles = [SO_ENGINEERED_REGION]
sc_pLac = sbol3.SubComponent(pLac)
sc_rbs2 = sbol3.SubComponent(rbs2)
sc_tetR = sbol3.SubComponent(tetR_cds)
sc_t2 = sbol3.SubComponent(term2)
casB.features = [sc_pLac, sc_rbs2, sc_tetR, sc_t2]
casB.constraints = [
    sbol3.Constraint(
        sbol3.SBOL_PRECEDES, sc_pLac, sc_rbs2),
    sbol3.Constraint(
        sbol3.SBOL_PRECEDES, sc_rbs2, sc_tetR),
    sbol3.Constraint(
        sbol3.SBOL_PRECEDES, sc_tetR, sc_t2)
]
doc.add(casB)

# --- Top-level toggle switch ---
toggle = sbol3.Component('toggle_switch',
    types=[sbol3.SBO_DNA])
toggle.roles = [SO_ENGINEERED_REGION]
sc_casA = sbol3.SubComponent(casA)
sc_casB = sbol3.SubComponent(casB)
toggle.features = [sc_casA, sc_casB]
doc.add(toggle)

# --- Interactions ---
prod_lacI = sbol3.Interaction('prod_lacI',
    types=[SBO_GENETIC_PRODUCTION])
prod_lacI.participations = [
    sbol3.Participation([sbol3.SBO_TEMPLATE], sc_lacI),
```

```
        sbol3.Participation([sbol3.SBO_PRODUCT], lacI_prot)
]
toggle.interactions.append(prod_lacI)

prod_tetR = sbol3.Interaction('prod_tetR',
    types=[SBO_GENETIC_PRODUCTION])
prod_tetR.participations = [
    sbol3.Participation([sbol3.SBO_TEMPLATE], sc_tetR),
    sbol3.Participation([sbol3.SBO_PRODUCT], tetR_prot)
]
toggle.interactions.append(prod_tetR)

# TetR represses pTetR (one arm of mutual repression)
repr_tetR = sbol3.Interaction('repr_tetR',
    types=[SBO_INHIBITION])
repr_tetR.participations = [
    sbol3.Participation(
        [sbol3.SBO_INHIBITOR], tetR_prot),
    sbol3.Participation(
        [sbol3.SBO_INHIBITED], sc_pTetR)
]
toggle.interactions.append(repr_tetR)

# FLAW: The second repression (LacI -| pLac) is
# absent. Only one arm of the mutual repression
# loop is defined.
```

**Observed Malfunction:** The toggle switch fails to exhibit bistability. Instead of maintaining two stable states, the circuit always converges to a single state where TetR is highly expressed and LacI is suppressed, regardless of initial inducer conditions. Adding IPTG (which should relieve LacI repression of pLac and flip the switch) has no effect on the steady-state output.

**Instructions:** (1) Identify the design flaw and its location. (2) Explain why this flaw produces the observed single-state behavior instead of bistability. (3) Provide corrected code in a single fenced code block.

### G.4.4. T7: DE NOVO DESIGN (STAGE 4—DESIGN)

The following shows a T7 prompt for a two-input NOR gate. Design prompts provide only a functional specification and parts library reference; the model must determine circuit topology, select parts, and produce complete construction code.

**T7 Prompt: De Novo Design (Two-Input NOR Gate)**

**Task:** Design a genetic circuit that implements the functional specification below. Generate complete, executable pysbol3 code using only parts from the provided library. The code must produce a valid SBOL3 document with correct regulatory interactions.

**Functional Specification:**

- **Circuit type:** Two-input NOR gate

- **Logic function:** $Y = \overline{A + B}$. Output Y is ON only when both inputs A and B are OFF.

- **Truth table:**

| A | B | Y |
|---|---|---|
| 0 | 0 | 1 |
| 0 | 1 | 0 |
| 1 | 0 | 0 |
| 1 | 1 | 0 |

- **Implementation constraints:**

    - Input A: controlled by the LacI repressor (IPTG-inducible). When IPTG is present, LacI is inactive and input A is ON.
    - Input B: controlled by the TetR repressor (aTc-inducible). When aTc is present, TetR is inactive and input B is ON.
    - Output: GFP reporter expression.
    - Both repressors must independently inhibit the output promoter; any active repressor blocks output.

- **Required:** Input cassette(s) for repressor production, output cassette with GFP reporter, all regulatory interactions modeled as SBOL Interaction objects with typed participation roles.

**Instructions:** Provide complete pysbol3 code in a single fenced code block. Include namespace setup, Document creation, all component definitions with correct ontology annotations, cassette assembly with ordering constraints, and all interaction definitions (genetic production and repression).

**Remarks on prompt design.** T1 and T6 both present code with errors, but differ in the diagnostic signal: T1 provides explicit error messages (Python tracebacks or SBOL validation errors), whereas T6 provides only a behavioral symptom phrased as an experimental observation. T4 descriptions use controlled vocabulary (specific part names, explicit interaction types) rather than free-form prose. T7 specifications include both the Boolean function formula and a truth table, providing redundant specification. T2 and T3 prompts follow analogous structures (partial code with `# TODO` markers for T2; working code with a modification instruction for T3) and are omitted for brevity.

G.4.5. FRONTIER MODEL EVALUATION PROMPTS

We evaluate Claude Opus 4.5 (with extended thinking) in zero-shot and 5-shot configurations. Both share an identical system preamble; the 5-shot setting prepends one solved example for each of five circuit types before the test prompt. Temperature is 0 for TSR evaluation and 0.7 for Pass@$k$ estimation, matching the settings used for trained models.

**System Preamble.** The shared system preamble provides approximately 1,200 tokens of task and API context.

---

**System Preamble (Core)**

You are an expert synthetic biology engineer. Your task is to generate executable Python code using the `pysbol3` library to construct genetic circuits in SBOL3 format.
**pysbol3 API Reference (Summary):**

- `sbol3.set_namespace(uri)` — Set the default URI namespace.

- `sbol3.Document()` — Create a new SBOL document.

- `sbol3.Component(id, types=[sbol3.SBO_DNA])` — Create a biological component. Set `.roles` to the appropriate Sequence Ontology term.

- `sbol3.SubComponent(component)` — Reference to a component for inclusion in a parent's `.features` list.

- `sbol3.Constraint(sbol3.SBOL_PRECEDES, subject, object)` — Ordering between SubComponents in the same parent.

- `sbol3.Interaction(id, types=[...])` — Regulatory interaction. Set `.participations` to `sbol3.Participation` objects with SBO roles.

- `doc.add(obj)` — Register a top-level object.

**Ontology Constants** (from `sbol_helpers`):

- Part types: SO_PROMOTER, SO_RBS, SO_CDS, SO_TERMINATOR, SO_OPERATOR, SO_ENGINEERED_REGION

- Interaction types: SBO_INHIBITION, SBO_STIMULATION, SBO_GENETIC_PRODUCTION

- Participation roles: SBO_INHIBITOR, SBO_INHIBITED, SBO_STIMULATOR, SBO_STIMULATED, SBO_TEMPLATE, SBO_PRODUCT

**Parts Library:** 48 characterized parts: 17 promoters (5 constitutive, 1 inducible, 11 repressible), 5 RBS, 8 CDS (5 repressors, 1 activator, 2 reporters), 3 terminators, and associated protein components. [Full listing in context.]
**Output:** Provide only executable Python code in a fenced code block. Begin with `import sbol3`, use namespace `https://synthia.org/`, and include all components, assembly, constraints, and interactions.

---

**Zero-Shot Format.** The test prompt follows the system preamble directly with no examples. The prompt structure matches the task-specific templates from Appendices G.4.1–G.4.4.

**Five-Shot Format.** We prepend one (prompt, canonical solution) pair for each of five circuit types: expression cassette, NOT gate, incoherent feed-forward loop, toggle switch, and repressilator. Examples are selected from the procedural training set subject to four constraints: no graph-isomorphic overlap with any test circuit, distinct circuit types, training-tier parts only, and a spread of complexity from cassette to oscillator. The five examples consume approximately 4,800 tokens combined.

The five-shot prompt has a fixed template. We show the structure below rather than reproducing the full code for all five

examples.

---

**Five-Shot Format (Structure)**

```
[System Preamble as above]
```
Below are five examples of circuit design tasks and solutions. Study these, then solve the final task.

---

**Example 1/5 — Expression Cassette**
**Task:** Design an expression cassette with a strong constitutive promoter (J23100) driving GFP through a medium RBS (B0032), terminated by B0015.
**Solution pattern:**

- import `sbol3` and set the namespace

- create promoter, RBS, CDS, and terminator components with Sequence Ontology roles

- assemble them into one engineered region via `SubComponent` objects

- add `SBOL_PRECEDES` constraints in promoter → RBS → CDS → terminator order

- add all top-level objects to the document

---

**Example 2/5 — NOT Gate** [Prompt and canonical solution for a TetR-based NOT gate]
**Example 3/5 — Incoherent FFL** [Prompt and canonical solution for an I1-FFL using AraC activation and LacI repression]
**Example 4/5 — Toggle Switch** [Prompt and canonical solution for a LacI/TetR toggle switch]
**Example 5/5 — Repressilator** [Prompt and canonical solution for a 3-gene repressilator using TetR, LacI, and cI]

---

**Now solve the following task:**
**Task:** [Test prompt in the same format as the examples]

---

**Evaluation fairness.** The pysbol3 API reference in the system preamble provides Claude Opus 4.5 with documentation that base Qwen3-8B also lacks prior to SFT. Our trained model sees approximately 2,993 training examples exercising the API, compared to the frontier model's static reference; this asymmetry favors our method. We provide 5-shot examples spanning five circuit types, but the trained model has seen hundreds of examples per type. We report both zero-shot and 5-shot results to bracket the range of few-shot benefit (Table 57).

G.4.6. SFT TRAINING FORMAT

The SFT phase uses two training formats: *standard pairs* and *construction trace pairs*. Both share the system preamble from Appendix G.4.5. Approximately 30% of training examples are augmented with construction traces; the remaining 70% use the standard format. Trace-augmented examples are sampled uniformly across circuit types to avoid biasing toward any particular construction pattern.

**Standard format.** Standard training examples consist of a task-specific prompt paired with canonical pysbol3 code as the target completion. The model is trained to predict code tokens autoregressively conditioned on the prompt.

**Construction trace format.** Trace-augmented examples decompose the target code into four logical stages, each preceded by a reasoning comment explaining the construction rationale. The stages mirror the verification hierarchy: (1) document setup, (2) component creation with annotations, (3) assembly and constraints, and (4) interaction definitions. By making construction logic explicit, we encourage the model to internalize a structured generation strategy aligned with hierarchical verification.

We show the construction trace format for a NOT gate (the same circuit as the standard format; the prompt is identical in both cases and is omitted here).

```python
# =============================================
# STAGE 1: Document Setup
# =============================================
# Initialize namespace and document. All components
# are registered under this namespace for URI
# resolution.
import sbol3
from sbol_helpers import (
    SO_PROMOTER, SO_RBS, SO_CDS, SO_TERMINATOR,
    SO_ENGINEERED_REGION, SBO_INHIBITION,
    SBO_GENETIC_PRODUCTION
)

sbol3.set_namespace('https://synthia.org/')
doc = sbol3.Document()

# =============================================
# STAGE 2: Component Creation
# =============================================
# A NOT gate requires two cassettes: an input
# cassette constitutively expressing the TetR
# repressor, and an output cassette whose promoter
# (pTetR) is repressed by TetR.
# Logic: when TetR is active -> pTetR OFF -> GFP OFF
#        when TetR inactive (aTc) -> pTetR ON -> GFP ON
# This implements Y = NOT(A).

# Input cassette parts
p_in = sbol3.Component('J23100_promoter',
    types=[sbol3.SBO_DNA])
p_in.roles = [SO_PROMOTER]
p_in.name = 'J23100'  # Strong constitutive
doc.add(p_in)

rbs_in = sbol3.Component('B0034_rbs',
    types=[sbol3.SBO_DNA])
rbs_in.roles = [SO_RBS]
doc.add(rbs_in)

tetR_cds = sbol3.Component('TetR_cds',
    types=[sbol3.SBO_DNA])
tetR_cds.roles = [SO_CDS]
doc.add(tetR_cds)

term1 = sbol3.Component('B0015_t1',
    types=[sbol3.SBO_DNA])
term1.roles = [SO_TERMINATOR]
doc.add(term1)

# Output cassette parts
p_out = sbol3.Component('pTetR_promoter',
    types=[sbol3.SBO_DNA])
p_out.roles = [SO_PROMOTER]
doc.add(p_out)

rbs_out = sbol3.Component('B0032_rbs',
    types=[sbol3.SBO_DNA])
rbs_out.roles = [SO_RBS]
doc.add(rbs_out)

gfp = sbol3.Component('GFP_cds',
    types=[sbol3.SBO_DNA])
gfp.roles = [SO_CDS]
doc.add(gfp)

term2 = sbol3.Component('B0015_t2',
    types=[sbol3.SBO_DNA])
term2.roles = [SO_TERMINATOR]
doc.add(term2)

# TetR protein (required for interaction modeling)
tetR_prot = sbol3.Component('TetR_protein',
    types=[sbol3.SBO_PROTEIN])
doc.add(tetR_prot)

# =============================================
```

```
# STAGE 3: Assembly and Constraints
# ============================================
# Each cassette follows canonical ordering:
# promoter -> RBS -> CDS -> terminator
# Enforced via SBOL_PRECEDES constraints.

# Input cassette
input_cas = sbol3.Component('input_cassette',
    types=[sbol3.SBO_DNA])
input_cas.roles = [SO_ENGINEERED_REGION]
sc_p_in = sbol3.SubComponent(p_in)
sc_rbs_in = sbol3.SubComponent(rbs_in)
sc_tetR = sbol3.SubComponent(tetR_cds)
sc_t1 = sbol3.SubComponent(term1)
input_cas.features = [
    sc_p_in, sc_rbs_in, sc_tetR, sc_t1]
input_cas.constraints = [
    sbol3.Constraint(
        sbol3.SBOL_PRECEDES, sc_p_in, sc_rbs_in),
    sbol3.Constraint(
        sbol3.SBOL_PRECEDES, sc_rbs_in, sc_tetR),
    sbol3.Constraint(
        sbol3.SBOL_PRECEDES, sc_tetR, sc_t1)
]
doc.add(input_cas)

# Output cassette
output_cas = sbol3.Component('output_cassette',
    types=[sbol3.SBO_DNA])
output_cas.roles = [SO_ENGINEERED_REGION]
sc_p_out = sbol3.SubComponent(p_out)
sc_rbs_out = sbol3.SubComponent(rbs_out)
sc_gfp = sbol3.SubComponent(gfp)
sc_t2 = sbol3.SubComponent(term2)
output_cas.features = [
    sc_p_out, sc_rbs_out, sc_gfp, sc_t2]
output_cas.constraints = [
    sbol3.Constraint(
        sbol3.SBOL_PRECEDES, sc_p_out, sc_rbs_out),
    sbol3.Constraint(
        sbol3.SBOL_PRECEDES, sc_rbs_out, sc_gfp),
    sbol3.Constraint(
        sbol3.SBOL_PRECEDES, sc_gfp, sc_t2)
]
doc.add(output_cas)

# ============================================
# STAGE 4: Interaction Definitions
# ============================================
# Two interactions define the NOT gate:
# 1. Genetic production: TetR CDS -> TetR protein
# 2. Inhibition: TetR protein -| pTetR promoter
# Together: constitutive TetR represses pTetR,
# blocking GFP output.

# Production
prod = sbol3.Interaction('prod_tetR',
    types=[SBO_GENETIC_PRODUCTION])
prod.participations = [
    sbol3.Participation(
        [sbol3.SBO_TEMPLATE], sc_tetR),
    sbol3.Participation(
        [sbol3.SBO_PRODUCT], tetR_prot)
]

# Repression: TetR -| pTetR
repr_int = sbol3.Interaction('repr_tetR',
    types=[SBO_INHIBITION])
repr_int.participations = [
    sbol3.Participation(
        [sbol3.SBO_INHIBITOR], tetR_prot),
    sbol3.Participation(
        [sbol3.SBO_INHIBITED], sc_p_out)
]
```

The construction trace differs from the standard format in three ways. First, stage headers (STAGE 1–4) decompose generation into subtasks aligned with verification levels. Second, reasoning comments before each stage explain *why* each

construction choice is made (e.g., "A NOT gate requires two cassettes. . . "), encouraging topology planning before code generation rather than token-level pattern matching. Third, inline annotations link biological function to code structure (e.g., "TetR protein inhibits pTetR promoter"), reinforcing the mapping from regulatory logic to SBOL representation.

We find that construction traces improve SFT performance on functional tasks by 3.2 percentage points (T6) and 4.1 percentage points (T7) compared to training exclusively on standard-format examples, with negligible effect on code fundamentals (T1–T2) where the reasoning overhead is unnecessary. The 30% augmentation ratio was selected via grid search over $\{10\%, 20\%, 30\%, 50\%, 100\%\}$; higher ratios degraded T1–T2 performance, as the model generated unnecessary commentary instead of direct code.

### G.5. Extended Future Directions

This section outlines extensions that follow from the current framework but are outside of scope.

**Simulation-Augmented Verification.**  Our hierarchical verification evaluates structural and semantic correctness and performs task-specific checks (Level 5) through symbolic analysis of regulatory graphs. It confirms topology, truth tables, and motif presence without kinetic simulation. This topological verification is a necessary first stage because a circuit with incorrect topology cannot exhibit correct dynamic behavior regardless of parameterization. While topology-based prediction is reliable for canonical circuit classes, quantitative properties such as oscillation period, switching threshold, and response time require dynamic modeling beyond the current verifier. A natural extension is to incorporate ordinary differential equation (ODE) simulation into the reward structure, export generated circuits to Systems Biology Markup Language (Hucka et al., 2019), parameterize them with published characterization data, and simulate their behavior directly. Preliminary experiments exporting model outputs to COPASI (Bergmann et al., 2017) demonstrate feasibility: toggle switch circuits can be assessed for bistability by verifying convergence to distinct steady states from different initial conditions, while oscillator circuits can be evaluated for sustained oscillation through autocorrelation analysis of simulated trajectories. The computational cost of simulation-in-the-loop training motivates surrogate models that predict dynamic behavior from circuit parameters and provide faster approximate feedback for RL optimization.

**Expanded Biological Scope.**  The current parts library, while sufficient for the circuit types in our benchmark, represents a fraction of characterized biological components. Extension to additional orthogonal repressor systems, CRISPR-based regulation, quorum-sensing modules (e.g., LuxR, RhlR), or light-inducible transcription factors would enable generation of more diverse circuit architectures. Our evaluation focuses on *E. coli* genetic circuits, the most thoroughly characterized context for synthetic biology; generalization to other chassis organisms would require organism-specific parts libraries and verification criteria reflecting distinct regulatory mechanisms. The modular design of our verification hierarchy facilitates such extensions: new part types require corresponding ontology mappings and constraint specifications, while the overall reward structure remains unchanged.

**Sequence-Level Integration.**  GenCircuit-RL operates at the Component abstraction level, treating parts as functional units with known properties. Integration with biological foundation models operating on nucleotide sequences could enable end-to-end design from specification to DNA sequence, incorporating sequence-level considerations such as codon optimization, RBS strength prediction, and avoidance of problematic secondary structures. Such integration would bridge the gap between abstract circuit topology and implementable genetic constructs.

**Experimental Validation Pipelines.**  Standardized SBOL outputs from GenCircuit-RL are compatible with DNA assembly workflows. The Literature-91 evaluation provides indirect validation by assessing whether models rediscover experimentally verified designs, but systematic wet-lab characterization of model-generated novel circuits would establish practical utility of learned design capabilities.

## H. Preliminary Results: Quantitative Refinement via RLAIF

GenCircuit-RL optimizes for topological correctness but does not address quantitative circuit performance, such as expression levels, fold change, or dynamic range. We present preliminary results integrating ML-based performance estimators into an RLAIF (RL from AI Feedback) refinement loop, following the paradigm demonstrated for electronic circuits by AUTOCIRCUIT-RL (Vijayaraghavan et al., 2025).

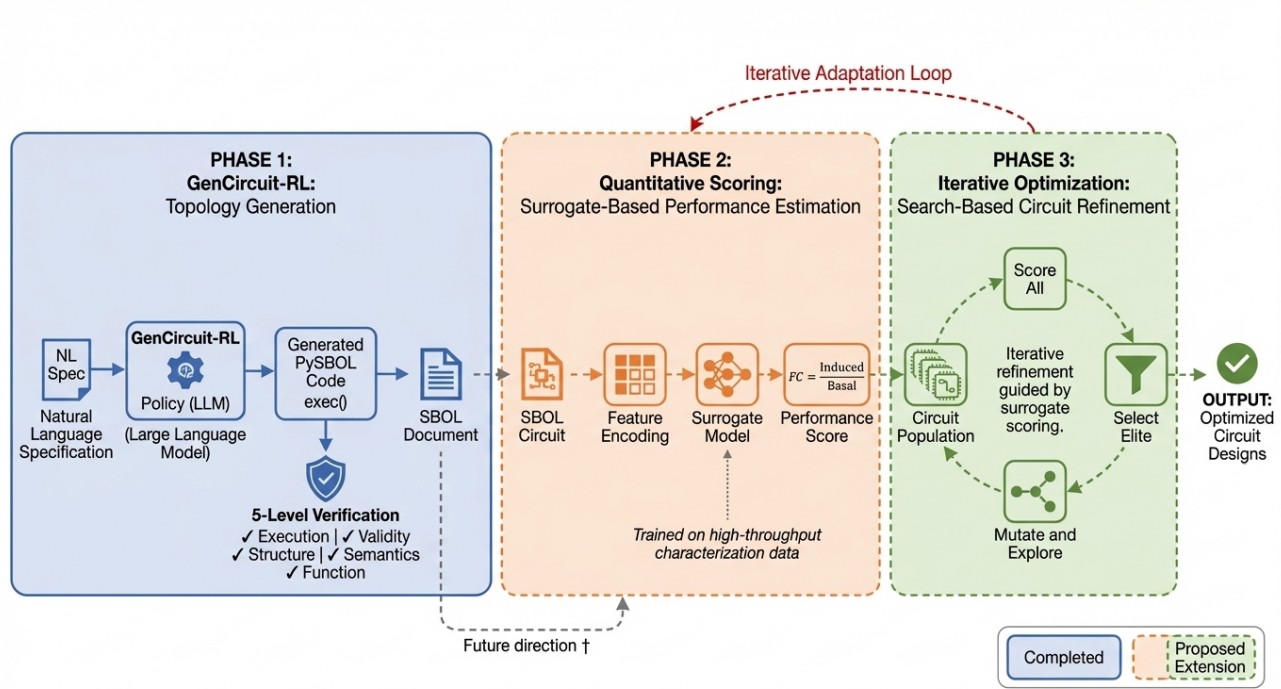

*Figure 7.* **Three-phase methodology for refining AI-generated genetic circuits using machine learning surrogates.** Phase 1 generates topologically valid circuits from natural language specifications via GenCircuit-RL and the five-level verification hierarchy. Phase 2 quantitatively scores circuits using a trained MLP surrogate operating on one-hot composition vectors derived from CLASSIC high-throughput data. Phase 3 employs an iterative RLAIF refinement loop: a pool of 2,000 circuits undergoes mutation (rate = 0.3, with 10% fresh random circuits), surrogate scoring, and elite selection (top 15%) over 8 iterations, enriching high fold-change circuits.

## H.1. Approach

Our proposed extension operates in three phases (Figure 7):

1. **Phase 1: Topology Generation (GenCircuit-RL).** The trained GenCircuit-RL agent generates structurally valid genetic circuits from natural language specifications, verified through the 5-level hierarchical reward. This phase ensures topological correctness (execution, validity, structural, semantic, and task-specific checks) but does not constrain quantitative behavior.

2. **Phase 2: Quantitative Estimation (CLASSIC Surrogate).** Generated circuits are encoded as one-hot composition vectors over their constituent genetic parts and scored by an MLP surrogate distilled from high-throughput experimental data and model weights from the CLASSIC platform (Rai et al., 2025). The surrogate predicts basal expression, induced expression, and fold change from the circuit's part composition.

3. **Phase 3: RLAIF Refinement.** The surrogate predictions serve as AI feedback for reward computation. Following AUTOCIRCUIT-RL's iterative adaptation procedure, we apply reward-weighted sampling: at each iteration, a pool of circuits is scored, the top-$k$ are selected as elite, and mutants of the elite (with exploration via fresh random samples) form the next generation. The composite reward is:

$$r(\hat{Y}) = 0.3 \cdot r_{\text{topo}}(\hat{Y}) + 0.5 \cdot r_{\text{fc}}(\hat{Y}) + 0.2 \cdot r_{\text{basal}}(\hat{Y}) \tag{23}$$

where $r_{\text{topo}}$ is the GenCircuit-RL hierarchical verification score (assumed 1.0 for valid circuits), $r_{\text{fc}}$ is a sigmoid-shaped fold-change reward centered at the target threshold, and $r_{\text{basal}}$ penalizes high basal expression (leaky circuits).

## H.2. Experimental Setup

**Design space.** We use the single-input inducible circuit library from CLASSIC ([Rai et al., 2025](#)), comprising 10 diversified genetic part categories (promoter, activation domain, IDP domain, zinc finger, terminator, $\text{spacer}_1$, orientation, binding site number, core promoter, $\text{spacer}_2$) spanning 165,888 possible compositions. CLASSIC experimentally characterized 121,292 of these (73% coverage) via combined Nanopore and Illumina sequencing.

**Surrogate model.** We train an MLP surrogate mirroring the CLASSIC architecture (34 one-hot inputs $\rightarrow 160 \rightarrow 80 \rightarrow 40 \rightarrow 20 \rightarrow 2$ outputs) with tanh activations on a synthetic dataset of whose expression values are generated from released CLASSIC MLP surrogates.

**RLAIF configuration.** Pool size $N = 2{,}000$, top-$k$ fraction 15%, mutation rate $\alpha = 0.3$ (standard), 10% fresh random exploration per iteration, 8 iterations. Target: high fold-change (HFC) circuits matching the HFC threshold used in CLASSIC's cluster analysis.

**Baselines.** We compare against: (i) *random baseline* (uniform sampling from the design space, representing GenCircuit-RL outputs without quantitative optimization); (ii) *low mutation* ($\alpha = 0.15$, reduced exploration); (iii) *high mutation* ($\alpha = 0.5$, increased exploration).

## H.3. Results

**Surrogate accuracy.** The original CLASSIC MLP predictions correlate strongly with flow cytometry measurements, confirming the viability of ML surrogates as reward signals.

**RLAIF convergence.** Figure 8 shows the refinement dynamics over 8 iterations, starting from a random baseline with mean FC of 8.2 and 7.8% HFC rate. The standard configuration achieves mean FC from 8.2 to 28.1 ($3.4\times$ improvement) and HFC rate from 7.8% to 52.1% ($6.7\times$ enrichment).

The low-mutation ablation ($\alpha = 0.15$) converges more slowly and plateaus at a lower HFC rate (34.8%), confirming that sufficient exploration is necessary to escape local optima in the combinatorial design space. The high-mutation variant ($\alpha = 0.5$) shows faster initial gains but slightly lower final performance (45.8% HFC), suggesting that excessive perturbation disrupts beneficial part combinations discovered in earlier iterations.

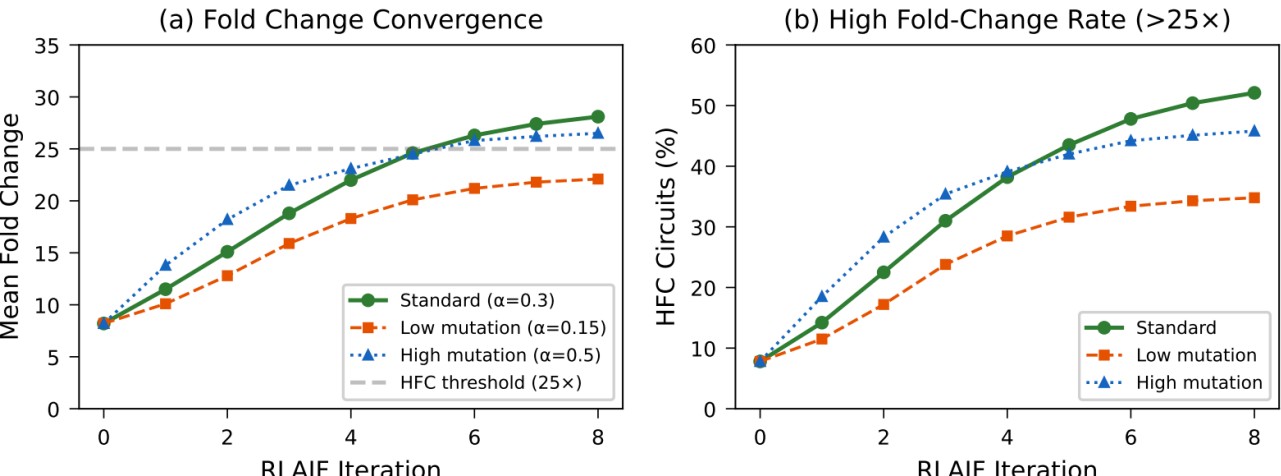

*Figure 8.* RLAIF refinement dynamics. (a) Mean fold change over iterations with three mutation rate configurations. (b) Percentage of circuits exceeding the HFC threshold ($> 25\times$).

## H.4. Discussion

Preliminary results demonstrate feasibility that (1) ML surrogates trained on high-throughput data can serve as effective RLAIF reward signals for genetic circuit optimization and (2) RLAIF refinement enriches for target quantitative behavior,

improving HFC rate from 7.8% to 52.1% over 8 iterations.

However, the MLP surrogate is distilled from the CLASSIS MLP rather than trained directly on the full CLASSIC experimental dataset (>121K measurements). Integration with the actual dataset would provide a stronger test of generalization. The surrogate is also limited to predicting steady-state expression levels. Extension to dynamic behaviors (oscillation period, switching kinetics) would require integration with kinetic simulators such as COPASI.

---

**Algorithm 3** GenerateTwoInputGate

---

**Require:** Parts library $\mathcal{L}$, gate type $\in$ {NOR, AND, OR, NAND}, seed $s$
**Ensure:** CircuitSpec for two-input gate
  Initialize random state with seed $s$

  **Repressor Assignment:**
  Select repressor pool from {LacI, TetR, cI, PhlF, SrpR}
  Assign repressors to gate layers based on gate type (see below)

  **if** gate_type = NOR **then**
    {NOR: Output ON iff both inputs OFF}
    **Architecture (2 cassettes + 1 output):**
    Select $(Rep_A, Rep_B)$ from repressor pool
    Let $P_{\text{out}}$ be a promoter repressible by both $Rep_A$ and $Rep_B$
    Input cassette A: $P_{\text{in,A}} \rightarrow$ RBS $\rightarrow Rep_A \rightarrow$ Term
    Input cassette B: $P_{\text{in,B}} \rightarrow$ RBS $\rightarrow Rep_B \rightarrow$ Term
    Output cassette: $P_{\text{out}} \rightarrow$ RBS $\rightarrow$ Reporter $\rightarrow$ Term

    **Interactions:**
    $Rep_A \dashv P_{\text{out}}$, $Rep_B \dashv P_{\text{out}}$
    truth_table $\leftarrow \{(0,0,1),(0,1,0),(1,0,0),(1,1,0)\}$

  **else if** gate_type = AND **then**
    {AND = NOT(NOR(NOT(A), NOT(B))): 3-layer architecture}
    **Architecture (4 cassettes + 1 output):**
    Select $(Rep_A, Rep_B, Rep_X, Rep_Y)$ from repressor pool

    {Layer 1: Input inversions}
    Cassette A: $P_{\text{in,A}} \rightarrow$ RBS $\rightarrow Rep_A \rightarrow$ Term
    Cassette B: $P_{\text{in,B}} \rightarrow$ RBS $\rightarrow Rep_B \rightarrow$ Term

    {Layer 2: Intermediate NOT gates producing NOT(A) and NOT(B)}
    Cassette X: $P_A \rightarrow$ RBS $\rightarrow Rep_X \rightarrow$ Term {$P_A$ repressed by $Rep_A$}
    Cassette Y: $P_B \rightarrow$ RBS $\rightarrow Rep_Y \rightarrow$ Term {$P_B$ repressed by $Rep_B$}

    {Layer 3: NOR(NOT(A), NOT(B)) = AND(A, B)}
    Output cassette: $P_{\text{out}} \rightarrow$ RBS $\rightarrow$ Reporter $\rightarrow$ Term
    {$P_{\text{out}}$ repressed by both $Rep_X$ and $Rep_Y$}

    **Interactions:**
    $Rep_A \dashv P_A$, $Rep_B \dashv P_B$
    $Rep_X \dashv P_{\text{out}}$, $Rep_Y \dashv P_{\text{out}}$
    truth_table $\leftarrow \{(0,0,0),(0,1,0),(1,0,0),(1,1,1)\}$

  **else if** gate_type = OR **then**
    {OR = NOT(NOR(A, B)): 2-layer architecture}
    **Architecture (3 cassettes + 1 output):**
    Select $(Rep_A, Rep_B, Rep_{\text{int}})$ from repressor pool

    {Layer 1: NOR(A, B) at intermediate node}
    Cassette A: $P_{\text{in,A}} \rightarrow$ RBS $\rightarrow Rep_A \rightarrow$ Term
    Cassette B: $P_{\text{in,B}} \rightarrow$ RBS $\rightarrow Rep_B \rightarrow$ Term
    Cassette Int: $P_{\text{int}} \rightarrow$ RBS $\rightarrow Rep_{\text{int}} \rightarrow$ Term
    {$P_{\text{int}}$ repressed by both $Rep_A$ and $Rep_B$}

    {Layer 2: NOT(NOR(A,B)) = OR(A,B)}
    Output cassette: $P_{\text{out}} \rightarrow$ RBS $\rightarrow$ Reporter $\rightarrow$ Term
    {$P_{\text{out}}$ repressed by $Rep_{\text{int}}$}

    **Interactions:**

---

**Algorithm 4** GenerateToggleSwitch

---

**Require:** Parts library $\mathcal{L}$, seed $s$
**Ensure:** CircuitSpec for toggle switch
  Initialize random state with seed $s$
  Select repressor pair $(Rep_1, Rep_2)$ from $\binom{5}{2}$ combinations
  Let $P_1, P_2$ be cognate promoters for $Rep_1, Rep_2$

  **Cassette 1:** $P_2 \rightarrow$ RBS $\rightarrow Rep_1 \rightarrow$ Term
  **Cassette 2:** $P_1 \rightarrow$ RBS $\rightarrow Rep_2 \rightarrow$ Term

  {Cassette 1 expresses $Rep_1$, which represses $P_1$ (driving Cassette 2)}
  {Cassette 2 expresses $Rep_2$, which represses $P_2$ (driving Cassette 1)}

  Create Interactions:
    $Rep_1 \dashv P_1$ (SBO:0000169)
    $Rep_2 \dashv P_2$ (SBO:0000169)

  **Ground Truth:**
  stable_states $\leftarrow$ {"$Rep_1$ high, $Rep_2$ low", "$Rep_1$ low, $Rep_2$ high"}
  bistable $\leftarrow$ True
  switching_inputs $\leftarrow$ inducers for $P_1$ and $P_2$ if inducible
  **return** CircuitSpec with mutual repression motif

---

---

**Algorithm 5** GenerateBranchedActivation

---

**Require:** Parts library $\mathcal{L}$, seed $s$
**Ensure:** CircuitSpec for branched activation motif
  Initialize random state with seed $s$

  **Topology:**
  Node A: AraC activator (master regulator, requires arabinose)
  Node B: Reporter 1 (e.g., GFP)
  Node C: Reporter 2 (e.g., RFP)
  Edges: $A \rightarrow B$ (activation), $A \rightarrow C$ (activation)
  {No regulatory edge between B and C; parallel output branches}

  **Cassette Construction:**
  Cassette A: $P_{\text{const}} \rightarrow$ RBS $\rightarrow$ AraC $\rightarrow$ Term
  Cassette B: $P_{\text{BAD}} \rightarrow$ RBS $\rightarrow$ Reporter$_1 \rightarrow$ Term
  Cassette C: $P_{\text{BAD}} \rightarrow$ RBS $\rightarrow$ Reporter$_2 \rightarrow$ Term
  {$P_{\text{BAD}}$ is activated by AraC in presence of arabinose}

  **Regulatory Interactions:**
  Create Interaction: AraC $\rightarrow P_{\text{BAD}}$ (SBO:0000170, stimulation) $\times 2$

  **Ground Truth:**
  expected_dynamics $\leftarrow$ "coordinated activation: B and C rise together upon arabinose induction"
  motif_type $\leftarrow$ "branched_activation"
  **return** CircuitSpec with branched topology

---

---

**Algorithm 6** GenerateFFL

---

**Require:** Parts library $\mathcal{L}$, FFL type $\in$ {C1, I1}, seed $s$
**Ensure:** CircuitSpec for feed-forward loop
  Initialize random state with seed $s$

  **if** FFL_type = C1 **then**
    {Coherent Type-1: all activating edges, AND-gate output logic}
    Node A: AraC (master activator, arabinose-inducible)
    Node B: LuxR (secondary activator, constitutively active when expressed)
    Node C: Output reporter

    **Cassette Construction:**
    Cassette A: $P_{\text{const}} \rightarrow$ RBS $\rightarrow$ AraC $\rightarrow$ Term
    Cassette B: $P_{\text{BAD}} \rightarrow$ RBS $\rightarrow$ LuxR $\rightarrow$ Term
    Cassette C: $P_{\text{lux/ara}} \rightarrow$ RBS $\rightarrow$ Reporter $\rightarrow$ Term
    {$P_{\text{lux/ara}}$ is a hybrid promoter requiring both AraC and LuxR for activation}

    **Regulatory Interactions:**
    Create Interaction: AraC $\rightarrow P_{\text{BAD}}$ (SBO:0000170, stimulation)
    Create Interaction: AraC $\rightarrow P_{\text{lux/ara}}$ (SBO:0000170, stimulation)
    Create Interaction: LuxR $\rightarrow P_{\text{lux/ara}}$ (SBO:0000170, stimulation)

    **Ground Truth:**
    expected_dynamics $\leftarrow$ "sign-sensitive delay: C activation delayed relative to B upon arabinose addition; rapid C deactivation upon arabinose removal"
    output_logic $\leftarrow$ "AND(A, B)"
  **else if** FFL_type = I1 **then**
    {Incoherent Type-1: A activates B and C, B represses C}
    Node A: AraC (activator)
    Node B: Select repressor $Rep$ from {LacI, TetR, cI, PhlF, SrpR}
    Node C: Output reporter
    Let $P_{\text{rep}}$ be the cognate promoter for $Rep$

    **Cassette Construction:**
    Cassette A: $P_{\text{const}} \rightarrow$ RBS $\rightarrow$ AraC $\rightarrow$ Term
    Cassette B: $P_{\text{BAD}} \rightarrow$ RBS $\rightarrow Rep \rightarrow$ Term
    Cassette C: $P_{\text{hybrid}} \rightarrow$ RBS $\rightarrow$ Reporter $\rightarrow$ Term
    {$P_{\text{hybrid}}$ is activated by AraC but repressible by $Rep$}

    **Regulatory Interactions:**
    Create Interaction: AraC $\rightarrow P_{\text{BAD}}$ (SBO:0000170, stimulation)
    Create Interaction: AraC $\rightarrow P_{\text{hybrid}}$ (SBO:0000170, stimulation)
    Create Interaction: $Rep \dashv P_{\text{hybrid}}$ (SBO:0000169, inhibition)

    **Ground Truth:**
    expected_dynamics $\leftarrow$ "pulse generation: C transiently activated then repressed as B accumulates"
    output_logic $\leftarrow$ "A AND NOT(B), with temporal delay on B"
  **end if**
  **return** CircuitSpec with FFL topology and expected dynamics

---

---

**Algorithm 7** GenerateOscillator

---

**Require:** Parts library $\mathcal{L}$, cycle length $n \in \{3, 5\}$, seed $s$
**Ensure:** CircuitSpec for $n$-node oscillator
  Initialize random state with seed $s$

  **Repressor Selection:**
  Select $n$ repressors $(Rep_1, \ldots, Rep_n)$ from 5 training-tier repressors
  Let $(P_1, \ldots, P_n)$ be their cognate promoters (i.e., $Rep_i \dashv P_i$)

  **Ring Construction:**
  **for** $i = 1$ **to** $n$ **do**
    $j \leftarrow ((i \mod n) + 1)$ {Index of upstream repressor}
    Cassette $i$: $P_j \rightarrow$ RBS $\rightarrow Rep_i \rightarrow$ Term
    {Cassette $i$ is driven by $P_j$, which is repressed by $Rep_j$}
    {Thus $Rep_j \dashv Rep_i$: the upstream repressor inhibits this cassette's output}
  **end for**

  {Resulting regulatory ring: $Rep_2 \dashv Rep_1 \dashv Rep_n \dashv Rep_{n-1} \dashv \cdots \dashv Rep_2$}

  **Regulatory Interactions:**
  **for** $i = 1$ **to** $n$ **do**
    Create Interaction: $Rep_i \dashv P_i$ (SBO:0000169)
  **end for**

  **Ground Truth:**
  cycle_length $\leftarrow n$
  **if** $n$ is odd **then**
    oscillation_expected $\leftarrow$ True
    rationale $\leftarrow$ "odd-length repression ring yields net negative feedback"
  **else**
    oscillation_expected $\leftarrow$ "bifurcation-dependent"
    rationale $\leftarrow$ "even-length ring yields net positive feedback; typically bistable, though oscillations possible with sufficient time delays or stochastic effects"
  **end if**
  **return** CircuitSpec with ring topology and oscillation prediction

---

*Table 59.* Extended parts library. All parts have published characterization data enabling topological verification.

| Category | Part | Function | Source |
|---|---|---|---|
| *Quorum Sensing Components* | | | |
| LuxI | synthase | Produces 3OC6-HSL autoinducer | *V. fischeri* |
| LuxR | regulator | Activates pLux in presence of 3OC6-HSL | *V. fischeri* |
| pLux | promoter | LuxR-activated, quorum-responsive | *V. fischeri* |
| LasI | synthase | Produces 3OC12-HSL autoinducer | *P. aeruginosa* |
| LasR | regulator | Activates pLas in presence of 3OC12-HSL | *P. aeruginosa* |
| pLas | promoter | LasR-activated | *P. aeruginosa* |
| RhlI | synthase | Produces C4-HSL autoinducer | *P. aeruginosa* |
| RhlR | regulator | Activates pRhl in presence of C4-HSL | *P. aeruginosa* |
| EsaR | repressor | Represses PesaR in absence of 3OC6-HSL | *P. stewartii* |
| AiiA | enzyme | AHL lactonase, degrades autoinducers | *Bacillus* |
| Gene E | lysis | Phage lysis protein for synchronized release | Phage $\phi$X174 |
| *Light Sensor Components* | | | |
| Cph8 | sensor | Red light-responsive histidine kinase | Cyanobacterial |
| OmpR | regulator | Response regulator for Cph8 | *E. coli* |
| pOmpC | promoter | OmpR-activated output promoter | *E. coli* |
| CcaS | sensor | Green/red switchable sensor kinase | *Synechocystis* |
| CcaR | regulator | Response regulator for CcaS | *Synechocystis* |
| pCpcG2 | promoter | CcaR-activated output promoter | *Synechocystis* |
| YF1 | sensor | Blue light-repressed LOV-histidine kinase | Chimeric |
| FixJ | regulator | Response regulator for YF1 | *B. japonicum* |
| pFixK2 | promoter | FixJ-activated output promoter | *B. japonicum* |
| ho1 | enzyme | Heme oxygenase for PCB biosynthesis | Cyanobacterial |
| pcyA | enzyme | Phycocyanobilin:ferredoxin oxidoreductase | Cyanobacterial |
| UirS | sensor | UV-violet light-responsive histidine kinase | *Synechocystis* |
| UirR | regulator | Response regulator for UirS | *Synechocystis* |
| PcsiR1 | promoter | UirR-activated output promoter | *Synechocystis* |
| BphP1 | sensor | NIR-responsive bacteriophytochrome | *R. palustris* |
| PpsR2 | repressor | Sequestered by BphP1-Pfr, represses PcrtE | *R. palustris* |
| PcrtE | promoter | PpsR2-repressed output promoter | *R. palustris* |
| *Additional Cello Repressors* | | | |
| HlyIIR | repressor | TetR-family, orthogonal to training set | Cello library |
| IcaRA | repressor | TetR-family, orthogonal to training set | Cello library |
| LitR | repressor | TetR-family, orthogonal to training set | Cello library |
| PsrA | repressor | TetR-family, orthogonal to training set | Cello library |
| LmrA | repressor | TetR-family, orthogonal to training set | Cello library |
| *Memory/Counter Components* | | | |
| T7 RNAP | polymerase | Orthogonal transcription system | Phage T7 |
| T3 RNAP | polymerase | Orthogonal transcription system | Phage T3 |
| Cre | recombinase | Site-specific recombination at loxP | Phage P1 |
| Flpe | recombinase | Site-specific recombination at FRT | *S. cerevisiae* |
| FimE | recombinase | Unidirectional DNA inversion (fim system) | *E. coli* |
| FimB | recombinase | Bidirectional DNA inversion (fim system) | *E. coli* |
| Hin | recombinase | DNA inversion (hin system) | *S. typhimurium* |
| *Metabolic Components (Oscillators)* | | | |
| Acs | enzyme | Acetyl-CoA synthetase | *E. coli* |
| Pta | enzyme | Phosphate acetyltransferase | *E. coli* |
| Ack | enzyme | Acetate kinase | *E. coli* |

*Table 60.* Prompt structure by task type. All tasks share the system preamble described in Appendix G.4.5. The output format directive is identical across all tasks: executable pysbol3 code in a fenced code block.

| Stage | Task | Task Instruction | Circuit Context |
|---|---|---|---|
| 1 | T1: Code repair | Identify and fix errors so code executes and validates. | Buggy pysbol3 code + error output |
| 1 | T2: Code completion | Complete missing sections of provided code. | Partial pysbol3 code with # TODO markers |
| 2 | T3: Part substitution | Modify the circuit by making the specified replacement. | Working pysbol3 code + modification instruction |
| 2 | T4: NL to code | Generate pysbol3 code constructing the described circuit. | Natural language circuit description |
| 3 | T5: Logic prediction | Predict output state for the given input conditions. | pysbol3 code + input conditions |
| 3 | T6: Debugging | Identify the design flaw causing the malfunction; provide fix. | Executable pysbol3 code + symptom description |
| 4 | T7: De novo design | Design a circuit meeting the functional specification. | Functional specification + parts library reference |

