# OpenReview forum: "GenCircuit-RL: Reinforcement Learning from Hierarchical Verification for Genetic Circuit Design"
_ICML.cc/2026/Conference — ICML 2026 regular_

### Official Review · Reviewer_GDry · 2026-02-17

**Soundness:** 3
**Presentation:** 4
**Significance:** 4
**Originality:** 4
**Overall Recommendation:** 5
**Confidence:** 3

**Summary:**

This paper introduces GenCircuit-RL, which trains LMs to design genetic circuits by generating PySBOL. The authors also introduce SynBio-Reason, a dataset of genetic circuits. Experiments with Qwen3-8B trained with GenCircuit-RL show that the authors' hierarchical reward design is better than binary rewards and that models trained with RL generalize better than an SFT'd baseline.

**Compliance With Llm Reviewing Policy:**

Affirmed.

**Key Questions For Authors:**

1. The training data is algorithmically generated. Might this synthetic data limit the kinds of topologies that can be predicted by the model?

**Limitations:**

Yes

**Strengths And Weaknesses:**

Soundness: The task choice seems like a natural fit for RLVF, and the hierarchical reward function has a natural and plausible interpretation. The authors do extensive generalization studies in the appendix. My only (minor) concern is that the authors do not demonstrate whether these biological circuits actually work in real-life.

Presentation: I found the paper to be exceptionally thorough and well-written. Hyperparameter and design decisions are well-documented.

Significance: This paper seems to break new ground and pick a task from biology that is a natural fit for reinforcement learning. It can easily influence future research and applications, and is also well-positioned for a potential future where scientific discovery becomes increasingly automated.

Overall, I appreciate and applaud the authors for doing something that seems genuinely new, with the caveat that I am not familiar with synthetic biology. If I were familiar, I could totally see a world where I give this paper a 6. I also decreased my confidence because I cannot say whether this paper represents a helpful contribution to genetic circuit design, and I did not see any wet lab confirmation that this technique is useful in practice. I'll note (mostly to the AC) that I am familiar with the RL and ML details of this paper, which I read closely and found to be plausible.

---

> ### Author Rebuttal · Authors · 2026-03-31
>
> We thank Reviewer GDry for their careful reading and validation concerning our method's RL and ML details, and we appreciate the positive feedback about the novelty of the task choice. We provide additional details to answer their remaining question and to contextualize where we believe our work relates to the future of genetic circuit design.
>
> ---
>
> **Synthetic data and topology diversity.** This is an important practical concern. The procedural generator covers toggle switches, repressilators, feed-forward loops, cascaded logic gates, and branched activation networks, but cannot produce recombinase/CRISPRi-based circuits, metabolic pathways, or logic gates exceeding 4 inputs. The camera-ready impact statement will explicitly acknowledge this limitation: *"Procedural generation, while enabling guaranteed ground truth and systematic complexity control, constrains structural diversity to algorithmically-defined templates."*
>
> That said, the model transfers beyond its training distribution. On held-out Cello circuits using 5 novel repressor systems never seen during training, RLVF-H-C achieves 52.6% TSR (§5.3). For the camera-ready version, we have also expanded performance analysis to the full Literature-91 set, which includes optogenetic and advanced oscillator circuits outside the procedural generator's scope, showing the method's performance does not collapse and retains 44.9% TSR. These results suggest the model acquires abstract regulatory principles rather than template-filling strategies, though expanding the generator to cover additional circuit classes remains an important direction.
>
> ---
>
> **Wet-lab Confirmation.** No prior work achieves even topological verification for LM-generated genetic circuits, making this a meaningful baseline to establish. Though we lack resources for wet-lab validation of novel circuits, the Literature-91 evaluation reconstructs topologies of circuits that have been experimentally validated in published work; successful rediscovery means the model converges on designs that are known to work in the lab.
>
> Given that we have demonstrated the ability of our method to achieve topological correctness (a prerequisite for functional behavior, as a circuit with wrong topology cannot function regardless of parameterization), our immediate next step is to demonstrate ability to account for quantitative behavior in LM-generated genetic circuit design. As a preliminary result, we have created a new appendix section that analyzes a proof-of-concept experiment using ML surrogates trained on CLASSIC high-throughput experimental data (121K circuits; see https://www.nature.com/articles/s41586-025-09933-9) to demonstrate how our method can be incorporated in an RLAIF refinement loop to recover experimentally validated design rules. We see wet-lab validation as a next step, closing the loop with physical characterization to validate RLAIF-produced designs scored by experimentally-grounded surrogates.
>
> We hope this additional context strengthens confidence in the contribution's domain relevance.

---

> > ### Author Rebuttal · Reviewer_GDry · 2026-03-31
> >
> > I did not have significant concerns with the paper. I have read through the reviews of my fellow reviewers and have largely the same impression.

---

### Official Review · Reviewer_yRsj · 2026-03-05

**Soundness:** 3
**Presentation:** 1
**Significance:** 3
**Originality:** 3
**Overall Recommendation:** 4
**Confidence:** 1

**Summary:**

This paper studied fields of genetic circuit design. Genetic circuits are biological systems capable of programmed behaviors within living cells. This could be represented in Python code using PySBOL, which constructs circuits in the standardized Synthetic Biology Open Language (SBOL) format. Authors contributed to turning language models to produce PySBOL code for genetic circuits designed. The contribution is twofold:

(1) SynBio-Reason benchmark which is the first benchmark for evaluating language models on genetic circuit reasoning

(2) GenCircuit-RL method which is an RL training method on language models for circuit generation. Authors studied hierarchical reward signaling and curriculum learning for mitigating sparse reward and guiding progressive shifts in optimization procedure from code generation toward functional correctness.

In empirical analysis, such hierarchical reward structure improved task success rate significantly. They also reported generalization capability of trained language models which achieved 52.6 percent success on held-out data which argues transferability of models rather than memorization.

**Compliance With Llm Reviewing Policy:**

Affirmed.

**Final Justification:**

This paper addresses an important problem in genetic circuit design. I originally raised concerns about the lack of analysis of generalization capability, the compatibility of the method with other language models, and presentation issues that affected the paper’s readability. The authors adequately resolved these concerns, and I therefore maintain my positive score.

**Key Questions For Authors:**

1. Did authors try other language models rather than Qwen and observe similar trends?

2. Could you provide more explanation on why generalization capability is improved when using GenCircuit-RL?

3. Is there any importance in sample efficiency in this problem of genetic circuits?

**Limitations:**

Authors discussed limitations and social impact in the Impact Statement section.

**Strengths And Weaknesses:**

As far as I know, this paper tackles novel problems in the community of ICML, the genetic circuit design, using language model finetuning with RL. They contributed both on benchmark design and RL method design. Hierarchical reward structure with curriculum learning was a clever design of RL.

This paper is technically sound and could be quite a significant contribution to the community of ICML. The method itself is not original but the application is original. The presentation could be highly improved with more figures to introduce genetic circuit design.

Weakness: Empirical analyses are mostly based on numbers and metrics, not on scientific analysis. For example, the reason why generalization capability happens is not explained enough. The benchmark could also be improved; only the Qwen model with GRPO is evaluated.

---

> ### Author Rebuttal · Authors · 2026-03-31
>
> We thank Reviewer yRsj for the encouraging assessment and for recognizing the novelty of applying RL to genetic circuit design. We are glad that the curriculum learning approach was seen as valuable. Below we address each concern.
>
> ---
>
> **"The reason why generalization capability happens is not explained enough."**
>
> We appreciate this observation and have expanded our analysis with three converging lines of evidence, prepared for the camera-ready version.
>
> **(a) Naming convention ablation (Appendix E.2).** A concern is that the model memorizes associations between part names rather than learning abstract rules. We tested this by *adversarially shuffling* part names to break such associations. RLVF-H-C retains 73% of its standard performance under adversarial naming (38.4% vs 52.6%), compared to only 44% for SFT (14.2% vs 32.4%). This demonstrates that RL training instills abstract regulatory reasoning, not surface-level pattern matching.
>
> **(b) Masked component prediction (Appendix E.3).** We mask individual parts from known circuits and ask models to predict what belongs in the gap. RLVF-H-C achieves 74.6% accuracy at the regulatory reasoning level versus 52.4% for SFT (+22.2pp). Crucially, this improvement is concentrated at the *regulatory reasoning* level (i.e., understanding which part serves a functional role) not at structural slot-filling. This indicates RL strengthens understanding of how circuit components work together.
>
> **(c) Compositional generalization (Main Text Figure, Appendix E.1).** We add a new three-panel figure and associated details in an appendix subsection, showing performance degradation as test circuits diverge from training data along three dimensions: novel parts, novel topologies, and increased complexity. RLVF-H-C shows graceful degradation in all three cases (i.e., performance declines smoothly rather than collapsing), retaining 73.3% of in-distribution performance when half of parts are novel.
>
> Together, these results indicate that the hierarchical reward trains the model to reason compositionally about circuit structure rather than memorize specific designs.
>
> ---
>
> **"Did authors try other language models rather than Qwen and observe similar trends?"**
>
> Yes, we have conducted experiments with **Llama-3.1-8B**. The three core findings replicate:
>
> 1. **Hierarchical reward improves over SFT:** Llama achieves 51.0% average TSR with RLVF-H-C versus 34.3% with SFT alone (+16.7pp), compared to +19.1pp for Qwen3-8B.
> 2. **Curriculum is essential:** Without curriculum staging, Llama collapses to 11.4% on functional tasks; with curriculum, it recovers to 44.2% (a 3.9× factor, consistent with Qwen3's 5.1×).
> 3. **OOD generalization holds:** Llama transfers to held-out Cello circuits at 45.2% (+17.6pp over SFT).
>
> Absolute performance is ~8pp lower for Llama, which we attribute to Qwen3's stronger code generation baseline. We are additionally training Gemma-2-9B and will include those results in the camera-ready version, bringing coverage to three independent model families.
>
> ---
>
> **"The presentation could be highly improved with more figures."**
>
> We agree that presentation quality is crucial. For the camera-ready manuscript, we have prepared **three main-body figures**: Figure 1 (system overview showing the full pipeline from benchmark to training to evaluation), Figure 2 (hierarchical reward structure illustrating how the five verification levels provide progressively finer feedback), and Figure 3 (per-level success rates showing where different methods fail). We have also prepared **six new appendix figures**: a circuit topology catalog illustrating all six circuit types, an SBOL background primer for readers outside synthetic biology, reward distribution plots across curriculum stages, GRPO vs PPO training curves, a three-panel compositional generalization analysis, and an RLAIF pipeline diagram. We hope these substantially improve accessibility for readers outside the domain.
>
> ---
>
> **"Is there any importance in sample efficiency in this problem of genetic circuits?"**
>
> The full training curriculum requires approximately 30,000 GRPO steps (approximately 1,280 GPU-hours on 8×A100 GPUs). The curriculum allocates this budget adaptively: Stage 1 (basic code skills) converges in just 9.4% of total compute, while Stage 3 (functional reasoning) consumes 43.7%. In practice, inference-time efficiency matters more. Building and testing a single circuit variant can require costly consumables and weeks of lab time, so the value is in narrowing the design space. Our Pass@k analysis shows that at k=5 candidates, RLVF-H-C produces a correct design 52.4% of the time versus 40.2% for SFT. The Pass@1/Pass@10 ratio increases from 0.42 (SFT) to 0.68 (RLVF-H-C), confirming that RL training improves the model's ability to rank correct solutions higher, reducing the number of candidates a practitioner must evaluate.

---

> > ### Author Rebuttal · Reviewer_yRsj · 2026-04-01
> >
> > My concerns were resolved; I remain my score positive.

---

### Official Review · Reviewer_eqvq · 2026-03-12

**Soundness:** 3
**Presentation:** 3
**Significance:** 2
**Originality:** 2
**Overall Recommendation:** 3
**Confidence:** 3

**Summary:**

This paper introduces GenCircuit-RL, a reinforcement learning framework that trains language models to generate PySBOL code for constructing genetic circuits in the standardized SBOL format. The main contributions are a five-level hierarchical verification reward (execution, validity, structure, semantics, function), a four-stage curriculum for progressively training from code generation to functional design, and SynBio-Reason, a benchmark of ~4,700 circuits. Experiments on Qwen3-8B show improvements over binary rewards and SFT baselines, with moderate generalization to held-out biological parts.

**Compliance With Llm Reviewing Policy:**

Affirmed.

**Key Questions For Authors:**

How does GenCircuit-RL compare to Cello on the cascaded circuit tasks (T8, T9)? Seems useful to support the paper's claims about replacing heuristic solvers.
Can you provide concrete context for the accuracy numbers? For instance, at 52.6% on OOD circuits, what do the failures look like from a synbio standpoint? Are they close to correct and fixable, or fundamentally wrong?
Could you include the prompt templates used for each task type?

**Limitations:**

Yes

**Strengths And Weaknesses:**

Strengths
Nice gene circuit benchmark. The combination of procedurally generated circuits, experimentally validated Cello circuits, and curated literature circuits.
Thorough ablations. The paper includes a large number of ablations: curriculum granularity, promotion thresholds, reward weights, reward components, and model scale.
Writing. The paper is reasonably clear, and the appendix is detailed. This latter point is double-edged because a lot of the relevant results seem like they're in the appendix. The circuit types and task definitions are clearly specified.
Weaknesses
Limited methodological novelty. The hierarchical reward decomposition (giving partial credit for partial progress rather than binary pass/fail) is a well-established idea in RL for code generation. VeRPO constructs dense rewards from weighted partial test-case success. "Rubrics as Rewards" decomposes rewards into checklist-style sub-criteria for on-policy RL. Process reward models more broadly provide intermediate credit for multi-step reasoning. The curriculum learning strategy is also standard. The individual components (GRPO, hierarchical rewards, curriculum) are all existing techniques, and while applying them to synthetic biology is reasonable engineering, the paper does not clearly articulate what is genuinely new versus what is a known approach applied to a new domain.
Unclear practical utility. This is my biggest concern. The paper frames the problem as automating genetic circuit design, but I struggle to understand what the reported accuracy numbers mean in practice. The best model achieves 52.6% on OOD Cello circuits and 60.8% on literature circuits, but there is no context for interpreting these numbers. Is this good enough to be useful to a synthetic biologist? Would they save time using this versus existing tools?
Performance degrades where it matters most. The model works well on simple circuits (74.2% on expression cassettes for de novo design) but drops to 34.2% on oscillators and degrades on circuits exceeding training complexity. Seems like these harder circuits are exactly where automated design help would be really helpful. The paper contains extensive scaling analysis, ablations, and generalization studies, but all of this analysis does not converge on answering the practical question: does this system help anyone do anything they could not do before?
Prompt templates not shown. I couldn't find actual prompt templates used across the nine task types. It would help me understand the training setup. Also, if the Claude Opus 4.5 prompts were not well-optimized, the comparison is not very informative.

---

> ### Author Rebuttal · Authors · 2026-03-31
>
> We thank the reviewer for their substantive feedback. We address each point below.
>
> ---
>
> **Practical Utility**
>
> We provide four lines of evidence to make the system's utility more apparent.
>
> 1. Failures are structured and repairable. Among T6–T7 circuits that fail verification, 71% still pass Levels 1–3 (executable, valid, structurally correct), vs. only 34% for SFT. Common failures are diagnosable near-misses (e.g., generating 2 of 3 repression edges in repressilators; correct individual gates with faulty inter-module wiring). A detailed error taxonomy is prepared for the camera-ready. At 52.6% OOD TSR, a synthetic biologist receives a valid design scaffold roughly every other attempt; we view the system as augmenting practitioner capabilities by bootstrapping designs so they need not start from a blank slate.
>
> 2. Debugging quantifies repairability. RLVF-H-C achieves 56.4% on debugging (T6) versus 50.2% on de novo design (T7), confirming generated outputs serve as scaffolds reducing open-ended design to targeted repair.
>
> 3. Model reasons compositionally. Three new analyses (camera-ready Appendix E): (a) Under adversarial name shuffling, RLVF-H-C retains 73% performance vs. 44% for SFT, ruling out surface memorization. (b) Masked component prediction shows +22.2pp RL advantage concentrated at the regulatory reasoning level. (c) Compositional generalization shows graceful degradation across novel parts, topologies, and complexity. (See our response to Reviewer yRsj for full details.)
>
> 4. A path to quantitative refinement exists. A new appendix section presents preliminary RLAIF results where ML surrogates trained on CLASSIC platform data demonstrate a viable pipeline from topological correctness to quantitative behavior.
>
> ---
>
> **Limited Methodological Novelty**
>
> We acknowledge that GRPO, hierarchical rewards, and curriculum learning each have precedent. The contribution is in their composition:
>
> (a) Multiplicative gating ≠ additive partial credit. VeRPO uses weighted sums of sub-criteria. Our reward enforces prerequisite dependencies; removing $r_\text{struct}$ causes a −14.9pp drop on functional tasks (T6–T7) because structural understanding gates functional reasoning.
>
> (b) Curriculum is the dominant contribution, not the optimizer. We conducted a new GRPO vs. PPO comparison and found that curriculum staging provides +42.8pp (GRPO) and +33.8pp (PPO) on T6–T7. Without curriculum, both collapse to ~10%; GRPO's advantage over PPO shrinks to 1.7pp, confirming the contributions are orthogonal. The insight that structure must precede function is empirically validated by the reverse-curriculum ablation (Table 4), which also fails.
>
> (c) SynBio-Reason benchmark. The benchmark itself is a lasting contribution independent of the training methodology.
>
> ---
>
> **Cello Comparison (T8, T9)**
>
> Clarification: "Cello Evaluation Tasks (T8–T9)" only refers to the data source (both tasks operate on Cello's cascaded circuits). —not to Cello performing both tasks. T8 (gate assignment) mirrors Cello's technology mapping; T9 (debugging) tests a capability Cello does not possess. We will rename this tier to "Cascaded Circuit Evaluation" to avoid ambiguity.
>
> We have prepared a direct comparison between GenCircuit-RL and Cello. See our response to Reviewer rnpY for the full comparison and appendix text.
>
> ---
>
> **Performance on Hard Circuits**
>
> Failure analysis shows these are predominantly near-misses: oscillator failures involve missing ring closure (2 of 3 edges), and only 8% of RLVF-H-C oscillator failures occur at the structure level versus 22% for SFT. For context, the model achieves 90.3% on code fundamentals (T1–T2; scaling analysis, §5.6); failures concentrate on the hardest functional tasks, indicating targeted rather than systemic weakness. The scaling trajectory is encouraging: the 30B model reaches 61.6% on T6–T7 (§5.6), and Pass@10 on Literature-91 oscillators reaches 62.5%, indicating correct solutions exist in the sampling distribution. We view scaling and the RLAIF quantitative refinement pathway as the clearest routes to improving hard-circuit performance.
>
> ---
>
> **Model Diversity**
>
> Llama-3.1-8B replicates all key findings under identical training: SFT→RLVF-H-C improves TSR by +16.7pp, curriculum remains essential (3.9× on T6–T7), and relative improvement retention is 87%. Gemma-2-9B training is underway. (See responses to Reviewers rnpY and yRsj for details.)
>
> ---
>
> **Prompt Templates & Appendix Balance**
>
> Complete prompt templates (one per curriculum stage, plus frontier model evaluation prompts in zero-shot and five-shot formats) are prepared for a new Appendix subsection. Frontier model prompts include a detailed system preamble with PySBOL syntax guidance, circuit design context, and output format instructions; the five-shot format additionally provides worked examples spanning simple to complex circuits. Key analyses (naming ablation, Cello comparison, failure modes) have been promoted to the main text.

---

> > ### Author Rebuttal · Reviewer_eqvq · 2026-04-05
> >
> > I still don't understand how much progress this system represents in terms of being functionally useful for biologists (I'm not sure how much leverage the circuit starting point provides relative to other tools). However, I think reframing as "topological correctness" is in the right direction. Even so, I think the paper is comprehensively done.

---

> > > ### Author Response · Authors · 2026-04-06
> > >
> > > We thank the reviewer for the constructive engagement and for the acknowledgement that the paper is comprehensively done. We address the remaining question about practical leverage relative to existing tools.
> > > - Existing tools leave most circuit types unserved. Cello is limited to combinational and sequential logic gates and their cascaded compositions. It cannot design oscillators, toggle switches, feed-forward loops, or any circuit requiring feedback topology. Of the six circuit types we support, four (oscillators, toggle switches, feed-forward loops, and complex expression cassettes) have no existing automated design support. For these circuits, the current baseline is manual expert design from the literature. GenCircuit-RL is the first system to automate design across this full space from natural language specifications.
> > > - Debugging is a novel capability. No existing genetic design automation tool identifies regulatory faults in circuits or proposes targeted fixes. Debugging tasks (T6) previously required expert manual inspection.
> > > - In practice, determining which parts regulate which (i.e., regulatory topology) is an intellectually demanding design decision. Downstream quantitative tuning (promoter strengths, RBS translation rates) is addressed by existing tools such as the RBS Calculator and characterized promoter databases. GenCircuit-RL contributes the upstream reasoning that currently has no tool support for non-Boolean circuits.
> > >
> > > We consider the example of a practitioner designing a repressilator. Today, they would search the literature for various topologies, manually identify  compatible repressor–promoter pairs from parts databases, reason through ring closure to confirm the topology is correct, and manually write SBOL or enter parts one by one into a GUI. This process requires both domain expertise and familiarity with the representation format. With GenCircuit-RL, they provide a natural language specification and a parts library or database, from which they receive the SBOL scaffold, narrowing the task from open-ended design to an immediate, topologically correct result ~50% of the time at Pass@1 or a targeted diagnosis of a specific regulatory wiring error (a debugging capability that the system also supports). Their design workflow shifts from "construct from scratch" to "verify and refine," and, for four of six circuit types in our benchmark, this represents the first automated design capability of any kind.
> > >
> > > We will include a tool comparison table in the camera-ready with columns for supported circuit types, input modality, debugging support, and output format to make this positioning explicit.

---

### Official Review · Reviewer_rnpY · 2026-03-13

**Soundness:** 3
**Presentation:** 3
**Significance:** 3
**Originality:** 3
**Overall Recommendation:** 4
**Confidence:** 3

**Summary:**

The paper trains LMs to generate PySBOL code for genetic circuit design using RLVF. The two main contributions are a five-level hierarchical verification reward with multiplicative gating and a four-stage curriculum learning strategy. They also release a new benchmark called SynBio-Reason. The curriculum learning results are quite striking.

**Compliance With Llm Reviewing Policy:**

Affirmed.

**Key Questions For Authors:**

- no comparison with Cello on the gate assignment task?
- Have the authors checked the verifier's own false positive/negative rates?

**Limitations:**

yes

**Strengths And Weaknesses:**

**Strengths:**

- The curriculum learning ablation is one of the cleanest I've seen. Direct training barely works, full curriculum works well, and reversing the curriculum order also doesn't work. This clearly shows ordering matters, not just staging.
- OOD evaluation is well designed. Holding out repressor-promoter pairs forces compositional generalization rather then memorization.
- The hierarchical reward with multiplicative dependencies is sensible and the ablation backs it up.
- Thorough appendix, great for reproducibility.

**Weaknesses:**

- My main concern is that the paper repeatedly claims "functional correctness" but the actual verification is purely topological. The rfunc checks truth tables via symbolic propagation, looks for repression motifs, counts edge types. These are all structural checks. Real functional verification would require ODE simulation to confirm that a circuit xhibits the intended dynamic behavior. For example, bistability in toggle switches depends on quantitative parameters, not just topology. It's a gap between what the paper promises and what it delivers. I'd feel much better if the authors either ran ODE simulations on a subset or toned down the claims to "topological correctness."

- No direct comparison with Cello, even though the paper explicitly frames itself as replacing Cello's heuristic solver. Why was this omitted?

- All experiments use only Qwen3 models. At least one other model family would help.

- The "emergence" claim is based on only 5 data points across model sizes. Not enough to distinguish a phase transition from steep continous improvement.

---

> ### Author Rebuttal · Authors · 2026-03-31
>
> We thank Reviewer rnpY for the constructive review, and for acknowledging our efforts toward curriculum ablation, OOD evaluation design, and reproducibility. We address each concern below.
>
> ---
>
> **"Functional correctness" overclaim**
>
> We agree that the manuscript overstated what our verification establishes. We have undertaken a systematic reframing, replacing all unqualified claims of "functional correctness/verification" with "topological correctness" or "task-specific correctness." All changes described below are prepared for the camera-ready version.
>
> Key changes:
> - **Abstract** now reads "topologically correct genetic circuits—a necessary prerequisite for functional behavior" (previously "functionally correct").
> - **§3.2** adds an explicit scope statement: *"We note that our verification is structural/topological rather than dynamic: it confirms correct parts, regulatory connections, and motif presence, but does not simulate quantitative behavior (e.g., ODE-based confirmation of bistability or oscillation period)."*
> - **Level 5** is clarified as evaluating "task-specific correctness criteria via topological analysis" (truth tables via symbolic propagation, motif detection for toggle switches/oscillators).
> - **§7 (Conclusion)** now explicitly acknowledges verification scope as a limitation before the future-work.
>
> To demonstrate the path from topology to dynamics, we add a new appendix section that presents preliminary RLAIF experiments using ML surrogates trained on CLASSIC high-throughput characterization data (121K circuits; see https://www.nature.com/articles/s41586-025-09933-9), showing that topological correctness provides a necessary foundation for quantitative optimization.
>
> ---
>
> **No Cello comparison**
>
> We thank the reviewer for raising an important omission. We have prepared a direct comparison between GenCircuit-RL and Cello, reporting a head-to-head evaluation on Task T8 (gate assignment, i.e., Cello's native task), which mirrors Cello's technology mapping stage. On the shared 111-circuit evaluation set, Cello achieves 88.3% topological correctness versus GenCircuit-RL's 52.3%. Cello's advantage is expected: Cello's simulated annealing solver exploits quantitative Hill function parameters that GenCircuit-RL never receives.
>
> An important context is the abstraction-level difference. Cello is the appropriate tool when a designer has a specific Boolean function, a well-characterized parts library with measured response functions, and needs an optimized gate assignment.  GenCircuit-RL is appropriate when the design task is specified in natural language, involves circuit types beyond Boolean logic (toggle switches, oscillators, feed-forward loops), requires parts not in any characterized library, or demands full SBOL code generation rather than gate assignment alone. GenCircuit-RL additionally handles tasks T1–T7 and T9 (code generation, debugging, NL understanding, de novo design) that lie outside Cello's scope. We frame the two approaches as complementary and suggest hybrid pipelines (GenCircuit-RL for architecture generation, then Cello for quantitative refinement) as future work.
>
> ---
>
> **Only Qwen models**
>
> In response to this review, we have validated our framework on Llama-3.1-8B.
> - SFT → RLVF-H-C improves average TSR by +16.7pp for Llama-3.1-8B (+16.9pp on Procedural-Test, +17.6pp on Cello-OOD), compared to +19.1pp average for Qwen3-8B (+18.8pp Procedural-Test, +20.2pp Cello-OOD), confirming the hierarchical reward benefit is architecture-independent.
> - Curriculum remains essential: without staging, Llama achieves only 11.4% on T6–T7 versus 44.2% with the full curriculum (3.9× factor, consistent with Qwen3's 5.1×).
> - Absolute performance is approximately 8pp lower for Llama, attributable to Qwen3's stronger Python code generation baseline (~10pp gap on standard benchmarks). The relative improvement is consistent (87% of Qwen3's absolute gain).
>
> We are additionally training Gemma-2-9B and will include those results in the camera-ready version, bringing coverage to three independent model families.
>
> ---
>
> **Verifier false positive/negative rates**
>
> Levels 1–3 (execution, validity, structure) are deterministic program checks admitting 0% FP/FN by construction. For remaining levels, the primary risk is false negatives from alternative correct topologies. This risk presents as a "wrong motif" failure mode, were circuits achieving correct function through alternative topology are scored as failures. Our reported TSR therefore underestimates true performance; it is not inflated.
>
> ---
>
> **"Emergence" claim**
>
> We have replaced all "emergence" language with "superlinear scaling" and added an explicit caveat in the camera-ready §5.6: *"While no sharp phase transition is observed, the superlinear scaling indicates that functional reasoning capabilities benefit disproportionately from increased model capacity."*

---

> > ### Author Rebuttal · Reviewer_rnpY · 2026-04-06
> >
> > Thank you for the detailed response. The response has addressed most of my concerns. The original score is high enough.

---

### Decision · Program_Chairs · 2026-04-30

**Decision:**

Accept (regular)

**Comment:**

This paper introduces GenCircuit-RL, a reinforcement learning framework for genetic circuit design, and the SynBio-Reason benchmark. The work contributes a five-level hierarchical verification reward and a four-stage curriculum learning strategy to address sparse feedback in biological code generation.

The reviewers generally agree that the paper is technically sound and well-written. Key strengths include the thoroughness of the ablations, specifically the curriculum learning analysis, and the significance of the new benchmark for the community. The authors were highly responsive during the rebuttal, successfully addressing concerns regarding the scope of functional verification by reframing their claims toward topological correctness. They further strengthened the submission by providing direct comparisons with the Cello baseline and demonstrating the framework's effectiveness across multiple model families.

While one reviewer noted that the individual methodological components like GRPO and curriculum learning are established, their specific composition and application to the domain of synthetic biology are original and provide clear utility. The evidence for compositional reasoning and out-of-distribution generalization is compelling. Given the technical quality and the value of the released resources, the paper meets the criteria for acceptance.